# Efficient Diffusion Models for Symmetric Manifolds

**Oren Mangoubi** [1]   **Neil He** [2]   **Nisheeth K. Vishnoi** [2]

## Abstract

We introduce a framework for designing efficient diffusion models for $d$-dimensional symmetric-space Riemannian manifolds, including the torus, sphere, special orthogonal group and unitary group. Existing manifold diffusion models often depend on heat kernels, which lack closed-form expressions and require either $d$ gradient evaluations or exponential-in-$d$ arithmetic operations per training step. We introduce a new diffusion model for symmetric manifolds with a spatially-varying covariance, allowing us to leverage a projection of Euclidean Brownian motion to bypass heat kernel computations. Our training algorithm minimizes a novel efficient objective derived via Ito's Lemma, allowing each step to run in $O(1)$ gradient evaluations and nearly-linear-in-$d$ ($O(d^{1.19})$) *arithmetic* operations, reducing the gap between diffusions on symmetric manifolds and Euclidean space. Manifold symmetries ensure the diffusion satisfies an "average-case" Lipschitz condition, enabling accurate and efficient sample generation. Empirically, our model outperforms prior methods in training speed and improves sample quality on synthetic datasets on the torus, special orthogonal group, and unitary group.

## 1. Introduction

Recently, denoising diffusion-based methods have achieved significant success in generating synthetic data, including highly realistic images and videos (OpenAI, 2023). Given a dataset $D$ sampled from an unknown probability distribution $\pi$, a diffusion generative model aims to learn a distribution $\nu$ that approximates $\pi$ and generates new samples from $\nu$. While most diffusion models operate in Euclidean space $\mathbb{R}^d$ (Ho et al., 2020; Rombach et al., 2022), several applications require data constrained to a $d$-dimensional non-Euclidean

manifold $\mathcal{M}$, such as robotics (Feiten et al., 2013), drug discovery (Cheng et al., 2021), and quantum physics (Cranmer et al., 2023), where configurations are often represented on symmetric-space manifolds like the torus, sphere, special orthogonal group $\mathrm{SO}(n)$, or unitary group $\mathrm{U}(n)$ where $d \approx n^2$. A common approach enforces manifold constraints by mapping samples from Euclidean space $\mathbb{R}^d$ to $\mathcal{M}$, but this often degrades sample quality due to distortions introduced by the mapping (see Appendix D for details).

To address this, several works have developed diffusion models constrained to non-Euclidean Riemannian manifolds (De Bortoli et al., 2022; Huang et al., 2022; Lou et al., 2024; Zhu et al., 2025; Yim et al., 2023). However, a significant gap remains between the runtime and sampling guarantees of Euclidean and manifold-based diffusion models. For instance, while Euclidean models have a per-iteration runtime of $O(d)$ arithmetic operations and $O(1)$ evaluations of the model's gradient, objectives of manifold diffusion models often require exponential-in-$d$ arithmetic operations, or evaluating Riemannian divergence operators which require $O(d)$ *gradient evaluations*. Reducing this gap, particularly for symmetric manifolds, remains an open challenge.

To understand the technical difficulty, first consider the Euclidean case. A diffusion model consists of two components: a forward process that adds noise over time $T > 0$ until the data is nearly Gaussian, and a reverse process that starts from a Gaussian sample and gradually removes the noise to generate samples approximating the original distribution $\pi$. A discrete-time Gaussian latent variable model is used to approximate the reverse diffusion. In the manifold case, the forward process corresponds to standard Brownian motion on the manifold, and the reverse diffusion is its time-reversal. However, Gaussians are not generally defined on manifolds. To address this, previous works move to continuous time, where infinitesimal updates converge to a Gaussian on the tangent space. The reverse diffusion is then governed by a stochastic differential equation (SDE) involving the manifold's heat kernel. The heat kernel $p_{\tau|b}(\cdot|b)$ represents the density of Brownian motion at time $\tau$, initialized at a point $b$. Training the reverse diffusion model thus requires minimizing an objective function dependent on the heat kernel.

Even in the Euclidean case, the training objective is non-convex, and there are no polynomial-in-dimension runtime

---

[1]Worcester Polytechnic Institute, USA. [2]Yale University, USA. Correspondence to: Oren Mangoubi <omangoubi@gmail.com>, Nisheeth K. Vishnoi <nisheeth.vishnoi@gmail.com>.

*Proceedings of the 42$^{nd}$ International Conference on Machine Learning*, Vancouver, Canada. PMLR 267, 2025. Copyright 2025 by the author(s).

guarantees for the overall training process. However, the closed-form expression of the Euclidean heat kernel allows each training iteration to run in $O(d)$ arithmetic operations with $O(1)$ gradient evaluations. For non-Euclidean manifolds, the lack of a closed-form heat kernel is a major bottleneck. On symmetric manifolds like orthogonal and unitary groups, it can only be computed via inefficient series expansions requiring exponential-in-$d$ runtimes. Alternatively, training with an *implicit* score matching (ISM) objective requires evaluating a Riemannian divergence, incurring $O(d)$ *gradient evaluations* per iteration. Due to these challenges, approximations are often used, degrading sample quality. Moreover, on manifolds with nonzero curvature, such as orthogonal and unitary groups, standard Brownian motion cannot be obtained via any projection from $\mathbb{R}^d$. As a result, prior works rely on numerical SDE or ODE solvers to sample the forward diffusion at each evaluation of the training objective, introducing significant computational overhead.

In addition to denoising diffusions, several other generative models on manifolds leverage probability flows, including Moser flows (Rozen et al., 2021) and Riemannian normalizing flows (Mathieu & Nickel, 2020; Ben-Hamu et al., 2022). More recent approaches include flow matching (Chen et al., 2024) and mixture models of Riemannian bridge processes (Jo & Hwang, 2024). These models often achieve sample quality comparable to denoising diffusion models on manifolds but frequently face similar computational bottlenecks.

**Our contributions.** We study the problem of designing efficient diffusion models when $\mathcal{M}$ is a symmetric-space manifold, such as the torus $\mathbb{T}_d$, sphere $\mathbb{S}_d$, special orthogonal group $\mathrm{SO}(n)$, and unitary group $\mathrm{U}(n)$, where $d \approx n^2$, as well as direct products of these manifolds, such as the special Euclidean group $\mathrm{SE}(n) \cong \mathbb{R}^n \times \mathrm{SO}(n)$. We present a new training algorithm (Algorithm 1) for these manifolds, achieving per-iteration runtimes of $O(d)$ arithmetic operations for $\mathbb{T}_d$ and $\mathbb{S}_d$, and $O(d^{\frac{\omega}{2}}) \approx O(d^{1.19})$ for $\mathrm{SO}(n)$ and $\mathrm{U}(n)$, where $\omega \approx 2.37$ is the matrix multiplication exponent. Each iteration requires only $O(1)$ gradient evaluations of a model for the drift and covariance terms of the reverse process. This significantly improves on previous methods (see Table 1). For $\mathrm{SO}(n)$ and $\mathrm{U}(n)$, our approach reduces gradient evaluations by a factor of $d$ and achieves an exponential-in-$d$ improvement in arithmetic operations, bringing runtime closer to the Euclidean case. We also provide a sampling algorithm (Algorithm 2) with guarantees on accuracy and runtime. Given an $\varepsilon$-minimizer of our training objective, the algorithm attains an $\varepsilon \times \mathrm{poly}(d)$ bound on total variation distance accuracy in $\mathrm{poly}(d)$ runtime (Theorem 2.2), improving on the sampling accuracy bounds of (De Bortoli et al., 2022), which are not polynomial in $d$. Theorem 2.2 holds for general manifolds satisfying an average-case Lipschitz condition (Assumption 2.1). Using techniques from random matrix theory, we prove this

condition holds for the manifolds of interest (Lemma B.4).

Our paper introduces several new ideas. For our training result: (i) We define a novel diffusion on $\mathcal{M}$. Unlike previous works, our diffusion incorporates a spatially varying covariance term to account for the manifold's nonzero curvature. As a result, our forward diffusion can be computed as a projection $\varphi$ of Brownian motion in $\mathbb{R}^d$ onto $\mathcal{M}$, which can be efficiently computed via singular value decomposition when $\mathcal{M}$ is $\mathrm{SO}(n)$ or $\mathrm{U}(n)$. This enables efficient sampling from our forward diffusion in a simulation-free manner—without SDE or ODE solvers—by directly sampling from a Gaussian in $\mathbb{R}^d$ and projecting onto $\mathcal{M}$. (ii) We introduce a new training objective that bypasses the need to compute the manifold's heat kernel. By applying Itô's Lemma from stochastic calculus, we project the SDE for a reverse diffusion in Euclidean space onto $\mathcal{M}$. The drift term of the resulting SDE is an expectation of the Euclidean heat kernel. Since the Euclidean kernel has a closed-form expression and the projection $\varphi$ can be computed efficiently, we evaluate the objective in time $O(d^{\frac{\omega}{2}})$. (iii) While our covariance term is a $d \times d$ matrix, we show that its structure, arising from manifold symmetries, allows it to be computed in time $O(d^{\frac{\omega}{2}})$—sublinear in its $d^2$ entries.

For the sampling result, we show that the reverse SDE on the manifold $\mathcal{M}$ is deterministically Lipschitz, provided the projection map satisfies our *average-case* Lipschitz condition (Lemma B.4). Since the projection introduces a spatially varying covariance in the SDE on $\mathcal{M}$, prior techniques based on Girsanov's theorem cannot be used to bound accuracy. To address this, we develop an optimal transport-based approach, leading to a novel probabilistic coupling argument that establishes the desired accuracy and runtime bounds. This approach differs fundamentally from previous proofs in Euclidean space (Chen et al., 2023b;a; Cheng et al., 2022; Benton et al., 2024) and manifold-based diffusion models (De Bortoli et al., 2022), which rely on Girsanov's theorem.

Empirically, our model trains significantly faster per iteration than previous manifold diffusion models on $\mathrm{SO}(n)$ and $\mathrm{U}(n)$, staying within a factor of 3 of Euclidean diffusion models even in high dimensions ($d > 1000$) (Table 3). Moreover, our model improves the quality of generated samples compared to previous diffusion models, achieving improved C2ST and likelihood scores and visual quality when trained on various synthetic datasets on wrapped Gaussian (mixture) models and quantum evolution operators constrained to the torus, $\mathrm{SO}(n)$, and $\mathrm{U}(n)$ (Table 2 and Figure 1). The magnitude of the improvements in runtime and sample quality increases with dimension.

Thus, our results reduce the gap in training runtime and sample quality between diffusion models on symmetric manifolds and Euclidean space, contributing towards the goal of developing efficient diffusion models on constrained spaces.

## 2. Results

We begin by describing the geometric setup, projection framework, and key assumptions used in our training and sampling algorithms. Notation is summarized in Appendix G, and relevant background on Riemannian geometry and manifold diffusions is provided in Appendix H.

### 2.1. Problem setup and projection framework

For a manifold $\mathcal{M}$, we are given a projection map $\varphi \equiv \varphi_{\mathcal{M}} : \mathbb{R}^d \to \mathcal{M}$ from a Euclidean space $\mathbb{R}^d$ of dimension $d = O(\dim(\mathcal{M}))$, and a restricted-inverse map $\psi \equiv \psi_{\mathcal{M}} : \mathcal{M} \to \mathbb{R}^d$ such that $\varphi(\psi(x)) = x$ for all $x \in \mathcal{M}$. We sometimes abuse notation and refer to the manifold's dimension as $d$ rather than "$O(d)$", as this does not change our runtime and accuracy guarantees beyond a small constant factor. Denote by $\mathcal{T}_x\mathcal{M}$ the tangent space of $\mathcal{M}$ at $x$. For our sampling algorithm (Algorithm 2), we assume access to the exponential map $\exp(x, v)$ on $\mathcal{M}$ for any $x \in \mathcal{M}$ and $v \in \mathcal{T}_x\mathcal{M}$. In the setting where $\mathcal{M}$ is a symmetric space, there are closed-form expressions which allow one to efficiently and accurately compute the exponential map. For instance, on $\mathrm{SO}(n)$ or $\mathrm{U}(n)$, the geodesic is given by the matrix exponential and can be computed in $O(n^\omega) = O(d^{\frac{\omega}{2}}) \approx O(n^{1.19})$ arithmetic operations. We are also given a dataset $D \subseteq \mathcal{M}$ sampled from $\pi$ with support on $\mathcal{M}$. These projection maps are efficient to compute and will be used throughout our framework for both training and sampling on $\mathcal{M}$.

We set $\varphi : \mathbb{R}^d \to \mathbb{R}^d$ and $\psi : \mathbb{R}^d \to \mathbb{R}^d$ as identity maps when $\mathcal{M} = \mathbb{R}^d$. For the torus $\mathbb{T}_d$, $\varphi(x)[i] = x[i] \mod 2\pi$ maps points to their angles, and $\psi$ is its inverse on $[0, 2\pi)^d$. For the sphere $\mathbb{S}_d$, $\varphi(x) = \frac{x}{\|x\|}$, and $\psi$ embeds the unit sphere into $\mathbb{R}^d$. For the unitary group $\mathrm{U}(n)$ (and special orthogonal group $\mathrm{SO}(n)$), we first define a map $\hat{\varphi}$ which takes each upper triangular matrix $X \in \mathbb{C}^{n \times n}$ (or $X \in \mathbb{R}^{n \times n}$), computes the spectral decomposition $U^*\Lambda U$ of $X + X^*$, and outputs $\hat{\varphi}(X) = U$. The spectral decomposition is unique only up to multiplication of each eigenvector $u_j$ by a root of unity $e^{i\phi_j}$, where the phases $(\phi_1, \cdots, \phi_n)$ lie on the $n$-dimensional torus $\mathbb{T}_n$ (or, in the real case, a subset of the torus). Thus, we define the projection map $\varphi : \mathbb{C}^{n \times n} \times \mathbb{R}^n \to \mathcal{M}$ to be the concatenated map $\varphi = (\hat{\varphi}, \varphi_{\mathbb{T}_n})$ where $\varphi_{\mathbb{T}_n}$ is the map defined above for the torus. The restricted-inverse map $\psi$ takes each matrix $U \in \mathcal{M}$, computes $U^*\Lambda U$ where $\Lambda = \frac{1}{n}\mathrm{diag}(n, n-1, \ldots, 1)$, scales the diagonal by $\frac{1}{2}$, and outputs the upper triangular entries of the result. For all of the above maps, $\psi(\mathcal{M})$ is contained in a ball of radius $\mathrm{poly}(d)$. Our general results hold under this assumption on $\psi$. For manifolds $\mathcal{M} = \mathcal{M}_1 \times \mathcal{M}_2$, which are direct products of manifolds $\mathcal{M}_1$ and $\mathcal{M}_2$, where one is given maps $\varphi_1, \psi_1$ for $\mathcal{M}_1$ and $\varphi_2, \psi_2$ for $\mathcal{M}_2$, one can use the concatenated maps $\varphi = (\varphi_1, \varphi_2)$ and $\psi = (\psi_1, \psi_2)$.

### 2.2. Training algorithm and runtime analysis

We now describe our training procedure and its computational benefits for symmetric manifolds.

**Training.** We give an algorithm (Algorithm 1) that minimizes a nonconvex objective function via stochastic gradient descent. This algorithm outputs trained models $f(x, t)$ and $g(x, t)$ for the drift and covariance terms of our reverse diffusion, and passes these trained models as inputs to our sample generation algorithm (Algorithm 2). We show that the time per iteration of Algorithm 1 is dominated by the computation of the objective function gradient (Lines 12 and 14 in Algorithm 1), which requires calculating the gradient of the projection map $\nabla\varphi$ as well as the model gradients $\nabla_\theta f$ and $\nabla_\phi g$, where $\theta$ and $\phi$ are the model parameters of $f$ and $g$. When $\mathcal{M}$ is one of the aforementioned symmetric manifolds, $\nabla\varphi$ can be computed at each iteration within error $\delta$ in $O(n^\omega \log(\frac{1}{\delta})) = O(d^{\omega/2} \log(\frac{1}{\delta}))$ arithmetic operations in the case of the special orthogonal group $\mathrm{SO}(n)$ or unitary group $\mathrm{U}(n)$, using the singular value decomposition of an $n \times n$ matrix, or in $O(d \log(\frac{1}{\delta}))$ operations for the sphere or torus. See Section 3.1 and Appendix E for details.

This significantly improves the per-iteration runtime of training diffusion models on symmetric manifolds (Table 1). For instance, it achieves exponential-in-$d$ savings in arithmetic operations compared to Riemannian Score-based Generative Models (RSGM) (De Bortoli et al., 2022), as RSGM requires summing $\Omega(2^d)$ terms in truncated heat kernel expansions for manifolds like the torus, sphere, orthogonal, or unitary groups, while our approach avoids this complexity. If RSGM is instead trained with an *implicit* score matching objective (ISM), which includes a Riemannian divergence term that requires $O(d)$ gradient evaluations, our model achieves a factor of $d$ improvement in the number of *gradient evaluations*. Similarly, we get a factor of $d$ improvement in gradient evaluations over Trivialized Momentum Diffusion Models (TDM) (Zhu et al., 2025), which also rely on ISM objectives, on manifolds like the orthogonal or unitary group. It also improves upon Scaling Riemannian Diffusion (SCRD) (Lou et al., 2024), where heat kernel computations for orthogonal or unitary groups involve expansions with $\Omega(2^d)$ terms (SCRD does not provide an ISM objective).

Additionally, while RSGM and Riemannian Diffusion Models (RDM) (Huang et al., 2022) use deterministic heat kernel approximations, these are asymptotically biased with fixed error bounds. Stochastic approximations to the implicit objective introduce dimension-dependent noise (Lou et al., 2024). Our method improves the accuracy dependence from polynomial to logarithmic in $\frac{1}{\delta}$. Unlike solvers for SDEs or ODEs, which require polynomial-in-$\frac{1}{\delta}$ iterations, our forward diffusion adds a Gaussian vector and projects onto the manifold, achieving high accuracy with only logarithmic cost in $\frac{1}{\delta}$.

Table 1: Arithmetic operations plus model gradient evaluations to compute objective function's gradient within any error $\delta$ at each iteration of training algorithm, on the unitary group $\mathrm{U}(n)$, special orthogonal group $\mathrm{SO}(n)$, sphere, or torus, of dimension $d \equiv n^2$ (number of grad. eval. depends on algorithm but not on manifold).

| Algorithm | Grad. eval. | Arithmetic Operations | | |
|---|---|---|---|---|
| | | SO(n) or U(n) | Sphere | Torus |
| RSGM (heat ker.) | 1 | $2^d + \mathrm{poly}(d, \frac{1}{\delta})$ | same | same |
| RSGM (ISM) | $d$ | $\mathrm{poly}(d, \frac{1}{\delta})$ | same | same |
| TDM (heat ker.) | 1 | —— | —— | $d\log(\frac{1}{\delta})$ |
| TDM (ISM) | $d$ | $\mathrm{poly}(d, \frac{1}{\delta})$ | $\mathrm{poly}(d, \frac{1}{\delta})$ | $d\log(\frac{1}{\delta})$ |
| RDM | $d$ | $\mathrm{poly}(d, \frac{1}{\delta})$ | same | same |
| SCRD | 1 | $2^d + \mathrm{poly}(d, \frac{1}{\delta})$ | $\mathrm{poly}(d, \frac{1}{\delta})$ | $d\log(\frac{1}{\delta})$ |
| This paper | 1 | $d^{\frac{\omega}{2}}\log(\frac{1}{\delta})$ | $d\log(\frac{1}{\delta})$ | $d\log(\frac{1}{\delta})$ |

## 2.3. Sampling algorithm and theoretical guarantees

Next, we present our sampling algorithm, which uses the trained models to simulate the reverse diffusion process.

**Sampling procedure.** Our training algorithm (Algorithm 1) outputs trained models $f(x, t)$ and $g(x, t)$ for the drift and covariance terms of our reverse diffusion. We then use these models to generate samples. First, we sample a point $z$ from the stationary distribution of the Ornstein-Uhlenbeck process $Z_t$ on $\mathbb{R}^d$, which is Gaussian distributed. Next, we project this point $z$ onto the manifold to obtain a point $y = \varphi(z)$, and solve the SDE $\mathrm{d}Y_t = f(Y_t, t)\mathrm{d}t + g(Y_t, t)\mathrm{d}B_t$ given by our trained model for the reverse diffusion's drift and covariance over the time interval $[0, T]$, starting at the initial point $y$. To simulate this SDE, we can use any off-the-shelf numerical SDE solver, which takes as input the trained model for $f$ and $g$, and the exponential map on $\mathcal{M}$. We give one such solver in Algorithm 2, and prove guarantees for the accuracy of the samples it generates, and its runtime, in Theorem 2.2. Our guarantees assume the trained models $f(x, t)$ and $g(x, t)$ we hand to this solver minimize our training objective within some error $\varepsilon > 0$.

**Symmetry and forward diffusion structure.** Our theoretical guarantees hold when $\mathcal{M}$ satisfies a symmetry property and $\varphi$ satisfies an "average-case" Lipschitz condition (Assumption 2.1). This symmetry property requires that each point $z \in \mathbb{R}^d$ can be parametrized as $z \equiv z(U, \Lambda)$ where $U = \varphi(z) \in \mathcal{M}$ and $\Lambda \equiv \Lambda(z) \in \mathcal{A}$ for some $\mathcal{A} \subseteq \mathbb{R}^{d-\dim(\mathcal{M})}$ is another parameter. For instance, on the sphere, $U = \frac{z}{\|z\|}$ is the projection onto the sphere, and $\Lambda = \|z\|$ is the distance to the origin. For $\mathrm{SO}(n)$ or $\mathrm{U}(n)$, the parametrization comes from the spectral decomposition $z = U\Lambda U^*$, where $U \in \mathcal{M}$ and $\Lambda$ is a diagonal matrix. On the torus, $U = \varphi(x)$ is the projection onto the torus, and $\Lambda \in 2\pi\mathbb{Z}^d$. $Z_t$, $t \geq 0$, is the Ornstein-Uhlenbeck process on $\mathbb{R}^d$, $X_t := \varphi(Z_t)$ is our forward diffusion process on $\mathcal{M}$, and $Y_t := X_{T-t}$ its time-reversal (see Section 3.1). This structure ensures that the reverse diffusion inherits well-behaved properties from the Euclidean process.

**Average-case Lipschitzness.**

**Assumption 2.1** (**Average-case Lipschitzness**). $\forall t \in [0, T]$ there exists $\Omega_t \subseteq \mathbb{R}^d$, whose indicator function $\mathbb{1}_{\Omega_t}(x)$ depends only on $\Lambda \equiv \Lambda(x)$, for which $\mathbb{P}(Z_t \in \Omega_t \ \forall\ t \in [0, T]) \geq 1 - \alpha$. For every $x \in \Omega_t$ we have $\|\nabla\varphi(x)\|_{2\to2} \leq L_1$, $\|\frac{\mathrm{d}}{\mathrm{d}U}\nabla\varphi(x)\|_{2\to2} \leq L_1$, $\|\nabla^2\varphi(x)\|_{2\to2} \leq L_2$, and $\|\frac{\mathrm{d}}{\mathrm{d}U}\nabla\varphi(x)\|_{2\to2} \leq L_2$. Moreover, $\|\frac{\mathrm{d}}{\mathrm{d}U}x\|_{2\to2} \leq \|x\|_2$.

Here $\|\cdot\|_{2\to2}$ denotes the operator norm and $\frac{d}{dU}x$ denotes the derivative of the parameterization of $x = x(U, \Lambda)$ with respect to $U \in \mathcal{M}$. This assumption allows us to show that the projected reverse diffusion is well-posed and numerically stable, enabling sample quality guarantees.

**Verifying the assumption on common manifolds.** We choose projection maps $\varphi$ that satisfy Assumption 2.1 with small Lipschitz constants. For example, for $\mathbb{T}_d$, $\varphi(x)[i] = x[i]\mathrm{mod}2\pi$, $i \in [d]$ is 1-Lipschitz on all $\mathbb{R}^d$, trivially satisfying the assumption. For the sphere, $\varphi(x) = \frac{x}{|x|}$ is 2-Lipschitz outside a ball of radius $\frac{1}{2}$ around the origin, where the forward diffusion remains with high probability $1 - O(2^{-d})$. For $\mathrm{U}(n)$ (or $\mathrm{SO}(n)$), $\varphi(X)$, which computes the spectral decomposition $U^*\Lambda U$ of $X + X^*$, has derivatives with magnitude bounded by the inverse eigenvalue gaps $\frac{1}{\lambda_i - \lambda_j}$. While singularities occur at points with duplicate eigenvalues, random matrix theory shows that eigengaps are w.h.p. bounded below by $\frac{1}{\mathrm{poly}(d)}$, ensuring $\varphi$ satisfies the average-case Lipschitz assumption. For the unitary group, we show that Assumption 2.1 holds for $L_1 = O(d^{1.5}\sqrt{T}\alpha^{-\frac{1}{3}})$ and $L_2 = O(d^2T\alpha^{-\frac{2}{3}})$ (Lemma B.4). For the sphere, it holds for $L_1 = L_2 = O(\alpha^{-\frac{1}{d}})$. For the torus, it holds for $L_1 = L_2 = 1$. These bounds, derived in Appendix E, imply the assumption holds with high probability under standard random matrix models.

**Theoretical guarantees.** We denote by $\psi(\mathcal{M}) := \psi(x) \ x \in \mathcal{M} \subseteq \mathbb{R}^d$ the pushforward of $\mathcal{M}$ w.r.t. $\psi$.

**Theorem 2.2** (**Accuracy and runtime of sampling algorithm**). *Let $\varepsilon > 0$, and suppose that $\varphi : \mathbb{R}^d \to \mathcal{M}$ satisfies Assumption 2.1 for some $L_1, L_2 \leq \mathrm{poly}(\mathrm{d})$ and $\alpha \leq \varepsilon$, and $\psi(\mathcal{M})$ is bounded by a ball of radius $\mathrm{poly}(d)$. Suppose that $\hat{f}$ and $\hat{g}$ are outputs of Algorithm 2, and that $\hat{f}$ and $\hat{g}$ minimize our training objective for the target distribution $\pi$ with objective function value $< \varepsilon$. Then Algorithm 2, with inputs $\hat{f}$ and $\hat{g}$, outputs a generated sample whose probability distribution $\nu$ satisfies*

$$\|\nu - \pi\|_{\mathrm{TV}} < O(\varepsilon(d^3 L_1 + d^2 L_2)\log\frac{d}{\varepsilon}) = \tilde{O}(\varepsilon \times \mathrm{poly}(d)).$$

*Moreover, Algorithm 2, takes*

$$O((d^4 L_1 + d^2 L_2)\log\left(\frac{d}{\varepsilon}\right)) = \mathrm{poly}(d) \times \log\left(\frac{d}{\varepsilon}\right)$$

*iterations, where each iteration requires one evaluation of $\hat{f}$ and $\hat{g}$, one evaluation of the exponential map on $\mathcal{M}$, plus $O(d)$ arithmetic operations.*

Plugging in our bounds on the average-case Lipschitz constants in the case of the torus, sphere, $\mathrm{SO}(n)$, and $\mathrm{U}(n)$ (Lemma B.4) into Theorem 2.2, we obtain the following guarantees for the accuracy and runtime of our sampling algorithm for these symmetric manifolds:

**Corollary 2.3.** *Suppose that $\mathcal{M}$ is $\mathbb{T}_d$, $\mathbb{S}_d$, $\mathrm{SO}(n)$, or $\mathrm{U}(n)$ with $n = \sqrt{d}$. Suppose that $\varphi$ and $\psi$ are chosen as specified above for these manifolds. Suppose that $\hat{f}$ and $\hat{g}$ are outputs of Algorithm 2, and that $\hat{f}$ and $\hat{g}$ minimize our training objective for the target distribution $\pi$ with objective function value $< \varepsilon$. Then Algorithm 2, with inputs $\hat{f}$ and $\hat{g}$, outputs a generated sample whose probability distribution $\nu$ satisfies*

$$\|\nu - \pi\|_{\mathrm{TV}} \leq O(\varepsilon \times d^6 \log\left(\tfrac{d}{\varepsilon}\right))$$

*for the torus and sphere, and*

$$\|\nu - \pi\|_{\mathrm{TV}} < O(\varepsilon \times d^9 \log\left(\tfrac{d}{\varepsilon}\right))$$

*for $\mathrm{SO}(n)$ and $\mathrm{U}(n)$. Moreover, Algorithm 2, takes $O(d^4 \log\left(\tfrac{d}{\varepsilon}\right))$ iterations for the torus and sphere, and $O(d^{5.5} \log\left(\tfrac{d}{\varepsilon}\right))$ iterations for $\mathrm{SO}(n)$ and $\mathrm{U}(n)$. Here each iteration requires one evaluation of $\hat{f}$ and $\hat{g}$, one evaluation of the exponential map on $\mathcal{M}$, plus $O(d)$ arithmetic operations.*

**Comparison with prior work.** Theorem 2.2 improves on the accuracy and runtime guarantees for sampling of (De Bortoli et al., 2022) when $\mathcal{M}$ is one of the aforementioned symmetric manifolds, since their accuracy and runtime bounds for sampling are not polynomial in the dimension $d$ (for instance, the "constant" term $C \equiv C(\mathcal{M}, d)$ in (De Bortoli et al., 2022) has an unspecified dependence on the manifold and its dimension). Finally, we note that (Lou et al., 2024; Huang et al., 2022) do not provide guarantees on the accuracy and runtime of their sampling algorithm, and that the runtime bounds for the sampling algorithm in (Zhu et al., 2025) are not polynomial in dimension. Improving the dependency on dimension remains an open question for future work.

**Extension beyond symmetric manifolds.** While our theoretical guarantees focus on symmetric manifolds, the algorithm itself applies more broadly. In Appendix F, we describe how projection maps $\varphi$ and exponential maps can be constructed for certain non-smooth or non-symmetric spaces, such as convex polytopes. Although proving Lipschitz properties in these settings is more subtle due to curvature or boundary singularities, our framework may still apply empirically. Extending theoretical guarantees to such general manifolds remains a promising direction for future work.

An overview of the proof is given in Appendix A; the full proof appears in Appendix B.

## 3. Algorithm derivation and proof highlights

### 3.1. Derivation of training & sampling algorithm

Given a standard Brownian motion $B_t$ in $\mathbb{R}^d$, a $\mu : \mathbb{R}^d \to \mathbb{R}^d$ and $R : \mathbb{R}^d \to \mathbb{R}^{d \times d}$, a stochastic process $X_t$ satisfies the SDE $\mathrm{d}X_t = \mu(X_t)\mathrm{d}t + R(X_t)\mathrm{d}B_t$ with initial condition $x \in \mathbb{R}^d$ if $X_t = x + \int_0^t \mu(X_s)\mathrm{d}s + \int_0^t R(X_s)\mathrm{d}B_s$.

**Lemma 3.1 (Itô's Lemma).** *Let $\psi : \mathbb{R}^d \to \mathbb{R}^k$ be a second-order differentiable function, and let $X(t) \in \mathbb{R}^d$ be an Itô diffusion. Then for all $t \geq 0$ and all $i \in [k]$, we have $\mathrm{d}\psi(X_t)[i] = \nabla\psi(X_t)[i]^\top \mathrm{d}X_t + \frac{1}{2}\mathrm{d}X_t^\top \nabla^2\psi(X_t)[i]\mathrm{d}X_t$.*

The *transition kernel* $p_{t|\tau}(y|x)$ is the probability (density) that $X$ takes the value $y$ at time $t$ conditional on $X$ taking the value $x$ at time $\tau$. Given an initial distribution $\pi$, the probability density at time $t$ is $p_t(x) = \int_{\mathcal{M}} p_{t|0}(x|z)\pi(z)\mathrm{d}z$. For any diffusion $X_t$, $t \in [0, T]$, its *time-reversal* $Y_t$ is the stochastic process such that $Y_t = X_{T-t}$ for $t \in [0, T]$. $Y_t$ is also a diffusion, governed by an SDE. In the special case where $X_t$ has identity covariance, $\mathrm{d}X_t = b(X_t)\mathrm{d}t + \mathrm{d}B_t$, the reverse diffusion satisfies (Anderson, 1982)

$$\mathrm{d}Y_t = -b(Y_t)\mathrm{d}t + \nabla \log p_t(Y_t)\mathrm{d}t + \mathrm{d}B_t. \tag{1}$$

One can also define diffusions on Riemannian manifolds, in which case $\mathrm{d}B_t$ is the derivative of Brownian motion on the tangent space (see (Hsu, 2002)). Below we show the key steps in deriving our diffusion model, training algorithm (Algorithm 1), and sampling algorithm (Algorithm 2).

**Forward diffusion.** Let $Z_t$ be a diffusion on $\mathbb{R}^d$ initialized at $q_0 = \psi(\pi)$. We choose $Z_t$ to be the Ornstein-Uhlenbeck process, $\mathrm{d}Z_t = -\frac{1}{2}Z_t\mathrm{d}t + \mathrm{d}B_t$, whose stationary distribution is $N(0, I_d)$. $Z_t$ is easy to sample as it has a closed-form Gaussian transition kernel $q_{t|\tau}$. Let $X_t := \varphi(Z_t)$, the projection of $Z_t$ onto $\mathcal{M}$. $X_t$ is our model's forward diffusion.

**Reverse diffusion SDE.** Let $Y_t := X_{T-t}$ denote the time-reversal of $X_t$. $Y_t$ is a diffusion on $\mathcal{M}$, and its distribution at time $T$ equals the target distribution $\pi$. It follows the SDE:

$$\mathrm{d}Y_t = f^\star(Y_t, t)\mathrm{d}t + g^\star(Y_t, t)\mathrm{d}B_t, \tag{2}$$

for some functions $f^\star(x, t) : \mathcal{M} \times [0, T] \to \mathcal{T}_x\mathcal{M}$ and $g^\star(x, t) : \mathcal{M} \to \mathcal{T}_x\mathcal{M} \times \mathcal{T}_x\mathcal{M}$. Here $\mathrm{d}B_t$ is the derivative of standard Brownian motion on $\mathcal{M}$'s tangent space. We write $\mathrm{d}B_t \equiv \mathrm{d}B_t^x$ when $x \in \mathcal{M}$ is clear from context.

We cannot directly apply (1) to derive a tractable SDE for the reverse diffusion $Y_t$ on $\mathcal{M}$, as the transition kernel $p_{t|\tau}$ of the forward diffusion $X_t$ on $\mathcal{M}$ lacks a closed form expression. Instead, we first use (1) to obtain an SDE for the reverse diffusion of $Z_t \in \mathbb{R}^d$, $\mathrm{d}H_t = (H_t/2 + 2\nabla \log q_{T-t}(H_t))\mathrm{d}t + \mathrm{d}B_t$. We use Itô's Lemma to project this SDE onto $\mathcal{M}$, giving an SDE for $Y_t$ (see Appendix B.1),

$$\mathrm{d}Y_t = \mathbb{E}[(\nabla\varphi(H_t)^\top + \tfrac{\mathrm{d}H_t^\top}{2}\nabla^2\varphi(H_t))\mathrm{d}H_t \big| \varphi(H_t) = Y_t] \tag{3}$$

**Training algorithm's objective function.** From (3), we show one can train a model $f, g$ for $f^\star, g^\star$ by solving an optimization problem (Lemma B.2). Here, $f, g \in \mathcal{C}(\mathcal{M} \times [0, T], \mathcal{T}_x \mathcal{M})$ are continuous functions from $\mathcal{M} \times [0, T]$ to the tangent space $\mathcal{T}_x \mathcal{M}$ and $t \sim \mathrm{Unif}[0, 1]$. $J_\varphi$ denotes the Jacobian of $\varphi$, that is, $J_\varphi(x) : \mathbb{R}^d \to \mathcal{T}_{\varphi(x)} \mathcal{M}$ is the linear operator which maps any $v \in \mathbb{R}^d$ to the derivative of $\varphi$ in the direction of $v$.

$$\min_f \mathbb{E}_t \mathbb{E}_{b \sim \pi}[\|(\nabla \varphi(Z_{T-t}))^\top \tfrac{Z_{T-t} - \psi(b)e^{-(T-t)/2}}{e^{-(T-t)} - 1}$$
$$+ \tfrac{1}{2} \mathrm{tr}(\nabla^2 \varphi(Z_{T-t})) - f(\varphi(Z_{T-t}), t)\|^2 | Z_0 = \psi(b)], (4)$$

$$\min_g \mathbb{E}_t \mathbb{E}_{b \sim \pi}[\|J_\varphi(Z_{T-t}))^\top J_\varphi(Z_{T-t})$$
$$- g(\varphi(Z_{T-t}), t)^2\|_F^2 | Z_0 = \psi(b)]. \quad (5)$$

**Sublinear computation of training objective.** For manifolds with non-zero curvature, such as the sphere, $\mathrm{SO}(n)$, and $\mathrm{U}(n)$, our forward and reverse diffusions differ from prior works and incorporate a spatially-varying covariance term to account for curvature. This allows the forward diffusion to be computed as a projection $\varphi$ of the Ornstein-Uhlenbeck process in $\mathbb{R}^d \equiv \mathbb{R}^{n \times n}$ (or $\mathbb{C}^{n \times n}$) onto the manifold. For $\mathrm{SO}(n)$ or $\mathrm{U}(n)$, $\varphi$ is computed by one singular value decomposition $U^* \Lambda U$ of the Gaussian matrix $Z_{T-t} + Z_{T-t}^*$, requiring $O(n^\omega) = O(d^{\frac{\omega}{2}})$ arithmetic operations, where $d = \Theta(n^2)$ is the manifold dimension. This enables computation of the drift term's gradient (4) in $O(d^{\frac{\omega}{2}})$ arithmetic operations and one gradient evaluation of $f$.

To train the reverse diffusion's SDE, we also need to model the covariance term (5), a $d \times d = n^2 \times n^2$ matrix. To achieve a per-iteration runtime sublinear in the $d^2 = n^4$ matrix entries, we leverage the special structure of the covariance matrix, which arises from the manifold's symmetries. For example, the forward diffusion $U(t) \in \mathrm{SO}(n)$ (or $U(t) \in \mathrm{U}(n)$) is governed by the following system of SDEs:

$$\mathrm{d}u_i(t) = \sum_{j \in [n] \setminus \{i\}} (\alpha_{ij}(t) \mathrm{d}B_{ij} u_j(t) - \tfrac{\beta_{ij}(t)}{2} u_i(t)) \mathrm{d}t \quad (6)$$

where $\alpha_{ij}(t) := \mathbb{E}[1/(\lambda_i(t) - \lambda_j(t)) | \varphi(Z_t) = U(t)]$ and $\beta_{ij}(t) := \mathbb{E}[1/(\lambda_i(t) - \lambda_j(t))^2 | \varphi(Z_t) = U(t)] \forall i, j \in [n]$. To train a model for this covariance term with sublinear runtime, we exploit the symmetries of the underlying group. These symmetries ensure the covariance term in (6) is fully determined by $n^2$ scalar terms $\alpha_{ij}(t)$ for $i, j \in [n]$ and the $n \times n$ matrix $U$. Thus, it suffices to train a model $\mathcal{A}(U, t) \in \mathbb{R}^{n \times n}$ for these $n^2$ terms by minimizing the objective $\|\mathcal{A}(U, t) - A\|_F^2$, where $A$ is the $n \times n$ matrix with entries $A_{ij} = 1/(\lambda_i(t) - \lambda_j(t))$, and $\lambda_i(t)$ is the $i$th diagonal entry of $\Lambda \equiv \Lambda(t)$. A similar method applies to efficiently train the covariance term for the sphere (see Appendix E).

**Sampling algorithm.** To (approximately) sample from $\pi$, we approximate the drift and covariance terms of the reverse diffusion (2) via trained models $\hat{f}, \hat{g}$ obtained by solving (4)

(in practice, $\hat{f}, \hat{g}$ are neural networks $\hat{f}_\theta, \hat{g}_\phi$, and $\theta, \phi$ outputs of Algorithm 1). We initialize this SDE at $\varphi(N(0, I_d))$, the pushforward of $N(0, I_d)$ onto $\mathcal{M}$ with respect to $\varphi$.

$$\mathrm{d}\hat{Y}_t = \hat{f}(\hat{Y}_t, t)\mathrm{d}t + \hat{g}(\hat{Y}_t, t)\mathrm{d}B_t, \quad \hat{Y}_0 \sim \varphi(N(0, I_d)). \quad (7)$$

To generate samples, we numerically simulate the SDE (7) for $\hat{Y}_T$ by discretizing it with a small time-step $\Delta > 0$:

$$\hat{y}_{i+1} = \exp(\hat{y}_i, \ \hat{f}(\hat{y}_i, t)\Delta + \hat{g}(\hat{y}_i, t)\sqrt{\Delta}\xi_i), \quad (8)$$

$i \in \{0, \ldots, T/\Delta\}$, initialized at $\hat{y}_0 \sim \varphi(N(0, I_d))$.

---

**Algorithm 1** Training algorithm

---

**Input:** A way to compute a "projection" $\varphi : \mathbb{R}^d \to \mathcal{M}$, and its gradient
**Input:** A way to compute a map $\psi : \mathcal{M} \to \mathbb{R}^d$ s.t. $\varphi(\psi(x)) = x$ $\forall x \in \mathcal{M}$
**Input:** Dataset $D = \{x_0^1, \ldots, x_0^m\} \subseteq \mathcal{M}$. Hyperparam. $T > 0$
**Input:** Models $f_{\hat{\theta}} : \mathcal{M} \times [0, T] \to \mathcal{T}\mathcal{M}$ and $g_{\hat{\phi}} : \mathcal{M} \times [0, T] \to \mathcal{T}\mathcal{M} \times \mathcal{T}\mathcal{M}$. $\hat{\theta} \in \mathbb{R}^{a_1}, \hat{\phi} \in \mathbb{R}^{a_2}$ denote trainable parameters
**Input:** Initial parameters $\theta_0 \in \mathbb{R}^{a_1}, \phi_0 \in \mathbb{R}^{a_2}$
**Input:** Hyperparameters: Number of stochastic gradient descent iterations $r \in \mathbb{N}$. Step size $\eta > 0$, batch size $\mathfrak{b}$
1: Define, $\forall \hat{\theta} \in \mathbb{R}^{a_1} \ \hat{z} \in \mathbb{R}^d, \ b, x \in \mathcal{M}, \ \hat{t} \in [0, T]$, the objective function $F(\hat{\theta}; b, \hat{z}, \hat{x}, \hat{t}) := \|(\nabla \varphi(\hat{z}))^\top \tfrac{\hat{z} - \psi(b)e^{-(T-t)/2}}{e^{-(T-t)} - 1} + \tfrac{1}{2}\mathrm{tr}(\nabla^2 \varphi(\hat{z})) - f(\hat{x}, \hat{t})\|^2$
2: Define $\forall \hat{\theta} \in \mathbb{R}^{a_2} \ \hat{z} \in \mathbb{R}^d, b, x \in \mathcal{M}, \hat{t} \in [0, T]$, the objective $G(\hat{\phi}; b, \hat{z}, \hat{x}, \hat{t}) := \|J_\varphi(\hat{z})^\top J_\varphi(\hat{z}) - (g_{\hat{\phi}}(\hat{x}, \hat{t}))^2\|_F^2$
3: Set $\theta \leftarrow \theta_0, \phi \leftarrow \phi_0$
4: **for** $i = 1, \ldots, r$ **do**
5:     Sample a random batch $S \subseteq [m]$ of size $\mathfrak{b}$,
6:     Sample $t \sim \mathrm{Unif}([0, T])$
7:     **for** $j \in S$ **do**
8:         Sample $\xi \sim N(0, I_d)$
9:         Set $z_j \leftarrow \psi(x_0^j)e^{-\frac{1}{2}(T-t)} + \sqrt{1 - e^{-(T-t)}}\xi$
10:        Set $x_j \leftarrow \varphi(z_j)$
11:     **end for**
12:     Compute $\Gamma \leftarrow \frac{1}{\mathfrak{b}} \sum_{j \in S} \nabla_\theta F(\theta; x_0^j, z_j, x_j, t)$
13:     $\theta \leftarrow \theta - \eta \Gamma$
14:     Compute $\Upsilon \leftarrow \frac{1}{\mathfrak{b}} \sum_{j \in S} \nabla_\phi G(\phi; x_0^j, z_j, x_j, t)$
15:     $\phi \leftarrow \phi - \eta \Upsilon$
16: **end for**
17: **output:** Parameters $\theta, \phi$ for the models $f_\theta$ and $g_\phi$

---

**Algorithm 2** Sampling algorithm

---

**Input:** A way to compute the value of the exponential map $\exp(x, v)$ on some manifold $\mathcal{M}$, for any $x \in \mathcal{M}, v \in \mathcal{T}_x \mathcal{M}$
**Input:** A way to compute the "projection" map $\varphi : \mathbb{R}^d \to \mathcal{M}$
**Input:** Models $f_{\hat{\theta}} : \mathcal{M} \times [0, T] \to \mathcal{T}\mathcal{M}$ and $g_{\hat{\phi}} : \mathcal{M} \times [0, T] \to \mathcal{T}\mathcal{M} \times \mathcal{T}\mathcal{M}$. $\hat{\theta} \in \mathbb{R}^{a_1}, \hat{\phi} \in \mathbb{R}^{a_2}$ denote trainable parameters
**Input:** Parameters $\theta, \phi$ (from output of Algorithm 1),
**Input:** $T > 0, N \in \mathbb{N}, \Delta > 0$ such that $T/\Delta \in N\mathbb{Z}$.
1: Sample $z_0 \sim N(0, I_d)$, and Set $\hat{y}_0 \leftarrow \varphi(z_0)$
2: **for** $i = 0, 1, \ldots, T/\Delta - 1$ **do**
3:     Sample $\xi \sim N(0, I_d)$.
4:     Set $\hat{y}_{i+1} \leftarrow \exp(\hat{y}_i, \ \hat{f}(\hat{y}_i, i\Delta)\Delta + \hat{g}(\hat{y}_i, i\Delta)\sqrt{\Delta}\xi_i)$
5: **end for**
6: **output** $\hat{y}_{T/\Delta}$

---

## 3.2. Highlights of proof of sampling guarantees

Girsanov transformations, used in prior works to bound accuracy, do not apply to our diffusion due to its spatially varying covariance. We adopt an optimal transport approach, selecting an optimal coupling between the "ideal" diffusion $Y_t$, governed by the SDE $dY_t = f^\star(Y_t, t)dt + g^\star(Y_t, t)dB_t$, and the diffusion $\hat{Y}_t$, where $f^\star, g^\star$ are replaced by our trained model $\hat{f}, \hat{g}$, within an error $\varepsilon$. We first construct a simple coupling between $Y_t$ and $\hat{Y}_t$ by setting the underlying $B_t$ in their SDEs equal. Applying comparison theorems for manifolds of non-negative curvature to the coupled SDEs, we prove a generalization of Gronwall's inequality to SDEs on manifolds (Lemma B.3).

$$W_2(\hat{Y}_t, Y_t) \leq (\rho^2(\hat{Y}_0, Y_0) + \varepsilon)e^{ct}, \tag{9}$$

where $\rho$ is geodesic distance on $\mathcal{M}$ and $W_2$ the Wasserstein distance. (9) holds if $f^\star, g^\star$ are $c$-Lipschitz on *all* of $\mathcal{M}$.

**Showing "average-case" Lipschitzness.** $\varphi$ is not in general Lipschitz. E.g., on $\mathrm{SO}(n)$ and $\mathrm{U}(n)$, $\varphi(Z)$ has singularities at points where the eigengaps of $Z + Z^*$ vanish. By using tools from random matrix theory, we instead show $\varphi$ satisfies an "average-case" Lipschitzness, on a set $\Omega_t$ on which the diffusion $Z_t \in \mathbb{R}^d$ remains w.h.p. (Lemma B.4). Next, we show that for $f^\star, g^\star$ to be $c$-Lipschitz *everywhere* on $\mathcal{M}$, it is sufficient for $\varphi$ to only satisfy average-case Lipschitzness. To do this, we express $f^\star$ (and $g^\star$) as an integral over the eigenvalues $\Lambda$ of $Z_t + Z_t^* = U\Lambda U^*$,

$$f^\star(U, t) \propto \int [\nabla\varphi(Z)^\top \nabla \log q_{T-t|0}(Z) + \cdots]\mathbb{1}_{\Omega_t}(Z)d\Lambda.$$

Due to the manifold's symmetries, we observe $\Omega_t$ (and the entire integrand) depend only on $\Lambda$, not the eigenvectors $U \in \mathrm{U}(n)$. This allows us to show the integral "smooths out" the singularities of $\varphi$, and that $f^\star(U, t)$ (and $g^\star$) are $\mathrm{poly}(d)$-Lipschitz at *every* $U \in \mathrm{U}(n)$ on the manifold (Lemma B.6).

**Improved coupling to obtain** $\mathrm{poly}(d)$ **bounds.** While we have shown our model's SDE is $c = \mathrm{poly(d)}$-Lipschitz on $\mathcal{M}$, after times $\tau > \frac{1}{c} = \frac{1}{\mathrm{poly(d)}}$, our Wasserstein bound (9) grows exponentially with $d$. To overcome this, we define a new coupling between $Y_t, \hat{Y}_t$ which we "reset" after time intervals of length $\tau = 1/c$ by converting our Wasserstein bound into a total variation (TV) bound after each interval. Key to converting the bound is to show that w.h.p. the projection $\varphi$ has $\mathrm{poly}(d)$-Lipschitz Jacobian everywhere in a ball of radius $1/\mathrm{poly}(d)$ around our diffusion. By alternating between Wasserstein and TV bounds, we get error bounds which grow proportional to $T/\tau = \mathrm{poly(d)}$ (Lemma B.7).

**Handling instability on** $\mathrm{SO}(n)$**.** These proof ideas extend easily to the torus and sphere. For $\mathrm{SO}(n)$, an additional challenge arises: w.h.p., gaps between neighboring eigenvalues become exponentially small in $d$ over short time intervals due to weaker "electrical repulsion" between eigenvalues of

real-valued random matrices. During these intervals, the diffusion moves at $\exp(d)$ velocity. Despite this, we show that a step size of $1/\mathrm{poly}(d, \frac{1}{\delta})$ suffices to simulate a random solution to the SDE with a distribution $\delta$-close to the correct one. During these intervals, interactions between eigenvectors nearly separate into slow-moving eigenvectors and pairs of fast-moving eigenvectors, with a simple transition kernel from the invariant measure on $\mathrm{SO}(2)$ (see Appendix B.7).

# 4. Empirical results

We provide proof-of-concept simulations to compare the quality and efficiency of our algorithms to key prior works.

**Datasets.** We evaluate the quality of samples generated by our model on synthetic datasets from the torus $\mathbb{T}_d$, the special orthogonal group $\mathrm{SO}(n)$, and the unitary group $\mathrm{U}(n)$. The torus provides a simple, zero-curvature geometry for initial validation, while $\mathrm{SO}(n)$ and $\mathrm{U}(n)$ test the model on more complex geometries. Datasets include unimodal wrapped Gaussians on $\mathbb{T}_d$, multimodal Gaussian mixtures on $\mathrm{SO}(n)$, and time-evolution operators of a quantum oscillator with random potentials on $\mathrm{U}(n)$, for varying dimensions $d = n^2$. We also separately analyze per-iteration runtime to study scaling across dimensions, requiring only a single training step and limited computational resources.

For the torus $\mathbb{T}_d$, following several works (De Bortoli et al., 2022; Zhu et al., 2025), we train diffusion models on data sampled from wrapped Gaussians on tori of different dimensions $d \in \{2, 10, 50, 100, 1000\}$, with mean 0 and covariance $0.2I_d$ (See Appendix C.1 for definition of wrapped Gaussian). For $\mathrm{U}(n)$, following (Zhu et al., 2025), we use a dataset on $\mathrm{U}(n)$ of unitary matrices representing time-evolution operators $e^{itH}$ of a quantum oscillator. In our simulations, we consider a wider range of $n \in \{3, 5, 9, 12, 15\}$ (corresponding to manifold dimensions $d = \frac{n(n-1)}{2} \in \{3, 10, 36, 66, 105\}$) when evaluating sample quality, and $n \in \{3, 5, 10, 30, 50\}$ ($d \in \{3, 45, 435, 1225\}$) when evaluating runtime. Here $t$ is time, $H = \frac{\hbar}{2m}\Delta - V$ is a Hamiltonian, and $\Delta$ is the Laplacian. $V$ is a random potential $V(x) = \frac{\omega^2}{2}\|x - x_0\|^2$ with angular momentum $\omega$ sampled uniformly on $[2, 3]$ and $x_0 \sim \mathcal{N}(0, 1)$. As $\Delta, V$ are infinite-dimensional operators, matrices in $\mathrm{U}(n)$ are obtained by retaining the (discretized) top-$n$ eigenvectors of $\Delta, V$. For $\mathrm{SO}(n)$, following (Zhu et al., 2025), we use datasets sampled from a mixture of a small number $k$ of wrapped Gaussians on $\mathrm{SO}(n)$. We set $k = 2$ and use the same values of $n$ as on $\mathrm{U}(n)$.

**Algorithm.** We use Algorithm 1 to train our model, and Algorithm 2 to generate samples. Each algorithm takes as input a projection map $\varphi$ and restricted inverse $\psi$, with choices for $\mathbb{T}_d$, $\mathrm{SO}(n)$, and $\mathrm{U}(n)$ detailed in Section 2. For $\mathrm{SO}(n)$ and $\mathrm{U}(n)$, we use the projection map $\hat{\varphi}$ defined in Section 2,

as it suffices to generate high-quality samples in our simulations. In both algorithms, the drift function $\hat{f}(\cdot, \cdot)$ for the reverse diffusion is parameterized by a neural network. Details on the network architecture, training iterations, batch size, and hardware are provided in Appendix C.2.[1]

The function $\hat{g}(\cdot, \cdot)$, which models the covariance in our model's reverse diffusion SDE, vanishes on $\mathbb{T}_d$. On $\mathrm{SO}(n)$, $\mathrm{U}(n)$, $\hat{g}$ is a $n^2 \times n^2$ matrix. This matrix has a special structure (see Section 3.1), allowing it to be parameterized by $d = n^2$ numbers. We may thus parametrize $\hat{g}(x, t)$ by a neural net with inputs $x$ of dimension $d$, $t$ of dimension 1, and output dimension $d$. Network architecture is the same as for $\hat{f}(\cdot, \cdot)$.

**Benchmarks.** We compare samples generated by our model to those from RSGM (De Bortoli et al., 2022), TDM (Zhu et al., 2025), and a "vanilla" Euclidean diffusion model. RSGM and TDM are included as they have demonstrated improved sample quality and runtime over previous manifold generative models, such as Moser Flow (Rozen et al., 2021) on $\mathbb{T}_d$ and $\mathrm{SO}(3)$. TDM also outperforms RSGM and Flow Matching (Chen et al., 2024) on higher-dimensional torus datasets and shows better sample quality on $\mathrm{SO}(n)$ and $\mathrm{U}(n)$, where RSGM and RDM (Huang et al., 2022) lack experiments for $n > 3$. All models are trained with the same iterations, batch size, and architecture (see Appendix C.2).

For the Euclidean diffusion model, samples are constrained to the manifold $\mathcal{M}$ by preprocessing via $\tilde{\varphi}$ and postprocessing via $\tilde{\psi}$. For $\mathcal{M} = \mathbb{T}_d$, $\tilde{\varphi}(x)[i] = x[i] \mod 2\pi$, and $\tilde{\psi}$ is its inverse on $[0, 2\pi)^d$. For $\mathrm{SO}(n)$ and $\mathrm{U}(n)$, $\tilde{\varphi}(X)$ computes the $U$ from the singular value decomposition $X = U\Sigma V^*$, and $\tilde{\psi}$ is the usual embedding.

RSGM and TDM are trained using their divergence-based ISM objective, which is prohibitively slow for large dimensions. To fully train these models and evaluate their sample quality, we follow their implementation and use a stochastic estimator for the ISM divergence. We do not compare to TDM on the torus, as their specialized heat kernel objective does not generalize beyond the torus or Euclidean space.

We do not compare to SCRD (Lou et al., 2024), as their implementation is limited to efficient heat kernel expansion for $\mathrm{SO}(n)$ and $\mathrm{U}(n)$ with $n = 3$. For $n > 3$, the cost of their expansion grows exponentially with $n$, making it infeasible for higher dimensions.

**Metrics.** For the torus, as in (De Bortoli et al., 2022; Zhu et al., 2025)), we evaluate the quality of generated samples by computing their log-likelihood. For $\mathrm{SO}(n)$ and $\mathrm{U}(n)$, we use the Classifier Two-Sample Test (C2ST) metric (Lopez-Paz & Oquab, 2017). The C2ST metric was previously used

---

in e.g. (Leach et al., 2022) to evaluate sample quality of diffusion models on $\mathrm{SO}(n)$ for $n = 3$. It measures the ability of a classifier neural network to differentiate between generated samples and samples in a test dataset. This metric allows one to evaluate sample quality in settings where computing the likelihood may be intractable, as may be the case for diffusion models on $\mathrm{SO}(n)$ and $\mathrm{U}(n)$ for larger $n$. We use the C2ST metric on $\mathrm{SO}(n)$ and $\mathrm{U}(n)$ instead of log-likelihood, as we found that computing log-likelihood for $n > 3$ poses additional computational challenges, while C2ST can be computed efficiently. See Appendix C.3 for definitions of log-likelihood and C2ST, and additional details.

**Results for generated sample quality.** For the torus, we compare our model, a Euclidean diffusion model, and RSGM on a dataset sampled from a wrapped Gaussian on the $d$-dimensional torus for different values of $d$. Our model has the lowest negative log-likelihood (NLL) for $d \geq 10$, and its NLL degrades with the dimension at a much slower rate than the Euclidean model and RSGM (Table 2; lower NLL indicates better-quality sample generation).

For $\mathrm{U}(n)$, we train our model, a Euclidean diffusion model, RSGM, and TDM on a dataset on $\mathrm{U}(n)$ comprised of discretized quantum oscillator time-evolution operators, for $n \in \{3, 5, 9, 12, 15\}$. For $n \geq 9$, our model achieves the lowest C2ST score; a lower C2ST score indicates higher-quality sample generation (Figure 1, top). Visually, we observe our model's generated samples more closely resemble the target distribution than benchmark models' for $n \geq 9$ (see Figure 1, bottom, for $n = 15$, Appendix C.4 for other $n$). Improvements on $\mathrm{SO}(n)$ were similar to those on $\mathrm{U}(n)$ (see Appendix C.4).

Table 2: Negative log-likelihood (NLL) when training on a wrapped Gaussian dataset on the torus of different dimensions $d$. Lower NLL indicates better-quality sample generation. Our model's NLL increases more slowly with $d$, and is lower for $d \geq 10$, than Euclidean and RSGM models.

| Method | $d = 2$ | $d = 10$ | $d = 50$ | $d = 100$ | $d = 1000$ |
|---|---|---|---|---|---|
| Euclidean | 0.61±.02 | 0.61±.02 | 0.66±.04 | 8.18±.42 | 8.7±.37 |
| RSGM | **0.42**±.03 | 0.49±.05 | 0.62±.06 | 1.45±.21 | 2.51±.17 |
| Ours | 0.49±.04 | **0.47**±.03 | **0.50**±.05 | **0.52**±.09 | **0.97**±.19 |

**Results for runtime.** We evaluate the per-iteration training runtime on $\mathrm{U}(n)$ on a wider range of $n = 3, 5, 10, 30, 50$ (corresponding to manifold dimensions of $d = \frac{n(n-1)}{2} \in \{3, 10, 45, 435, 1225\}$). Our model's per-iteration runtime remains within a factor of 3 of the Euclidean model's for all $n$. Per-iteration runtimes of TDM and RSGM, with ISM objective, increase more rapidly with dimension and are, respectively, 45 and 57 times greater than the Euclidean model's for $n = 50$ (Table 3). Similar runtime improvements were observed on $\mathrm{SO}(n)$ (Table 6 in Appendix C.2).

---

[1]Our code can be found at github.com/mangoubi/Efficient-Diffusion-Models-for-Symmetric-Manifolds

Table 3: Per-iteration training runtime in seconds on $\mathrm{U}(n)$. The fastest manifold-constrained diffusion model is in bold; the Euclidean model is in gray for comparison. Our model's runtime remains within a factor of 3 of the Euclidean model's for all $n$. Runtimes of TDM, RSGM increase more rapidly with dimension $d = \Theta(n^2)$ and are, respectively, 45 and 57 times greater than the Euclidean model's for $n = 50$ ($d = 1225$).

| Method | $n = 3$ | $n = 5$ | $n = 10$ | $n = 30$ | $n = 50$ |
|---|---|---|---|---|---|
| Euclidean | $0.19 \pm .01$ | $0.19 \pm .01$ | $0.20 \pm .01$ | $0.20 \pm .01$ | $0.21 \pm .01$ |
| RSGM | $1.03 \pm .02$ | $1.22 \pm .08$ | $1.51 \pm .03$ | $3.98 \pm .19$ | $11.55 \pm .31$ |
| TDM | $0.91 \pm .08$ | $1.07 \pm .06$ | $2.46 \pm .09$ | $3.77 \pm .17$ | $9.43 \pm .23$ |
| **Ours** | $\mathbf{0.36 \pm .00}$ | $\mathbf{0.36 \pm .00}$ | $\mathbf{0.36 \pm .00}$ | $\mathbf{0.46 \pm .01}$ | $\mathbf{0.60 \pm .01}$ |

**Summary.** We find that, as predicted by our theoretical training runtime bounds in Table 1, the per-iteration training runtime of our model is significantly faster, and grows more slowly with dimension, than previous manifold diffusion models on $\mathrm{U}(n)$ (similar improvements were observed for $\mathrm{SO}(n)$). Our algorithm's runtime remains within a small constant factor of the per-iteration runtime of the Euclidean diffusion model, at least for $n \leq 50$ (corresponding to a manifold dimension of $d \leq 1225$), nearly closing the gap with the per-iteration runtime of the Euclidean model.

Moreover, we find that (except in very low dimensions) our model is capable of improving on the quality of samples generated by previous diffusion models, when trained on different synthetic datasets on the torus, $\mathrm{SO}(n)$ and $\mathrm{U}(n)$. The magnitude of the improvement increases with dimension.

| Method | $n = 3$ | $n = 5$ | $n = 9$ | $n = 12$ | $n = 15$ |
|---|---|---|---|---|---|
| Euclidean | $.69 \pm .03$ | $\mathbf{.75 \pm .04}$ | $.87 \pm .04$ | $.97 \pm .02$ | $1.00 \pm .01$ |
| RSGM | $.79 \pm .04$ | $.92 \pm .04$ | $.97 \pm .03$ | $1.00 \pm .02$ | $1.00 \pm .01$ |
| TDM | $.73 \pm .04$ | $.91 \pm .02$ | $.99 \pm .02$ | $1.00 \pm .01$ | $1.00 \pm .01$ |
| Ours | $.75 \pm .05$ | $.80 \pm .04$ | $\mathbf{.84 \pm .04}$ | $\mathbf{.88 \pm .05}$ | $\mathbf{.90 \pm .04}$ |

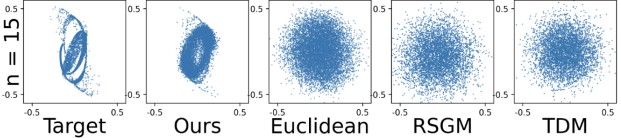

Figure 1: C2ST scores when training on datasets of quantum evolution operators on $\mathrm{U}(n)$ (top). Lower scores indicate better-quality generated samples (range is $[0.5, 1]$). For $n \geq 9$, our model has the best C2ST scores. Generated samples are plotted for $n = 15$ (bottom); axes are two matrix entries.

## 5. Conclusion and future work

We introduce a new diffusion model with a spatially varying covariance structure, enabling efficient training on symmetric manifolds with non-zero curvature. By leveraging manifold symmetries, we ensure the reverse diffusion satisfies an "average-case" Lipschitz condition, which underpins both the accuracy and efficiency of our sampling algorithm.

Our approach improves training runtime and sample quality on symmetric manifolds, significantly narrowing the gap between manifold-based diffusion models and their Euclidean counterparts. Furthermore, the model naturally extends to conditional generation: given a conditioning variable $y$, one can feed $y$ as an additional input to the learned drift and covariance functions, mirroring conditional diffusion models in Euclidean settings.

Several open directions remain. One is to extend our framework to more general manifolds—such as the manifold of positive semi-definite matrices, or other domains admitting a projection oracle satisfying suitable average-case smoothness properties (see Appendix F). Another direction is to handle distributions supported on a union of manifolds with varying dimensions, such as the GEOM-DRUGS dataset (Jing et al., 2022), which lies on a union of tori.

Finally, while our method yields polynomial-in-$d$ bounds on sampling accuracy—improving upon prior works that lacked such guarantees—tightening this dependence remains an important challenge for future research.

## Impact statement

Diffusion generative models that generate data constrained to symmetric manifolds can significantly enhance the societal impact of machine learning applications. These include areas such as drug discovery and robotics, where molecular or robotic configurations can be represented as points on symmetric manifolds, as well as quantum mechanics applications like quantum chemistry, materials science, and microelectronics. Moreover, our focus on improving the efficiency of generative models has clear environmental benefits. Reducing the computational costs associated with training and sampling minimizes energy consumption, helping to lower the carbon footprint of machine learning applications. This is particularly relevant as the demand for large-scale generative models continues to grow. However, generative modeling also presents potential risks. For example, such models can be used to generate fake news videos or to train robots and drones for harmful purposes when misused by malicious actors. While our work is primarily theoretical, limiting its direct negative impact, it is important to remain mindful of both the positive and negative implications of applying generative modeling technologies in sensitive domains. Balancing these considerations is essential to responsibly harness the potential of generative models in advancing critical scientific and societal challenges while mitigating risks and environmental impacts.

## Acknowledgments

OM was supported in part by a Google Research Scholar award. NV was supported in part by NSF CCF-2112665.

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

## A. Proof outline of Theorem 2.2

In the following, for any random variable $X$, we denote its probability distribution by $\mathcal{L}_X$. As already mentioned, previous works use Girsanov's theorem to bound the accuracy of diffusion methods. However, Girsanov transformations do not exist for our diffusion as it has a non-constant covariance term which varies with the position $x$. Thus, we depart from previous works and instead use an optimal transport approach based on a carefully chosen optimal coupling between the "ideal diffusion" $Y_t$ and the algorithm's process $\hat{y}_t$ Specifically, denoting by $\mu_t$ the distribution of $Y_t$ and by $\nu_t$ the distribution of $\hat{Y}_t$, the goal is to bound the Wasserstein optimal transport distance $W_2(\mu_t, \nu_t) := \inf_{\kappa \in \mathcal{K}(\mu_t, \nu_t)} \mathbb{E}_{(Y_t, \hat{Y}_t)}[\rho^2(\hat{Y}_2, Y_2)]$ where $\mathcal{K}(\mu, \nu)$ is the collection of all couplings of the distributions $\mu$ and $\nu$, and $\rho$ is the geodesic distance on $\mathcal{M}$. Towards this end, we would like to find a coupling $\kappa$ which (approximately) minimizes $\mathbb{E}_{(Y_t \sim \mu_t, \hat{Y}_t \sim \nu_t)}[\rho^2(\hat{Y}_t, Y_2)]$ at any given time $t$.

As a first attempt, we consider the simple coupling where we couple the "ideal" reverse diffusion $Y_t$,

$$\mathrm{d}Y_t = f^\star(Y_t, t)\mathrm{d}t + g^\star(Y_t, t)\mathrm{d}B_t, \tag{10}$$

and the reverse diffusion $\hat{Y}_t$ given by our trained model $\hat{f}, \hat{g}$,

$$\mathrm{d}\hat{Y}_t = \hat{f}(\hat{Y}_t, t)\mathrm{d}t + \hat{g}(Y_t, t)\mathrm{d}B_t. \tag{11}$$

To couple these two diffusions, we set their Brownian motion terms $\mathrm{d}B_t$ to be equal to each other at every time $t$. In a similar manner, we can also couple $\hat{Y}_t$ and the discrete-time algorithm $\hat{y}_i$ by setting the Gaussian term $\xi_i$ in the stochastic finite difference equation (8) to be equal to $\xi_i = \frac{1}{\sqrt{\Delta}} \int_{\Delta i}^{\Delta(i+1)} \mathrm{d}B_t \mathrm{d}t$ for every $i$.

**Step 1: Bounding the Wasserstein distance for everywhere-Lipschitz SDEs.** To bound the Wasserstein distance $W_2(Y_t, \hat{y}_t) \leq W_2(Y_t, \hat{Y}_t) + W_2(\hat{Y}_t, \hat{y}_t)$, we first prove a generalization of Gronwall's inequality to Stochastic differential equations on manifolds (Lemma B.3). Gronwall's inequality (Gronwall, 1919) says that if $R : [0, T] \to \mathbb{R}$ satisfies the differential inequality $\frac{\mathrm{d}}{\mathrm{d}t}R(t) \leq \beta(t)R(t)$ for all $t > 0$, where the coefficient $\beta(t) : [0, T] \to \mathbb{R}$ may also be a function of $t$, then the solution to this differential inequality satisfies $R(t) \leq R(0)e^{\int_0^t \beta(s)\mathrm{d}s}$.

Towards this end, we first couple $Y_t$ and $\hat{Y}_t$ by setting their Brownian motion terms $\mathrm{d}B_t$ equal to each other and then derive an SDE for the squared geodesic distance $\rho^2(\hat{Y}_t, Y_t)$ using Itô's lemma. Taking the expectation of this SDE gives an ODE for $\mathbb{E}[\rho^2(\hat{X}_t, X_t)]$,

$$\mathrm{d}\mathbb{E}[\rho^2(\hat{X}_t, X_t)]$$
$$= \mathbb{E}\left[\nabla\rho^2(\hat{X}_t, X_t)^\top \begin{pmatrix} f^\star(X_t, t) \\ \hat{f}(\hat{X}_t, t) \end{pmatrix} + \frac{1}{2}\mathrm{Tr}\begin{pmatrix} g^\star(X_t, t) & 0 \\ \hat{g}(X_t, t) & 0 \end{pmatrix}^\top \nabla^2\rho^2(\hat{X}_t, X_t)\begin{pmatrix} g^\star(X_t, t) & 0 \\ \hat{g}(X_t, t) & 0 \end{pmatrix}\right]\mathrm{d}t. \tag{12}$$

To bound each term on the r.h.s., we first observe that, roughly speaking, due to the non-negative curvature of the manifold, by the Rauch comparison theorem (Rauch, 1951), each derivative on the r.h.s. is no larger than in the Euclidean case $\mathcal{M} = \mathbb{R}^d$ where $\rho^2(\hat{X}_t, X_t) = \|\hat{X}_t - X_t\|_2^2$. Hence, we have that

$$\left|\nabla\rho^2(\hat{X}_t, X_t)^\top \begin{pmatrix} f^\star(X_t, t) \\ \hat{f}(\hat{X}_t, t) \end{pmatrix}\right| \leq 2\|\hat{X}_t - X_t\| \times \|f^\star(X_t, t) - \hat{f}(\hat{X}_t, t)\| \leq 2\|\hat{X}_t - X_t\|(c\|\hat{X}_t - X_t\| + \varepsilon),$$

as long as we can show that $f^\star$ is $c$-Lipschitz for some $c > 0$ (see Step 2 below). Bounding the covariance term in a similar manner, and applying Gronwall's lemma to the differential inequality, we get that

$$W_2(\hat{Y}_t, Y_t) \leq \mathbb{E}[\rho^2(\hat{Y}_t, Y_t)] \leq (\rho^2(\hat{Y}_0, Y_0) + \varepsilon)e^{ct}. \tag{13}$$

**Step 2: Showing that our diffusion satisfies an "average-case" Lipschitz condition.** To apply (13), we must first show that the drift and diffusion terms $f^\star$ and $g^\star$ are Lipschitz on $\mathcal{M}$. Towards this end, we would ideally like to apply bounds on the derivatives of the projection map $\varphi : \mathbb{R}^d \to \mathcal{M}$ which defines our diffusion $Y_t$. Unfortunately, in general, $\varphi$ may not be differentiable at every point. This is the case for the sphere, where the map $\varphi(z) = \frac{z}{\|z\|}$ has a singularity at $z = 0$. This issue also arises in the case of the unitary group and orthogonal group, since the derivative of the spectral decomposition $\varphi(z) = U^*\Lambda U$ has singularities at any matrix $z$ which has an eigenvalue gap $\lambda_i - \lambda_{i+1} = 0$.

To tackle this challenge, we show that, for the aforementioned symmetric manifolds, the forward diffusion $Z_t$ in $\mathbb{R}^d$ remains in some set $\Omega_t \subseteq \mathbb{R}^d$ with high probability $1 - \alpha$, on which the map $\varphi(Z_t)$ has derivatives bounded by $\mathrm{poly}(d)$ (Assumption 2.1 and Lemma B.4). We then show how to "remove" the rare outcomes of our diffusion that do not fall inside $\Omega_t$. As our forward diffusion $X_t$ (and thus the reverse diffusion $Y_t = X_{T-t}$) remains at every $t$ inside $\Omega_t$ with probability $\geq 1 - \alpha$, removing these "bad" outcomes only adds a cost of $\alpha$ to the total variation error.

*Showing that $\varphi$ has $\mathrm{poly}(d)$ derivatives w.h.p. (showing that Assumption 2.1 holds).* We first consider the sphere, which is the simplest case (aside from the trivial case of the torus, where the derivatives of $\varphi$ are all $O(1)$ at every point). In the case when data is on the sphere, which we embed as a unit sphere in $\mathbb{R}^d$, one can easily observe that e.g. $\|\nabla\varphi(z)\| \leq O(1)$ for any $z$ outside a ball of radius $r \geq \Omega(1)$ centered at the origin. As the volume of a ball of radius $r = \alpha$ is $\frac{1}{r^d}$, one can use standard Gaussian concentration inequalities to show that the Brownian motion $X_t$ will remain outside this ball for time $T$ with probability roughly $1 - O(\frac{1}{r^d T})$.

We next show that the Lipschitz property holds for the unitary group $\mathrm{U}(n)$. We first recall results from random matrix theory, which allow us to bound the eigenvalue gaps of a matrix with Gaussian entries. Specifically, these results say that roughly speaking, if $X_0$ is any matrix and $X_t = X_0 + B(t)$, where $B(t)$ is a symmetric matrix with i.i.d. $N(0, t)$ entries undergoing Brownian motion, one has that the eigenvalues $\gamma_1(t) \geq \cdots \geq \gamma_n(t)$ of $X_t$ satisfy for all $s \geq 0$ (see e.g. (Anderson et al., 2010; Mangoubi & Vishnoi, 2023; 2025)),

$$\mathbb{P}\left(\bigcap_{t \in [t_0, T]} \left\{\gamma_{i+1}(t) - \gamma_i(t) \leq s \frac{1}{\mathrm{poly}(n)\sqrt{t}}\right\}\right) \leq O\left(s^{\frac{1}{2}}\right). \tag{14}$$

Thus, if we define $\Omega_t$ to be the set of outcomes of such that $\gamma_{i+1}(t) - \gamma_i(t) \leq \alpha^2 \frac{1}{\mathrm{poly}(n)\sqrt{t}}$, we have that $\mathbb{P}(X_t \in \Omega_t \ \forall t \in [t_0, T]) \geq 1 - \alpha$.

Our high-probability bound on $\Omega_t$ allows us to show that $\varphi$ satisfies a Lipschitz property at "most" points $\Omega_t$. However, if we wish to apply (13), we need to show that the drift term $f^\star$ and the covariance term $g^\star$ in our diffusion satisfy a Lipschitz property at *every* point in $\mathbb{R}^d$. Towards this end, we first make a small modification to the objective function which allows us to exclude outcomes $\{X_t\}_{t \in [0, T]}$ of the forward diffusion such that $X_t \notin \Omega_t$ for some $t \in [0, T]$. Specifically, we multiply the objective function (4) by the indicator function $\mathbb{1}_{\Omega_t}(z)$. As determining whether a point $z \in \Omega_t$ requires only checking the eigenvalue gaps (when $\mathcal{M}$ is the unitary or orthogonal group), computing $\mathbb{1}_{\Omega_t}(z)$ can be done efficiently using the singular value decomposition.

*Bounding the Lipschitz constant of $f^\star$ and $g^\star$.* Recall that (when, e.g., $\mathcal{M}$ is one of the aforementioned symmetric manifolds) we may decompose any $z \in \mathbb{R}^d$ as $z \equiv z(U, \Lambda)$ where $U \in \mathcal{M}$. Note that $\mathbb{1}_{\Omega_t}(z)$ is *not* a continuous function of $z$. However, we will show that, as $\mathbb{1}_{\Omega_t}(z(U, \Lambda))$ depends only on $\Lambda$, multiplying our objective function by $\mathbb{1}_{\Omega_t}$ does not make $f^\star$ and $g^\star$ discontinuous (and thus does not prevent them from being Lipschitz). This is because $f^\star$ and $g^\star$ are given by conditional expectations conditioned on $U$, and can thus be decomposed as integrals over $\Lambda$. Towards this end we express $f^\star$ as an integral over the parameter $\Lambda$,

$$f^\star(U, t) = c_U \int_{\Lambda \in \mathcal{A}} \left[\nabla\varphi(z(U, \Lambda))^\top \nabla \log q_{T-t|0}(z(U, \Lambda)) + \frac{1}{2}\mathrm{tr}\nabla^2\varphi(z(U, \Lambda))\right] q_{T-t}(z(U, \Lambda))\mathbb{1}_{\Omega_t}(\Lambda)\mathrm{d}\Lambda,$$

where $c_U$ is a normalizing constant. Differentiating with respect to $U$,

$$\frac{\mathrm{d}}{\mathrm{d}U}f^\star(U, t) = \mathbb{E}_{z(U, \Lambda) \sim q_{T-t}}\left[\frac{\mathrm{d}}{\mathrm{d}U}(\nabla\varphi(z(U, \Lambda))^\top \nabla_U \log q_{T-t|0}(z(U, \Lambda))\right.$$
$$\left. + \frac{1}{2}\mathrm{tr}(\nabla^2\varphi(z(U, \Lambda))))\mathbb{1}_{\Omega_t}(\Lambda) \ \middle| \ V = U\right] + \cdots, \tag{15}$$

where "$\cdots$" includes three other similar terms. To bound the terms on the r.h.s. of (15), we apply Assumption 2.1 which says that the operator norms of $\nabla\varphi$, $\nabla^2\varphi$, $\frac{\mathrm{d}}{\mathrm{d}U}\nabla\varphi$ and $\frac{\mathrm{d}}{\mathrm{d}U}\nabla^2\varphi$ are all bounded above by $\mathrm{poly}(d)$ whenever $z \in \Omega_t$. To bound the term $\nabla_U \log q_{T-t|0}(z(U, \Lambda))$ we note that $\nabla \log q_{T-t|0}(z(U, \Lambda))$ is the drift term of the reverse diffusion in Euclidean space. This term was previously shown to be $dC^2$-Lipschitz for all $t \geq \Omega(\frac{1}{d})$ when the support of the data distribution in $\mathbb{R}^d$ lies in a ball of radius $C$ (see, e.g., Proposition 20 of (Chen et al., 2023b)). Thus, plugging in the above bounds into (15) we

have that $\|\frac{\mathrm{d}}{\mathrm{d}U} f^\star(U, t)\|_{2\to 2} \le \mathrm{poly}(d)$. A similar calculation shows that $\|\frac{\mathrm{d}}{\mathrm{d}U} g^\star(U, t)\|_{2\to 2} \le \mathrm{poly}(d)$. This immediately implies that $f^\star(U, t)$ and $g^\star(U, t)$ are $\mathrm{poly}(d)$-Lipschitz at *every* $U \in \mathcal{M}$.

**Step 3: Improving the coupling to obtain polynomial-time bounds.** Now that we have shown that $f^\star$ and $g^\star$ are $\mathrm{poly}(d)$-Lipschitz, we can apply (13) to bound the Wasserstein distance: $W_2(\hat{Y}_{t+\tau}, Y_{t+\tau}) \le (\rho^2(\hat{Y}_t, Y_t) + \varepsilon)e^{c\tau}$ $\quad \forall \tau \ge 0$, where $c \le \mathrm{poly}(d)$.

Moreover, with slight abuse of notation, we may define $\hat{y}_{t+\tau}$ to be a continuous-time interpolation of the discrete process $\hat{y}$. Applying (13) to this process, we get that, roughly, $W_2(\hat{Y}_{t+\tau}, \hat{y}_{t+\tau}) \le (\rho^2(\hat{y}_t, Y_t) + \varepsilon + \Delta)e^{c\tau}$ for $\tau \ge 0$. Thus, we get a bound on the Wasserstein error,

$$W_2(Y_{t+\tau}, \hat{y}_{t+\tau}) \le W_2(\hat{Y}_{t+\tau}, Y_{t+\tau}) + W_2(\hat{Y}_{t+\tau}, \hat{y}_{t+\tau}) \le (\rho^2(\hat{y}_t, Y_t) + \varepsilon + \Delta)e^{c\tau}, \qquad \tau \ge 0. \tag{16}$$

Unfortunately, after times $\tau > \frac{1}{c} = \frac{1}{\mathrm{poly}(d)}$, this bound grows exponentially with the dimension $d$. To overcome this challenge, we define a new coupling between $Y_t$ and $\hat{Y}_t$ which we "reset" after time intervals of length $\tau = \frac{1}{c}$ by converting our Wasserstein bound into a total variation bound after each time interval. Towards this end, we use the fact that if at any time $t$ the total variation distance satisfies $\|\mathcal{L}_{Y_t} - \mathcal{L}_{\hat{y}_t}\|_{\mathrm{TV}} \le \alpha$, then there exists a coupling such that $Y_t = \hat{Y}_t$ with probability at least $1 - \alpha$. In other words, w.p. $\ge 1 - \alpha$, we have $\rho(\hat{y}_{t+\tau}, Y_{t+\tau}) = 0$, and we can apply inequality (16) over the next time interval of $\tau$ without incurring an exponential growth in time. Repeating this process $\frac{T}{\tau}$ times, we get that $\|\mathcal{L}_{Y_T} - \mathcal{L}_{\hat{y}_T}\| \le \alpha \times \frac{T}{\tau}$, where the TV error grows only *linearly* with $T$.

*Converting Wasserstein bounds on the manifold to TV bounds.* To complete the proof, we still need to show how to convert the Wasserstein bound into a TV bound (Lemma B.7). Towards this end, we begin by showing that the transition kernel $\tilde{p}_{t+\tau+\hat{\Delta}|t+\tau}(\cdot | H_{t+\tau})$ of the reverse diffusion $H_t$ in $\mathbb{R}^d$ is close to a Gaussian in KL distance:

$$D_{\mathrm{KL}}(N(H_{t+\tau} + \hat{\Delta}\nabla\tilde{p}_{T-t-\tau}(H_{t+\tau}), \hat{\Delta}I_d) \,\|\, \tilde{p}_{t+\tau+\hat{\Delta}|t+\tau}(\cdot | H_{t+\tau})) \le \frac{\alpha\tau}{T}.$$

One can do this via Girsanov's theorem, since, unlike the diffusion $Y_t$ on the manifold, the reverse diffusion in Euclidean space $H_t$ *does* have a constant diffusion term (see e.g. Theorem 9 of (Chen et al., 2023b)).

Next, we use the fact that with probability at least $1 - \alpha\frac{\tau}{T}$ the map $\varphi$ in a ball of radius $\frac{1}{\mathrm{poly}(d)}$ about the point $H_{t+\tau}$ has $c$-Lipschitz Jacobian where $c = \mathrm{poly}(d)$, and that the inverse of the exponential map $\exp(\cdot)$ has $O(1)$-Lipschitz Jacobian, to show that the transition kernel $p_t$ of $Y_t = \varphi(H_t)$ satisfies

$$D_{\mathrm{KL}}(\nu_1 \,\|\, p_{t+\tau+\hat{\Delta}|t+\tau}(\cdot | Y_{t+\tau})) \le (1 + \hat{\Delta}c)^d \frac{\alpha\tau}{T} \le 2\frac{\alpha\tau}{T}$$

if we choose $\hat{\Delta} \le O(\frac{1}{cd})$, where $\nu_1 := \exp_{Y_{t+\tau}}(N(Y_{t+\tau} + \hat{\Delta}f^\star(Y_{t+\tau}, t+\tau), \; \hat{\Delta}g^{\star 2}(Y_{t+\tau}, t+\tau)I_d))$.

Next, we plug in our Wasserstein bound $W(Y_{t+\tau}, \hat{y}_{t+\tau}) \le O(\varepsilon)$ into the formula for the KL divergence between two Gaussians to bound $\|\mathcal{L}_{Y_{t+\tau+\hat{\Delta}}} - \mathcal{L}_{\hat{y}_{t+\tau+\hat{\Delta}}}\|_{\mathrm{TV}}$. Specifically, noting that $\mathcal{L}_{\hat{y}_{t+\tau+\hat{\Delta}}|\hat{y}_t} = \exp_{\hat{y}_{t+\tau}}(N(\hat{y}_{t+\tau} + \hat{\Delta}f(\hat{y}_{t+\tau}, t+\tau), \hat{\Delta}g^2(\hat{y}_{t+\tau}, t+\tau)I_d))$, we have that

$$\begin{aligned} D_{\mathrm{KL}}(\nu_1, \mathcal{L}_{\hat{y}_{t+\tau+\hat{\Delta}}|\hat{y}_{t+\tau}}) &= \mathrm{Tr}\left(\left(g^{\star 2}(Y_{t+\tau}, t+\tau)\right)^{-1} g^2(\hat{y}_{t+\tau}, t+\tau)\right) \\ &\quad -d + \log\frac{\det g^{\star 2}(Y_{t+\tau}, t+\tau)}{\det g^2(\hat{y}_{t+\tau}, t+\tau)} + w^\top\left(\hat{\Delta}g^{\star 2}(Y_{t+\tau}, t)\right)^{-1}w. \end{aligned}$$

where $w := Y_{t+\tau} - \hat{y}_{t+\tau} + \hat{\Delta}(f^\star(Y_{t+\tau}, t+\tau) - f(\hat{y}_{t+\tau}, t+\tau))$. Since with probability $\ge 1 - \alpha\frac{\tau}{T}$ we have $g^\star(Y_{t+\tau}) \succeq \mathrm{poly}(d)$, plugging in the error bounds $\|f^\star(Y_{t+\tau}, t) - f(Y_{t+\tau}, t)\| \le \varepsilon$ and $\|g^\star(Y_{t+\tau}, t) - g(Y_{t+\tau}, t)\|_F \le \varepsilon$ and the $c$-Lipschitz bounds on $f^\star$ and $g^\star$, where $c = \mathrm{poly}(d)$, (Assumption 2.1), we get that $D_{\mathrm{KL}}(\nu_1, \mathcal{L}_{\hat{y}_{t+\tau+\hat{\Delta}}}) \le O(\varepsilon^2 c^2)$. Thus, by Pinsker's inequality, we have

$$\begin{aligned} \|\mathcal{L}_{Y_{t+\tau+\hat{\Delta}}} - \mathcal{L}_{\hat{y}_{t+\tau+\hat{\Delta}}}\|_{\mathrm{TV}} - \|\mathcal{L}_{Y_t} - \mathcal{L}_{\hat{y}_t}\|_{\mathrm{TV}} \quad &\le \quad \sqrt{D_{\mathrm{KL}}(\nu_1 \,\|\, p_{t+\tau+\hat{\Delta}|t+\tau}(\cdot | Y_{t+\tau}))} \\ &\quad + \sqrt{D_{\mathrm{KL}}(\nu_1 \| \mathcal{L}_{\hat{y}_{t+\tau+\hat{\Delta}}|\hat{y}_t})} \le O(\varepsilon c). \end{aligned} \tag{17}$$

**Step 4: Bounding the accuracy.** Recall that $q_t$ is the distribution of the forward diffusion $Z_t$ in Euclidean space after time $t$, which is an Ornstein-Uhlenbeck process. Standard mixing bounds for the Ornstein-Uhlenbeck process imply that, $\|q_t - N(0, I_d)\|_{TV} \leq O(Ce^{-t})$ for all $t > 0$ (see e.g. (Bakry et al., 2014)), where $C \leq \text{poly}(d)$ is the diameter of the support of $\psi(\pi)$. Thus, it is sufficient to choose $T = \log(\frac{C}{\varepsilon})$ to ensure $\|\mathcal{L}_{Y_T} - \pi\|_{TV} = \|q_T - N(0, I_d)\|_{TV} \leq O(\varepsilon)$.

As (17) holds for all $t \in \tau\mathbb{N}$, the distribution $\nu = \mathcal{L}_{\hat{y}_T}$ of our sampling algorithm's output satisfies, since $\tau = \frac{1}{c}$,

$$\|\pi - \nu\|_{TV} = \|\mathcal{L}_{Y_T} - \pi\|_{TV} + \|\mathcal{L}_{Y_T} - \nu\|_{TV} \leq O\left(\varepsilon + \varepsilon c \frac{T}{\tau}\right) = O\left(\varepsilon c^2 \log\left(\frac{dC}{\varepsilon}\right)\right) = \tilde{O}(\varepsilon \times \text{poly}(d)).$$

**Step 5: Bounding the runtime.** Since our accuracy bound requires $T = \log(dC/\varepsilon)$, and requires a time-step size of $\Delta = cd \leq \frac{1}{\text{poly}(d)}$, the number of iterations is bounded by $\frac{T}{\Delta} = cdT \leq O\left(\text{poly}(d) \times \log\left(dC/\varepsilon\right)\right).$

**Step 6: Extension of sampling guarantees to special orthogonal group.** Similar techniques can be used in the case of the special orthogonal group. However, in the case of the special orthogonal group we encounter the additional challenge that, with high probability $\Omega(1)$, the gaps between neighboring eigenvalues $\gamma_{i+1}(t) - \gamma_i(t)$ may become exponentially small in $d$, over very short time intervals of length $O(\frac{1}{e^d})$. Over these intervals, our diffusion moves at $\exp(d)$ velocity. Despite this, we show that a $1/\text{poly}(d, \frac{1}{\delta})$ step size is sufficient to simulate a *random* solution to its SDE with *probability distribution* $\delta$-close to the correct distribution. Specifically, from the matrix calculus formula for $\varphi$ one can show that the SDE for the eigenvectors of the forward diffusion satisfy (these evolution equations, discovered by Dyson, are referred to as Dyson Brownian motion (Dyson, 1962))

$$d\gamma_i(t) = dB_{ii}(t) + \sum_{j \neq i} \frac{dt}{\gamma_i(t) - \gamma_j(t)}, \tag{18}$$

$$du_i(t) = \sum_{j \neq i} \frac{dB_{ij}(t)}{\gamma_i(t) - \gamma_j(t)} u_j(t) - \frac{1}{2} \sum_{j \neq i} \frac{dt}{(\gamma_i(t) - \gamma_j(t))^2} u_i(t), \quad \forall i \in [n]. \tag{19}$$

Roughly speaking, this implies that only the interactions in (19) between eigenvectors with neighboring eigenvalues which fall below $O(\frac{1}{n^{10}})$ are significant, while interactions between eigenvectors with larger eigenvalue gaps are negligible over these short time intervals. Thus, one can analyze the evolution of the eigenvectors over these short time intervals as a collection of separable two-body problems consisting of interactions between pair(s) of eigenvectors with a closed-form transition kernel given by the invariant measure on $SO(2)$. For a detailed sketch of how one can extend our proof to the case of the special orthogonal group, see Appendix B.7.

# B. Full proof of Theorem 2.2

In the following, we denote by $\rho(x, y)$ the geodesic distance between $x, y \in \mathcal{M}$, and by $\Gamma_{x \to y}(v)$ the parallel transport of a vector $v \in \mathcal{T}_x$ from $x$ to $y$. For convenience, we denote $\varphi_i(\cdot) := \varphi(\cdot)[i]$. Recall that we have assumed that $\psi(\mathcal{M})$ is contained in a ball of radius $C = \text{poly}(d)$. We will prove our results under the more general assumption (Assumption B.1($\psi, \pi, C$)), which is satisfied whenever $\psi(\mathcal{M}) \leq C$.

**Assumption B.1 (Bounded Support ($\psi, \pi, C$)).** The pushforward of $\psi(\pi)$ of $\pi$ with respect to the map $\psi : \mathcal{M} \to \mathbb{R}^d$ has support on a ball of radius $C$ centered at 0.

## B.1. Correctness of the training objective functions

**Lemma B.2.** *$f^\star$ and $g^\star$ are solutions to the following optimization problems:*

$$\min_{f \in \mathcal{C}(\mathbb{R}^d, \mathbb{R}^d)} \mathbb{E}_{t \sim \text{Unif}([0,1])} \mathbb{E}_{b \sim \pi} \left[ \left\| (\nabla\varphi(Z_{T-t}))^\top \frac{Z_{T-t} - \psi(b)e^{-\frac{1}{2}(T-t)}}{e^{-(T-t)} - 1} \right. \right.$$
$$\left. \left. + \frac{1}{2}\text{tr}(\nabla^2\varphi(Z_{T-t})) - f(\varphi(Z_{T-t}), t) \right\|^2 \, \middle| \, Z_0 = \psi(b) \right],$$

$$\min_{g \in \mathcal{C}(\mathbb{R}^d, \mathbb{R}^{d \times d})} \mathbb{E}_{t \sim \text{Unif}([0,1])} \mathbb{E}_{b \sim \pi} \left[ \left\| (J_\varphi(Z_{T-t}))^\top J_\varphi(Z_{T-t}) - (g(\varphi(Z_{T-t}), t))^2 \right\|_F^2 \, \middle| \, Z_0 = \psi(b) \right],$$

*where $J_\varphi$ denotes the Jacobian of $\varphi$.*

*Proof.* **Step 1: Obtaining an expression for the reverse diffusion SDE in $\mathbb{R}^d$.** We cannot in general directly apply (1) to obtain a tractable expression for the SDE of the reverse diffusion $Y_t$ in $\mathcal{M}$, since we do not have a tractable formula for the transition kernel $p_t$ of the forward diffusion $X_t$ on $\mathcal{M}$. Instead, we will first obtain an SDE for the reverse diffusion of $Z_t$ in $\mathbb{R}^d$, and then "project" this SDE onto $\mathcal{M}$. Let $H_t := Z_{T-t}$ denote the time-reversed diffusion of $Z_t$. $H_t$ is a diffusion in $\mathbb{R}^d$. From (1), we have that the SDE for the reverse diffusion $H_t$ on $\mathbb{R}^d$ is given by the following formula:

$$dH_t = \left(\frac{1}{2}H_t + 2\nabla \log q_{T-t}(H_t)\right)dt + dW_t, \tag{20}$$

where $W_t$ is a standard Brownian motion on $\mathbb{R}^d$. Equation (20) can be re-written as

$$dH_t = \left(\frac{1}{2}H_t + 2\mathbb{E}_{b\sim q_{0|t}(\cdot|H_t)}[\nabla \log q_{T-t|0}(H_t|b)]\right)dt + dW_t. \tag{21}$$

The r.h.s. of (21) is tractable since we have a tractable expression for the transition kernel $q_{T-t|0}$ (it is just a time re-scaling of a Gaussian kernel, the transition kernel of Brownian motion).

**Step 2: Obtaining an expression for the reverse diffusion SDE in $\mathcal{M}$.** Note that there exists a coupling between $Z_t$ and $H_t$ such that $H_t = Z_{T-t}$ and that $Y_t = X_{T-t}$ for all $t \in [0,T]$. Thus, under this choice of coupling, we have that $Y_t = X_{T-t} = \varphi(Z_{T-t}) = \varphi(H_t)$ for all $t \in [0,T]$. In the special case when there is only one datapoint $x_0$, the SDE for the reverse diffusion $Y_t$ on $\mathcal{M}$ can be obtained by applying Itô's lemma (Lemma 3.1) to $Y_t = \varphi(H_t)$:

$$dY_t[i] = \nabla\varphi_i(H_t)^\top dH_t + \frac{1}{2}(dH_t)^\top(\nabla^2\varphi_i(H_t))dH_t \qquad \forall i \in [d]. \tag{22}$$

In the following, to simplify notation, we drop the "$i$" index from the notation $\varphi_i$ and $dY_t[i]$. Unfortunately, the r.h.s. of (22) is not a (deterministic) function of $Y_t = \varphi(H_t)$, since $\varphi$ is not an invertible map. To solve this problem, we can take the conditional expectation of (22) with respect to $Y_t = \varphi(H_t)$:

$$dY_t = E[dY_t|Y_t] = E[dY_t|\varphi(H_t)] = E\left[\nabla\varphi(H_t)^\top dH_t + \frac{1}{2}(dH_t)^\top(\nabla^2\varphi(H_t))dH_t \middle| \varphi(H_t)\right]. \tag{23}$$

The drift term on the r.h.s. of (23) is a deterministic function of $Y_t$. Denote this function by $f^\star : \mathcal{M} \times [0,T] \to \mathcal{TM}$ for any input $x \in \mathcal{M}$ and output in the tangent space $\mathcal{T}_x\mathcal{M}$ of $\mathcal{M}$ at $x$.

Moreover, by (1), the covariance term on the r.h.s. of (23) must be the same as the covariance term for the forward diffusion $Y_t$ on $\mathcal{M}$. This covariance term can be obtained from the covariance term $dW_t$ on $\mathbb{R}^d$, via Itô's lemma, implying that the covariance term is $\mathbb{E}[\nabla\varphi(H_t)[i]^\top dW_t|\varphi(H_t)]$, $i \in [d]$. Using the notation for the Jacobian, this covariance term can be written more concisely as $\mathbb{E}[J_\varphi(H_t)^\top dW_t|\varphi(H_t)]$. Thus, the diffusion term is also a deterministic function $g^\star$ of $Y_t = \varphi(H_t)$, where $g^\star(Y_t)$ is a symmetric $d \times d$ matrix,

$$E[J_\varphi(H_t)dW_t|\varphi(H_t)] = g^\star(Y_t,t)d\tilde{W}_t, \tag{24}$$

where $\tilde{W}_t$ is a standard Brownian motion on $\mathcal{M}$.

Since $dW_t$ is the derivative of a standard Brownian motion in $\mathbb{R}^d$, and $d\tilde{W}_t$ is the derivative of a standard Brownian motion on the tangent space of $\mathcal{M}$, we have that

$$E[J_\varphi(H_t)^\top J_\varphi(H_t)|\varphi(H_t)] = (g^\star(Y_t,t))^2. \tag{25}$$

Thus, (23) can be expressed as:

$$dY_t = E\left[\nabla\varphi(H_t)^\top dH_t + \frac{1}{2}(dH_t)^\top(\nabla^2\varphi(H_t))dH_t \middle| \varphi(H_t)\right] = f^\star(Y_t,t)dt + g^\star(Y_t,t)d\tilde{W}_t. \tag{26}$$

In the more general setting when there is more than one datapoint, (26) generalizes to:

$$\begin{aligned} dY_t &= \mathbb{E}_{b\sim\pi}\left[E\left[\nabla\varphi(H_t)^\top dH_t + \frac{1}{2}(dH_t)^\top(\nabla^2\varphi(H_t))dH_t \middle| \varphi(H_t), H_T = b\right]\right] \tag{27}\\ &= f^\star(Y_t,t)dt + g^\star(Y_t,t)d\tilde{W}_t. \tag{28} \end{aligned}$$

Since $Y_t = \varphi(H_t)$, we can bring $f^\star(Y_t, t)\mathrm{d}t$ and $g^\star(Y_t, t)\mathrm{d}\tilde{W}_t$ inside the conditional expectation:

$$\mathbb{E}_{b \sim \pi}\left[E\left[\nabla\varphi(H_t)^\top \mathrm{d}H_t + \frac{1}{2}(\mathrm{d}H_t)^\top(\nabla^2\varphi(H_t))\mathrm{d}H_t - f^\star(Y_t, t)\mathrm{d}t \middle| \varphi(H_t), H_T = b\right]\right] = g^\star(Y_t, t)\mathrm{d}\tilde{W}_t.$$

We can rewrite this as

$$\mathbb{E}_{b \sim \pi}\left[E_{\varphi(H_t)}\left[E_{H_t|\varphi(H_t)}\left[\nabla\varphi(H_t)^\top \mathrm{d}H_t + \frac{1}{2}(\mathrm{d}H_t)^\top(\nabla^2\varphi(H_t))\mathrm{d}H_t - f^\star(Y_t, t)\mathrm{d}t \middle| H_t, H_T = b\right]\right]\right]$$
$$= g^\star(Y_t, t)\mathrm{d}\tilde{W}_t.$$

This simplifies to

$$\mathbb{E}_{b \sim \pi}\left[\nabla\varphi(H_t)^\top \mathrm{d}H_t + \frac{1}{2}(\mathrm{d}H_t)^\top(\nabla^2\varphi(H_t))\mathrm{d}H_t - f^\star(Y_t, t)\mathrm{d}t \middle| H_T = b\right] = g^\star(Y_t, t)\mathrm{d}\tilde{W}_t. \tag{29}$$

where the expectation is taken over the outcomes of $H_t$. Plugging in (21) into (29), and separating the drift and the diffusion terms on both sides of the equation (and noting that the higher-order differentials $(\mathrm{d}t)^2$ and $\mathrm{d}W_t\mathrm{d}t$ vanish), we get that the drift terms satisfy

$$\mathbb{E}_{b \sim \pi}\left[(\nabla\varphi(H_t))^\top\left(H_t + 2\nabla\log q_{T-t|0}(H_t|b)\right)\mathrm{d}t + \frac{1}{2}(\mathrm{d}W_t)^\top(\nabla^2\varphi(H_t))\mathrm{d}W_t - f^\star(Y_t, t)\mathrm{d}t \middle| H_T = b\right] = 0. \tag{30}$$

Noting that $(\mathrm{d}W_t[i])^2 = \mathrm{d}t$ and $\mathrm{d}W_t[i]\mathrm{d}W_t[j] = 0$ for all $i \neq j$, we get

$$\mathbb{E}_{b \sim \pi}\left[(\nabla\varphi(H_t))^\top\left(H_t + 2\nabla\log q_{T-t|0}(H_t|b)\right)\mathrm{d}t + \frac{1}{2}\mathrm{tr}(\nabla^2\varphi(H_t))\mathrm{d}t - f^\star(Y_t, t)\mathrm{d}t \middle| H_T = b\right] = 0. \tag{31}$$

Dividing both sides by $\mathrm{d}t$, we get an expression for the drift term $f^\star$

$$\mathbb{E}_{b \sim \pi}\left[(\nabla\varphi(H_t))^\top\left(H_t + 2\nabla\log q_{T-t|0}(H_t|b)\right) + \frac{1}{2}\mathrm{tr}(\nabla^2\varphi(H_t)) - f^\star(Y_t, t) \middle| H_T = b\right] = 0. \tag{32}$$

Finally, from (25), we have that diffusion term $g^\star$ satisfies

$$\mathbb{E}_{b \sim \pi}\left[E\left[J_\varphi(H_t)^\top J_\varphi(H_t) - (g^\star(Y_t, t))^2 \middle| \varphi(H_t)\right] \middle| H_T = b\right] = 0. \tag{33}$$

**Step 3: Training the drift term.** From (32), we have that the function $f^\star$ is the solution to the following optimization problem:

$$\min_f \mathbb{E}_{t \sim \mathrm{Unif}([0,1])}\mathbb{E}_{b \sim \pi}\left[\left\|(\nabla\varphi(H_t))^\top\left(\frac{1}{2}H_t + 2\nabla\log q_{T-t|0}(H_t|b)\right) + \frac{1}{2}\mathrm{tr}(\nabla^2\varphi(H_t)) - f(Y_t, t)\right\|^2 \middle| H_T = b\right]. \tag{34}$$

where the inner expectation is taken over $b \sim \pi$ and over the outcomes of $H_t$ at time $t$ conditioned on $H_T = b$ (Note that $Y_t = \varphi(H_t)$ is a deterministic function of $H_t$).

Now, $H_t|\{H_T = b\}$ has the same probability distribution as $Z_{T-t}|\{Z_0 = b\}$ (and that $Y_t|\{H_T = b\}$ has the same probability distribution as $X_{T-t}|\{Z_0 = b\}$). Thus, we can re-write (34) as

$$\min_f \mathbb{E}_{t \sim \mathrm{Unif}([0,1])}\mathbb{E}_{b \sim \pi}\left[\left\|(\nabla\varphi(Z_{T-t}))^\top\left(Z_{T-t} + 2\nabla\log q_{T-t|0}(Z_{T-t}|b)\right)\right.\right.$$
$$\left.\left. + \frac{1}{2}\mathrm{tr}(\nabla^2\varphi(Z_{T-t})) - f(X_{T-t}, t)\right\|^2 \middle| Z_0 = b\right], \tag{35}$$

**Step 4: Training the diffusion term.** From (33) we have that $g^\star$ is the solution to the following optimization problem:

$$\min_g \mathbb{E}_{t \sim \text{Unif}([0,1])} \mathbb{E}_{b \sim \pi} \left[ \left\| J_\varphi(H_t)^\top J_\varphi(H_t) - (g(Y_t, t))^2 \right\|_F^2 \bigg| H_T = b \right],$$

where $\| \cdot \|_F$ is the Frobenius norm. Since $H_t | \{H_T = b\}$ has the same probability distribution as $Z_{T-t} | \{Z_0 = b\}$ (and $Y_t | \{H_T = b\}$ has the same probability distribution as $X_{T-t} | \{Z_0 = b\}$), we can re-write (34) as

$$\min_g \mathbb{E}_{t \sim \text{Unif}([0,1])} \mathbb{E}_{b \sim \pi} \left[ \left\| J_\varphi(Z_{T-t})^\top J_\varphi(Z_{T-t}) - (g(X_{T-t}, t))^2 \right\|_F^2 \bigg| Z_0 = b \right].$$

$\square$

## B.2. Proof of Lemma B.3

In the proof of Theorem 2.2, we will use the following lemma.

**Lemma B.3** (**Gronwall-like inequality for SDEs on a manifold of non-negative curvature**). *Suppose that $\mathcal{M}$ is a Riemannian manifold with non-negative curvature, and let $\rho(x, y)$ denote the geodesic distance between any $x, y \in \mathcal{M}$. Suppose also that $X_t$ and $\hat{X}_t$ are two diffusions on $\mathcal{M}$ such that*

$$\mathrm{d}X_t = b(X_t, t) + \sigma(X_t, t)\mathrm{d}W_t,$$

*and*

$$\mathrm{d}\hat{X}_t = \hat{b}(\hat{X}_t, t) + \hat{\sigma}(X_t, t)\mathrm{d}W_t,$$

*where $b$ is $C_1(t)$-Lipschitz and $\sigma$ is $C_2(t)$-Lipschitz at every time $t \in [0, T]$. Moreover, assume that*

$$\|b(x, t) - \hat{b}(x, t)\| \le \varepsilon$$

*and*

$$\|\sigma(x, t) - \hat{\sigma}(x, t)\|_F^2 \le \varepsilon$$

*for all $x \in \mathcal{M}$, $t \in [0, T]$. Then there exists a coupling between $X_t$ and $\hat{X}_t$ such that, for all $t \ge 0$,*

$$\mathbb{E}[\rho^2(\hat{X}_t, X_t)] \le \left( \mathbb{E}[\rho^2(\hat{X}_0, X_0)] + \inf_{s \in [0,t]} \frac{5\varepsilon^2}{2C_1(s) + 3C_2(s)^2 + 2} \right) e^{\int_0^t (2C_1(s) + 3C_2(s)^2 + 2)\mathrm{d}s}.$$

*Proof of Lemma B.3.* We first couple $X_t$ and $\hat{X}_t$ by setting their underlying Brownian motion terms $\mathrm{d}W_t$ to be equal to each other. Next, we compute the distance $\rho^2(\hat{X}_t, X_t)$ using Itô's Lemma. Letting $h(x, y) := \rho^2(x, y)$, by Itô's Lemma we have that

$$
\begin{aligned}
\mathrm{d}\rho^2(\hat{X}_t, X_t) &= \mathrm{d}h(\hat{X}_t, X_t) \\
&= \nabla h(\hat{X}_t, X_t)^\top \begin{pmatrix} b(X_t, t) \\ \hat{b}(\hat{X}_t, t) \end{pmatrix} \mathrm{d}t \\
&\quad + \frac{1}{2} \text{Tr}\left[ \begin{pmatrix} \sigma(X_t, t) & 0 \\ \hat{\sigma}(X_t, t) & 0 \end{pmatrix}^\top [\nabla^2 h(\hat{X}_t, X_t)] \begin{pmatrix} \sigma(X_t, t) & 0 \\ \hat{\sigma}(X_t, t) & 0 \end{pmatrix} \right] \mathrm{d}t \\
&\quad + \nabla h(\hat{X}_t, X_t)^\top \begin{pmatrix} \sigma(X_t, t) & 0 \\ \hat{\sigma}(X_t, t) & 0 \end{pmatrix} \mathrm{d}\begin{pmatrix} W_t \\ \hat{W}_t \end{pmatrix}.
\end{aligned}
$$

Therefore,

$$
\begin{aligned}
\mathrm{d}\mathbb{E}[\rho^2(\hat{X}_t, X_t)] &= \mathbb{E}\left[ \nabla h(\hat{X}_t, X_t)^\top \begin{pmatrix} b(X_t, t) \\ \hat{b}(\hat{X}_t, t) \end{pmatrix} \right] \mathrm{d}t \\
&\quad + \frac{1}{2}\mathbb{E}\left[ \text{Tr}\left[ \begin{pmatrix} \sigma(X_t, t) & 0 \\ \hat{\sigma}(X_t, t) & 0 \end{pmatrix}^\top [\nabla^2 h(\hat{X}_t, X_t)] \begin{pmatrix} \sigma(X_t, t) & 0 \\ \hat{\sigma}(X_t, t) & 0 \end{pmatrix} \right] \right] \mathrm{d}t \\
&\quad + 0.
\end{aligned}
\tag{36}
$$

Now, since $\mathcal{M}$ has non-negative curvature, by the Rauch comparison theorem we have

$$
\left| \nabla h(\hat{X}_t, X_t)^\top \begin{pmatrix} b(X_t, t) \\ \hat{b}(\hat{X}_t, t) \end{pmatrix} \right| \leq 2\rho(\hat{X}_t, X_t) \times \|\hat{b}(\hat{X}_t, t) - \Gamma_{X_t \to \hat{X}_t}(b(X_t, t))\|
$$
$$
\leq 2\rho(\hat{X}_t, X_t) \times \left( \|b(\hat{X}_t, t) - \Gamma_{X_t \to \hat{X}_t}(b(X_t, t))\| + \|b(\hat{X}_t, t) - \hat{b}(\hat{X}_t, t)\| \right)
$$
$$
\leq 2\rho(\hat{X}_t, X_t) \times (C_1(t)\rho(\hat{X}_t, X_t) + \varepsilon). \tag{37}
$$

where the last inequality holds since $b$ is $C_1(t)$-Lipschitz. Moreover, since $\mathcal{M}$ has non-negative curvature, by the Rauch comparison theorem, we also have that

$$
\frac{1}{2}\mathrm{Tr}\left[ \begin{pmatrix} \sigma(X_t, t) & 0 \\ \hat{\sigma}(X_t, t) & 0 \end{pmatrix}^\top [\nabla^2 h(\hat{X}_t, X_t)] \begin{pmatrix} \sigma(X_t, t) & 0 \\ \hat{\sigma}(X_t, t) & 0 \end{pmatrix} \right]
$$
$$
\leq \left\| \hat{\sigma}(\hat{X}_t, t) - \Gamma_{X_t \to \hat{X}_t}(\sigma(X_t, t)) \right\|_F^2
$$
$$
\leq \left( \left\| \sigma(\hat{X}_t, t) - \Gamma_{X_t \to \hat{X}_t}(\sigma(X_t, t)) \right\|_F + \left\| \hat{\sigma}(\hat{X}_t, t) - \sigma(\hat{X}_t, t) \right\|_F \right)^2
$$
$$
\leq 3 \left\| \sigma(\hat{X}_t, t) - \Gamma_{X_t \to \hat{X}_t}(\sigma(X_t, t)) \right\|_F^2 + 3 \left\| \hat{\sigma}(\hat{X}_t, t) - \sigma(\hat{X}_t, t) \right\|_F^2
$$
$$
\leq 3C_2(t)^2 \rho^2(\hat{X}_t, X_t) + 3\varepsilon^2. \tag{38}
$$

Plugging (37) and (38) into (36), we have

$$
\frac{\mathrm{d}}{\mathrm{d}t}\mathbb{E}[\rho^2(\hat{X}_t, X_t)] \leq 2\mathbb{E}[C_1(t)\rho^2(\hat{X}_t, X_t) + \varepsilon\rho(\hat{X}_t, X_t)] + 3C_2(t)^2 \mathbb{E}[\rho^2(\hat{X}_t, X_t)] + 3\varepsilon^2 \qquad \forall t \geq 0. \tag{39}
$$

Hence,

$$
\frac{\mathrm{d}}{\mathrm{d}t}\mathbb{E}[\rho^2(\hat{X}_t, X_t)] \leq 2\mathbb{E}[C_1(t)\rho^2(\hat{X}_t, X_t) + \rho^2(\hat{X}_t, X_t)] + 3C_2(t)^2 \mathbb{E}[\rho^2(\hat{X}_t, X_t)] + 5\varepsilon^2
$$
$$
= 2\mathbb{E}[C_1(t)\rho^2(\hat{X}_t, X_t) + \rho^2(\hat{X}_t, X_t)] + 3C_2(t)^2 \mathbb{E}[\rho^2(\hat{X}_t, X_t)] + 5\varepsilon^2
$$
$$
= (2C_1(t) + 3C_2(t)^2 + 2)\mathbb{E}[\rho^2(\hat{X}_t, X_t)] + 5\varepsilon^2.
$$

Let $\tau \in [0, T]$ be some number, and define $R(t) := \mathbb{E}[\rho^2(\hat{X}_t, X_t)] + \inf_{s \in [0, \tau]} \frac{5\varepsilon^2}{2C_1(s) + 3C_2(s)^2 + 2}$ for all $t \in [0, \tau]$. Then we have,

$$
\frac{\mathrm{d}}{\mathrm{d}t}R(t) \leq (2C_1(t) + 3C_2(t)^2 + 2)R(t), \qquad \forall t \geq 0. \tag{40}
$$

Thus, plugging (40) into Gronwall's lemma, we have, for all $t \geq 0$,

$$
R(t) \leq R(0)e^{\int_0^t (2C_1(s) + 3C_2(s)^2 + 2\mathrm{d}s}
$$
$$
= \left( \mathbb{E}[\rho^2(\hat{X}_0, X_0)] + \inf_{s \in [0, \tau]} \frac{5\varepsilon^2}{2C_1(s) + 3C_2(s)^2 + 2} \right) e^{\int_0^t 2C_1(s) + 3C_2(s)^2 + 2\mathrm{d}s}.
$$

Thus,

$$
\mathbb{E}[\rho^2(\hat{X}_t, X_t)] + \inf_{s \in [0, \tau]} \frac{5\varepsilon^2}{2C_1(s) + 3C_2(s)^2 + 2}
$$
$$
\leq \left( \mathbb{E}[\rho^2(\hat{X}_0, X_0)] + \inf_{s \in [0, T]} \frac{5\varepsilon^2}{2C_1(s) + 3C_2(s)^2 + 2} \right) e^{\int_0^t 2C_1(s) + 3C_2(s)^2 + 2\mathrm{d}s}.
$$

Hence, for all $t \geq 0$,

$$
\mathbb{E}[\rho^2(\hat{X}_t, X_t)] \leq \left( \mathbb{E}[\rho^2(\hat{X}_0, X_0)] + \inf_{s \in [0, \tau]} \frac{5\varepsilon^2}{2C_1(s) + 3C_2(s)^2 + 2} \right) e^{\int_0^t 2C_1(s) + 3C_2(s)^2 + 2\mathrm{d}s}.
$$

Plugging in $\tau = t$ in the above equation, we have, for all $t \geq 0$,

$$
\mathbb{E}[\rho^2(\hat{X}_t, X_t)] \leq \left( \mathbb{E}[\rho^2(\hat{X}_0, X_0)] + \inf_{s \in [0, t]} \frac{5\varepsilon^2}{2C_1(s) + 3C_2(s)^2 + 2} \right) e^{\int_0^t 2C_1(s) + 3C_2(s)^2 + 2\mathrm{d}s}.
$$

$\square$

**B.3. Proof that average-case Lipschitzness holds on symmetric manifolds of interest (Lemma B.4)**

**Lemma B.4** (**Average-case Lipschitzness**). *For the unitary group, Assumption 2.1 holds with $L_1 = O(d^{1.5}\sqrt{T}\alpha^{-1/3})$ and $L_2 = O(d^2 T \alpha^{-2/3})$. For the sphere, it holds for $L_1 = L_2 = O(\alpha^{-\frac{1}{d}})$. For the torus, it holds for $L_1 = L_2 = 1$.*

*Proof.* For the torus, the map $\varphi(x)$ has $\nabla\varphi(x) = I_d$ at every $x \in \mathbb{R}^d$, which implies that Assumption 2.1 is satisfied for $L_1 = L_2 = 1$.

*Sphere.* In the case of the sphere, which we embed via the map $\psi$ as a unit sphere in $\mathbb{R}^d$, one can easily observe that e.g. $\|\nabla\varphi(z)\| \leq O(1)$ for any $z$ outside a ball of radius $r \geq \Omega(1)$ centered at the origin. As the volume of a ball of radius $r = \alpha$ is $\frac{1}{r^d}$ times the volume of the unit ball, one can use standard Gaussian concentration inequalities to show that the Ornstein-Uhlenbeck process $Z_t$, which is a Gaussian process, will remain outside this ball for time $T$ with probability at least $1 - 4\frac{1}{r^d T}$.

Moreover, by standard Gaussian concentration inequalities (Rudelson & Vershynin, 2013), we have that $\|Z_t\| \leq 2\sqrt{Td}\log(\frac{1}{\alpha})$ with probability at least $1 - 2\alpha$ for all $t \in [0, T]$. This motivates defining the set $\Omega_t := \{z \in \mathbb{R}^d : (4\frac{1}{\alpha T})^{\frac{1}{d}} \leq \|z\| \leq 2\sqrt{Td}\log(\frac{1}{\alpha})\}$, as we then have

$$\mathbb{P}(Z_t \in \Omega_t \ \forall \ t \in [0, T]) \geq 1 - \alpha.$$

Since $\|z\| \geq (4\frac{1}{\alpha T})^{\frac{1}{d}}$ for any $z \in \Omega_t$ and any $t \in [0, T]$, we must have that

$$\|\nabla\varphi(z(U, \Lambda))\|_{2\to 2} \leq 3\left(4\frac{1}{\alpha T}\right)^{\frac{2}{d}} = L_1,$$

$$\left\|\frac{\mathrm{d}}{\mathrm{d}U}\nabla\varphi(z(U, \Lambda))\right\|_{2\to 2} \leq 3\left(4\frac{1}{\alpha T}\right)^{\frac{2}{d}} = L_1,$$

$$\|\nabla^2\varphi(z(U, \Lambda))\|_{2\to 2} \leq 3\left(4\frac{1}{\alpha T}\right)^{\frac{3}{d}} = L_2,$$

$$\left\|\frac{\mathrm{d}}{\mathrm{d}U}\nabla\varphi(z(U, \Lambda))\right\|_{2\to 2} \leq 3\left(4\frac{1}{\alpha T}\right)^{\frac{3}{d}} = L_2,$$

$$\left\|\frac{\mathrm{d}}{\mathrm{d}U}(z(U, \Lambda))\right\|_{2\to 2} \leq \|x\|,$$

**Unitary group.** We next show that the Lipschitz property holds for the unitary group $\mathrm{U}(n)$. Similar techniques can be used for the case of the special orthogonal group, and we omit those details. We first recall results from random matrix theory, which allow us to bound the eigenvalue gaps of a matrix with Gaussian entries. Specifically, these results say that, roughly speaking, if $Z_0 \in \mathbb{C}^{n\times n}$ is any matrix and $Z_t = Z_0 + B(t)$, where $B(t)$ is a matrix with (complex) iid $N(0, t)$ entries undergoing Brownian motion, one has that the eigenvalues $\gamma_1(t) \geq \cdots \geq \gamma_n(t)$ of $Z_t + Z_t^*$ satisfy (see e.g. (Anderson et al., 2010; Mangoubi & Vishnoi, 2023; 2025))

$$\mathbb{P}\left(\inf_{s\in[t_0,T]}(\gamma_{i+1}(t) - \gamma_i(t)) \leq s\frac{1}{\mathrm{poly}(d)\sqrt{t}}\right) \leq O(s^{\frac{1}{2}}) \qquad \forall s \geq 0. \tag{41}$$

Thus, if we define $\Omega_t$ to be the set of outcomes of $Z_t$ such that $\gamma_{i+1}(t) - \gamma_i(t) \leq \alpha^2\frac{1}{\mathrm{poly}(n)\sqrt{t}}$, we have that $\mathbb{P}(Z_t \in \Omega_t \ \forall t \in [t_0, T]) \geq 1 - \alpha$.

From the matrix calculus formulas for $\nabla\varphi(U^\top\Lambda U)$, $\frac{\mathrm{d}}{\mathrm{d}U}\nabla\varphi(U^\top\Lambda U)$, $\nabla\varphi(U^\top\Lambda U)$, and $\frac{\mathrm{d}}{\mathrm{d}U}\nabla^2\varphi(U^\top\Lambda U)$, we have that,

for all $z(U, \Lambda) = U\Lambda U^\top \in \Omega_t$,

$$\|\nabla\varphi(z(U, \Lambda))\|_{2\to 2} \leq \sum_{i=1}^{d} \frac{1}{\lambda_{i+1} - \lambda_i} \leq d^{1.5}\sqrt{t}\alpha^{-\frac{1}{3}} = L_1,$$

$$\left\|\frac{\mathrm{d}}{\mathrm{d}U}\nabla\varphi(z(U, \Lambda))\right\|_{2\to 2} \leq \|\Lambda\|_{2\to 2}\sum_{i=1}^{d} \frac{1}{\lambda_{i+1} - \lambda_i}$$

$$\leq \left(C + \sqrt{T}d\log\left(\frac{1}{\alpha}\right)\right) \times \sum_{i=1}^{d} \frac{1}{\lambda_{i+1} - \lambda_i} \leq d^{1.5}\sqrt{t}\alpha^{-\frac{1}{3}} = L_1,$$

$$\left\|\nabla^2\varphi(z(U, \Lambda))\right\|_{2\to 2} \leq \sum_{i=1}^{d} \frac{1}{(\lambda_{i+1} - \lambda_i)^2} \leq d^2 t\alpha^{-\frac{2}{3}} = L_2,$$

$$\left\|\frac{\mathrm{d}}{\mathrm{d}U}\nabla\varphi(z(U, \Lambda))\right\|_{2\to 2} \leq \|\Lambda\|_{2\to 2}\sum_{i=1}^{d} \frac{1}{(\lambda_{i+1} - \lambda_i)^2}$$

$$\leq \left(C + \sqrt{T}d\log\left(\frac{1}{\alpha}\right)\right) \times \sum_{i=1}^{d} \frac{1}{(\lambda_{i+1} - \lambda_i)^2} \leq d^2 t\alpha^{-\frac{2}{3}} = L_2,$$

$$\left\|\frac{\mathrm{d}}{\mathrm{d}U}(z(U, \Lambda))\right\|_{2\to 2} \leq \|\Lambda\|_{2\to 2}$$

since $\lambda_{i+1} - \lambda_i \leq \alpha^{\frac{1}{3}} \frac{1}{\sqrt{d}\sqrt{t}}$ for all $i \in [d]$ and $\|\Lambda\|_{2\to 2} \leq 2\sqrt{T}d\log(\frac{1}{\alpha})$ whenever $z(U, \Lambda) \in \Omega_t$

$\square$

## B.4. Proof of Lipschitzness of $f^\star$ and $g^\star$ on all of $\mathcal{M}$ (Lemma B.6)

Recall that we denote by $q_{t|\tau}(y|x)$ the transition kernel of the Ornstein-Uhlenbeck process $Z_t$ at any $x, y \in \mathbb{R}^d$, and by $q_t(x) = \int_{\mathcal{M}} q_{t|0}(x|z)\pi(z)\mathrm{d}z$ the distribution of $Z_t$ at any time $t \geq 0$. We will use the following Proposition of (Chen et al., 2023b):

**Proposition B.5** (**Proposition 20 of (Chen et al., 2023b)**). *Suppose that $\psi(\pi)$ has support on a ball of radius $C > 0$. For any $\alpha > 0$, define the "early stopping time" $t_0 := \min(\frac{\alpha}{C}, \frac{\alpha^2}{d})$. Then the drift term $\nabla\log q_t(\cdot)$ of the reverse diffusion SDE in Euclidean space is $O(\frac{1}{\alpha^2}dC^2(\min(C, \sqrt{d})^2))$-Lipschitz at every time $t > t_0$. Moreover, $W_2(q_{t_0}, \pi) \leq \alpha$.*

Recall that we denote by $\Gamma_{x\to y}(v)$ the parallel transport of a vector $v$ from $x$ to $y$.

**Lemma B.6.** *Suppose that Assumption 2.1($\varphi, L_1, L_2, \alpha$) and Assumption B.1($\psi, \pi, C$) both hold. Then for every $t \in [t_0, T]$,*

$$\|f^\star(x, t) - \Gamma_{x\to y}(f^\star(x, t))\| \leq \mathcal{C} \times \rho(x, y), \qquad \forall x, y \in \mathcal{M} \tag{42}$$

*and*

$$\|g^\star(y, t) - \Gamma_{x\to y}(g^\star(x, t))\|_F \leq \mathcal{C} \times \rho(x, y) \qquad \forall x, y \in \mathcal{M} \tag{43}$$

*where $\mathcal{C} := (C + \sqrt{T}d\log(\frac{1}{\alpha}))^4 \times L_3^2 \times L_1 + (C + \sqrt{T}d\log(\frac{1}{\alpha}))^2 \times L_3 \times L_2$ and $t_0 := \min(\frac{\alpha}{C}, \frac{\alpha^2}{d})$, and $L_3 = O(\frac{1}{\alpha^2}dC^2(\min(C, \sqrt{d})^2))$.*

*Proof.* Recall that (when, e.g., $\mathcal{M}$ is one of the aforementioned symmetric manifolds) we may decompose any $z \in \mathbb{R}^d$ as $z \equiv z(U, \Lambda)$ where $U \in \mathcal{M}$.

We have the following expression for $f^\star(U, t)$

$$f^\star(U, t) = c_U \int_{\Lambda \in \mathcal{A}} \left[(\nabla\varphi(z(U, \Lambda)))^\top \nabla\log q_{T-t|0}(z(U, \Lambda)) + \frac{1}{2}\mathrm{tr}(\nabla^2\varphi(z(U, \Lambda)))\right]$$

$$\times q_{T-t}(z(U, \Lambda))\mathbb{1}_\Omega(\Lambda)\mathrm{d}\Lambda,$$

where $c_U = \left( \int_{\Lambda \in \mathcal{A}} q_{T-t}(z(U, \Lambda)) \mathbb{1}_\Omega(\Lambda) \mathrm{d}\Lambda \right)^{-1}$ is a normalizing constant. Then

$$
\begin{aligned}
\frac{\mathrm{d}}{\mathrm{d}U} f^\star(U, t) = c_U \times \frac{\mathrm{d}}{\mathrm{d}U} \int_{\Lambda \in \mathcal{A}} &\left[ (\nabla \varphi(z(U, \Lambda)))^\top \nabla \log q_{T-t}(z(U, \Lambda)) + \frac{1}{2} \mathrm{tr}(\nabla^2 \varphi(z(U, \Lambda))) \right] \\
&\times q_{T-t}(z(U, \Lambda)) \mathbb{1}_\Omega(\Lambda) \mathrm{d}\Lambda \\
+ \left( \frac{\mathrm{d}}{\mathrm{d}U} c_U \right) \times \int_{\Lambda \in \mathcal{A}} &\left[ (\nabla \varphi(z(U, \Lambda)))^\top \nabla \log q_{T-t}(z(U, \Lambda)) + \frac{1}{2} \mathrm{tr}(\nabla^2 \varphi(z(U, \Lambda))) \right] \\
&\times q_{T-t}(z(U, \Lambda)) \mathbb{1}_\Omega(\Lambda) \mathrm{d}\Lambda.
\end{aligned}
\tag{44}
$$

For the first term on the r.h.s. of (44) we have,

$$
\begin{aligned}
c_U \times \frac{\mathrm{d}}{\mathrm{d}U} \int_{\Lambda \in \mathcal{A}} &\left[ (\nabla_U \varphi(z(U, \Lambda)))^\top \nabla \log q_{T-t}(z(U, \Lambda)) + \frac{1}{2} \mathrm{tr}(\nabla^2 \varphi(z(U, \Lambda))) \right] \\
&\times q_{T-t}(z(U, \Lambda)) \mathbb{1}_\Omega(\Lambda) \mathrm{d}\Lambda \\
= c_U \times \int_{\Lambda \in \mathcal{A}} &\left( \frac{\mathrm{d}}{\mathrm{d}U} \left[ (\nabla \varphi(z(U, \Lambda)))^\top \nabla \log q_{T-t}(z(U, \Lambda)) + \frac{1}{2} \mathrm{tr}(\nabla^2 \varphi(z(U, \Lambda))) \right] \right) \\
&\times q_{T-t}(z(U, \Lambda)) \mathbb{1}_\Omega(\Lambda) \mathrm{d}\Lambda \\
+ c_U \times \int_{\Lambda \in \mathcal{A}} &\left[ (\nabla \varphi(z(U, \Lambda)))^\top \nabla \log q_{T-t}(z(U, \Lambda)) + \frac{1}{2} \mathrm{tr}(\nabla^2 \varphi(z(U, \Lambda))) \right] \\
&\times \frac{\mathrm{d}}{\mathrm{d}U} q_{T-t}(z(U, \Lambda)) \mathbb{1}_\Omega(\Lambda) \mathrm{d}\Lambda \\
= c_U \times \int_{\Lambda \in \mathcal{A}} &\left( \frac{\mathrm{d}}{\mathrm{d}U} \left[ (\nabla \varphi(z(U, \Lambda)))^\top \nabla \log q_{T-t}(z(U, \Lambda)) + \frac{1}{2} \mathrm{tr}(\nabla^2 \varphi(z(U, \Lambda))) \right] \right) \\
&\times q_{T-t}(z(U, \Lambda)) \mathbb{1}_\Omega(\Lambda) \mathrm{d}\Lambda \\
+ c_U \times \int_{\Lambda \in \mathcal{A}} &\left[ (\nabla \varphi(z(U, \Lambda)))^\top \nabla \log q_{T-t}(z(U, \Lambda)) + \frac{1}{2} \mathrm{tr}(\nabla^2 \varphi(z(U, \Lambda))) \right] \\
&\times \nabla_U \log q_{T-t}(z(U, \Lambda)) \times q_{T-t}(z(U, \Lambda)) \mathbb{1}_\Omega(\Lambda) \mathrm{d}\Lambda \\
= \mathbb{E}_{z(U, \Lambda) \sim q_{T-t}} &\left[ \frac{\mathrm{d}}{\mathrm{d}U} \left( (\nabla \varphi(z(U, \Lambda)))^\top \nabla_U \log q_{T-t|0}(z(U, \Lambda)) \right. \right. \\
&\left. \left. + \frac{1}{2} \mathrm{tr}(\nabla^2 \varphi(z(U, \Lambda))) \right) \mathbb{1}_\Omega(\Lambda) \middle| V = U \right] \\
+ \mathbb{E}_{z(U, \Lambda) \sim q_{T-t}} &\left[ \left( (\nabla \varphi(z(U, \Lambda)))^\top \nabla_U \log q_{T-t}(z(U\Lambda)) + \frac{1}{2} \mathrm{tr}(\nabla^2 \varphi(z(U, \Lambda))) \right) \right. \\
&\left. \times \nabla_U \log q_{T-t}(z(U, \Lambda)) \mathbb{1}_\Omega(\Lambda) \middle| V = U \right].
\end{aligned}
$$

For the second term on the r.h.s. of (44) we have,

$$
\begin{aligned}
\frac{\mathrm{d}}{\mathrm{d}U} c_U &= c_U^2 \int_{\Lambda \in \mathcal{A}} \frac{\mathrm{d}}{\mathrm{d}U} (q_{T-t}(z(U, \Lambda))) \mathbb{1}_\Omega(\Lambda) \mathrm{d}\Lambda \\
&= c_U^2 \int_{\Lambda \in \mathcal{A}} \frac{\mathrm{d}}{\mathrm{d}U} (e^{\log q_{T-t}(z(U, \Lambda))}) \mathbb{1}_\Omega(\Lambda) \mathrm{d}\Lambda \\
&= c_U^2 \int_{\Lambda \in \mathcal{A}} \nabla_U \log q_{T-t}(z(U, \Lambda)) (e^{\log q_{T-t}(z(U, \Lambda))}) \mathbb{1}_\Omega(\Lambda) \mathrm{d}\Lambda \\
&= c_U^2 \int_{\Lambda \in \mathcal{A}} \nabla_U \log q_{T-t}(z(U, \Lambda)) \times q_{T-t}(z(U, \Lambda)) \mathbb{1}_\Omega(\Lambda) \mathrm{d}\Lambda \\
&= c_U \times \mathbb{E}_{z(U, \Lambda) \sim q_{T-t}} \left[ \nabla_U \log q_{T-t}(z(U, \Lambda)) \mathbb{1}_\Omega(\Lambda) \middle| V = U \right]
\end{aligned}
$$

and hence,

$$
(\frac{\mathrm{d}}{\mathrm{d}U}c_U) \times \int_{\Lambda \in \mathcal{A}} \left[ (\nabla\varphi(z(U,\Lambda)))^\top \nabla \log q_{T-t}(z(U,\Lambda)) + \frac{1}{2}\mathrm{tr}(\nabla^2\varphi(z(U,\Lambda))) \right]
$$
$$
\times q_{T-t}(z(U,\Lambda))\mathbb{1}_\Omega(\Lambda)\mathrm{d}\Lambda
$$
$$
= \mathbb{E}_{z(U,\Lambda)\sim q_{T-t}} \left[ \nabla_U \log q_{T-t}(z(U,\Lambda))\mathbb{1}_\Omega(\Lambda) \,\middle|\, V = U \right]
$$
$$
\times \mathbb{E}_{z(U,\Lambda)\sim q_{T-t}} \left[ \left( (\nabla\varphi(z(U,\Lambda)))^\top \nabla \log q_{T-t}(z(U,\Lambda)) \right.\right.
$$
$$
\left.\left. + \frac{1}{2}\mathrm{tr}(\nabla^2\varphi(z(U,\Lambda))) \right) \mathbb{1}_\Omega(\Lambda) \,\middle|\, V = U \right].
$$

Thus

$$
\frac{\mathrm{d}}{\mathrm{d}U}f^\star(U,t) = \mathbb{E}_{z(U,\Lambda)\sim q_{T-t}} \left[ \frac{\mathrm{d}}{\mathrm{d}U}\left( (\nabla\varphi(z(U,\Lambda)))^\top \nabla_U \log q_{T-t|0}(z(U,\Lambda)) \right.\right.
$$
$$
\left.\left. + \frac{1}{2}\mathrm{tr}(\nabla^2\varphi(z(U,\Lambda))) \right) \mathbb{1}_\Omega(\Lambda) \middle| V = U \right],
$$
$$
+ \mathbb{E}_{z(U,\Lambda)\sim q_{T-t}} \left[ \left( (\nabla\varphi(z(U,\Lambda)))^\top \nabla_U \log q_{T-t}(z(U,\Lambda)) + \frac{1}{2}\mathrm{tr}(\nabla^2\varphi(z(U,\Lambda))) \right) \right.
$$
$$
\left. \times \nabla_U \log q_{T-t}(z(U,\Lambda))\mathbb{1}_\Omega(\Lambda) \middle| V = U \right]
$$
$$
+ \mathbb{E}_{z(U,\Lambda)\sim q_{T-t}} \left[ \nabla_U \log q_{T-t}(z(U,\Lambda))\mathbb{1}_\Omega(\Lambda) \,\middle|\, V = U \right]
$$
$$
\times \mathbb{E}_{z(U,\Lambda)\sim q_{T-t}} \left[ \left( (\nabla\varphi(z(U,\Lambda)))^\top \nabla \log q_{T-t}(z(U,\Lambda)) \right.\right. \tag{45}
$$
$$
\left.\left. + \frac{1}{2}\mathrm{tr}(\nabla^2\varphi(z(U,\Lambda))) \right) \mathbb{1}_\Omega(\Lambda) \,\middle|\, V = U \right]. \tag{46}
$$

Moreover, by standard Gaussian concentration inequalities and Assumption B.1, without loss of generality we have that $\|z(U,\Lambda)\|_F \le C + \sqrt{T}d\log(\frac{1}{\alpha})$ for all $z(U,\Lambda) \in \Omega_t$. From Proposition B.5 we have that $\nabla \log p_{T-t|0}(z(U,\Lambda))$ is $L_3$-Lipschitz where $L_3 := O(\frac{1}{\alpha^2}dC^2(\min(C,\sqrt{d})^2))$ and hence that

$$
\|\nabla_U \log p_{T-t|0}(z(U,\Lambda))\|_{2\to2} \le \|\frac{\mathrm{d}}{\mathrm{d}U}(z(U,\Lambda))\|_{2\to2} \times \|\nabla \log p_{T-t|0}(z(U,\Lambda))\|_{2\to2}
$$
$$
\le \|\frac{\mathrm{d}}{\mathrm{d}U}(z(U,\Lambda))\|_{2\to2} \times L_3 \times \|z(U,\Lambda)\|_F
$$
$$
\le L_3 \times \|z(U,\Lambda)\|_F^2
$$
$$
\le L_3 \times \left( C + \sqrt{T}d\log(\frac{1}{\alpha}) \right)^2, \tag{47}
$$

where the third inequality holds by Assumption 2.1, and the last inequality holds since $\|z(U,\Lambda)\|_F \le C + \sqrt{T}d\log(\frac{1}{\alpha})$ for all $z(U,\Lambda) \in \Omega_t$. Thus, plugging Assumption 2.1 and (47) into (45), we have that

$$
\left\| \frac{\mathrm{d}}{\mathrm{d}U}f^\star(U,t) \right\|_{2\to2} \le (C + \sqrt{T}d\log(\frac{1}{\alpha}))^4 \times L_3^2 \times L_1 + (C + \sqrt{T}d\log(\frac{1}{\alpha}))^2 \times L_3 \times L_2. \tag{48}
$$

Replacing $f^\star$ with $g^\star$ in the above calculation, we also get that

$$
\left\| \frac{\mathrm{d}}{\mathrm{d}U}g^\star(U,t) \right\|_{2\to2} \le (C + \sqrt{T}d\log(\frac{1}{\alpha}))^4 \times L_3^2 \times L_1 + (C + \sqrt{T}d\log(\frac{1}{\alpha}))^2 \times L_3 \times L_2. \tag{49}
$$

Thus, (48) and (49) imply that

$$\|f^\star(y,t) - \Gamma_{x\to y}(f^\star(x,t))\| \le \mathcal{C} \times \rho(x,y), \qquad \forall x,y \in \mathcal{M} \tag{50}$$

and

$$\|g^\star(y,t) - \Gamma_{x\to y}(g^\star(x,t))\|_F \le \mathcal{C} \times \rho(y,x) \qquad \forall x \in \mathcal{M}, \tag{51}$$

where $\mathcal{C} := (C + \sqrt{T}d\log(\frac{1}{\alpha}))^4 \times L_3^2 \times L_1 + (C + \sqrt{T}d\log(\frac{1}{\alpha}))^2 \times L_3 \times L_2$.

□

### B.5. Wasserstein to TV conversion on the manifold (Lemma B.7)

**Lemma B.7** (**Wasserstein to TV conversion on the manifold**). *There is a number $c \le \mathrm{poly}(d)$ such that for every $t \in [t_0, T]$ and any $\tau \le \frac{1}{c}$ we have*

$$\|\mathcal{L}_{Y_{t+\tau+\hat{\Delta}}} - \mathcal{L}_{\hat{y}_{t+\tau+\hat{\Delta}}}\|_{\mathrm{TV}} - \|\mathcal{L}_{Y_t} - \mathcal{L}_{\hat{y}_t}\|_{\mathrm{TV}}$$
$$\le \sqrt{D_{\mathrm{KL}}(\nu_1 \,\|\, p_{t+\tau+\hat{\Delta}|t+\tau}(\,\cdot\,|Y_{t+\tau}))} + \sqrt{D_{\mathrm{KL}}(\nu_1 \| \mathcal{L}_{\hat{y}_{t+\tau+\hat{\Delta}}|\hat{y}_t})} \le O(\varepsilon c). \tag{52}$$

*Proof of Lemma B.7.* Now that we have shown that $f^\star$ and $g^\star$ are $\mathrm{poly}(d)$-Lipschitz (by Lemmas B.4 and B.6), we can apply Lemma B.3 to bound the Wasserstein distance: $W_2(\hat{Y}_{t+\tau}, Y_{t+\tau}) \le (\rho^2(\hat{Y}_t, Y_t) + \varepsilon)e^{c\tau} \qquad \forall \tau \ge 0$, where $c \le \mathrm{poly}(d)$.

Moreover, with slight abuse of notation, we may define $\hat{y}_{t+\tau}$ to be a continuous-time interpolation of the discrete process $\hat{y}$. Applying (13) to this process, we get that, roughly, $W_2(\hat{Y}_{t+\tau}, \hat{y}_{t+\tau}) \le (\rho^2(\hat{y}_t, Y_t) + \varepsilon + \Delta)e^{c\tau}$ for $\tau \ge 0$. Thus, we get a bound on the Wasserstein error,

$$W_2(Y_{t+\tau}, \hat{y}_{t+\tau}) \le W_2(\hat{Y}_{t+\tau}, Y_{t+\tau}) + W_2(\hat{Y}_{t+\tau}, \hat{y}_{t+\tau}) \le (\rho^2(\hat{y}_t, Y_t) + \varepsilon + \Delta)e^{c\tau} \qquad \tau \ge 0 \tag{53}$$

Unfortunately, after times $\tau > \frac{1}{c} = \frac{1}{\mathrm{poly}(d)}$, this bound grows exponentially with the dimension $d$.

To overcome this challenge, we define a new coupling between $Y_t$ and $\hat{Y}_t$ which we "reset" after time intervals of length $\tau = \frac{1}{c}$ by converting our Wasserstein bound into a total variation bound after each time interval. Towards this end, we use the fact that if at any time $t$ the total variation distance satisfies $\|\mathcal{L}_{Y_t} - \mathcal{L}_{\hat{y}_t}\|_{\mathrm{TV}} \le \alpha$, then there exists a coupling such that $Y_t = \hat{Y}_t$ with probability at least $1 - \alpha$. In other words, w.p. $\ge 1 - \alpha$, we have $\rho(\hat{y}_{t+\tau}, Y_{t+\tau}) = 0$, and we can apply inequality (53) over the next time interval of $\tau$ without incurring an exponential growth in time. Repeating this process $\frac{T}{\tau}$ times, we get that $\|\mathcal{L}_{Y_T} - \mathcal{L}_{\hat{y}_T}\| \le \alpha \times \frac{T}{\tau}$, where the TV error grows only *linearly* with $T$.

**Converting Wasserstein bounds on the manifold to TV bounds.** To complete the proof, we still need to show how to convert the Wasserstein bound into a TV bound. Towards this end, we begin by showing that the transition kernel $\tilde{p}_{t+\tau+\hat{\Delta}|t+\tau}(\,\cdot\,|H_{t+\tau})$ of the reverse diffusion $H_t$ in $\mathbb{R}^d$ is close to a Gaussian in KL distance over short time steps $\hat{\Delta}$:

$$D_{\mathrm{KL}}(N(H_{t+\tau} + \hat{\Delta}\nabla\tilde{p}_{T-t-\tau}(H_{t+\tau}), \hat{\Delta}I_d) \,\|\, \tilde{p}_{t+\tau+\hat{\Delta}|t+\tau}(\,\cdot\,|H_{t+\tau})) \le \frac{\alpha\tau}{T}.$$

One can do this using Girsanov's theorem, since, unlike the diffusion $Y_t$ on the manifold, the reverse diffusion in Euclidean space $H_t$ *does* have a constant diffusion term (see e.g. Theorem 9 of (Chen et al., 2023b)).

Next, we use the fact that with probability at least $1 - \alpha\frac{\tau}{T}$ the map $\varphi$ in a ball of radius $\frac{1}{\mathrm{poly}(d)}$ about the point $H_{t+\tau}$ has $c$-Lipschitz Jacobian where $c = \mathrm{poly}(d)$, and that the inverse of the exponential map $\exp(\cdot)$ has $O(1)$-Lipschitz Jacobian, to show that the transition kernel $p_t$ of $Y_t = \varphi(H_t)$ satisfies

$$D_{\mathrm{KL}}(\nu_1 \,\|\, p_{t+\tau+\hat{\Delta}|t+\tau}(\,\cdot\,|Y_{t+\tau})) \le (1 + \hat{\Delta}c)^d \frac{\alpha\tau}{T} \le 2\frac{\alpha\tau}{T},$$

if we choose $\hat{\Delta} \le O(\frac{1}{cd})$, where $\nu_1 := \exp_{Y_{t+\tau}}(N(Y_{t+\tau} + \hat{\Delta}f^\star(Y_{t+\tau}, t+\tau), \hat{\Delta}g^{\star 2}(Y_{t+\tau}, t+\tau)I_d))$.

Next, we plug in our Wasserstein bound $W(Y_{t+\tau}, \hat{y}_{t+\tau}) \le O(\varepsilon)$ into the formula for the KL divergence between two Gaussians to bound $\|\mathcal{L}_{Y_{t+\tau+\hat{\Delta}}} - \mathcal{L}_{\hat{y}_{t+\tau+\hat{\Delta}}}\|_{\mathrm{TV}}$. Specifically, noting that $\mathcal{L}_{\hat{y}_{t+\tau+\hat{\Delta}}|\hat{y}_t} = \exp_{\hat{y}_{t+\tau}}(N(\hat{y}_{t+\tau} + \hat{\Delta}f(\hat{y}_{t+\tau}, t+$

$\tau), \hat{\Delta} g^2(\hat{y}_{t+\tau}, t + \tau) I_d))$, we have that

$$D_{\mathrm{KL}}(\nu_1, \mathcal{L}_{\hat{y}_{t+\tau+\hat{\Delta}}|\hat{y}_{t+\tau}}) = \mathrm{Tr}\left((g^{\star 2}(Y_{t+\tau}, t + \tau))^{-1} g^2(\hat{y}_{t+\tau}, t + \tau)\right)$$
$$- d + \log \frac{\det g^{\star 2}(Y_{t+\tau}, t + \tau)}{\det g^2(\hat{y}_{t+\tau}, t + \tau)} + w^\top (\hat{\Delta} g^{\star 2}(Y_{t+\tau}, t))^{-1} w,$$

where $w := Y_{t+\tau} - \hat{y}_{t+\tau} + \hat{\Delta}(f^\star(Y_{t+\tau}, t + \tau) - f(\hat{y}_{t+\tau}, t + \tau))$. Since with probability $\geq 1 - \alpha \frac{\tau}{T}$ we have $g^\star(Y_{t+\tau}) \succeq$ poly$(d)$, plugging in the error bounds $\|f^\star(Y_{t+\tau}, t) - f(Y_{t+\tau}, t)\| \leq \varepsilon$ and $\|g^\star(Y_{t+\tau}, t) - g(Y_{t+\tau}, t)\|_F \leq \varepsilon$ and the $c$-Lipschitz bounds on $f^\star$ and $g^\star$ due to Lemmas B.4 and B.6, where $c = $ poly$(d)$, we get that $D_{\mathrm{KL}}(\nu_1, \mathcal{L}_{\hat{y}_{t+\tau+\hat{\Delta}}}) \leq O(\varepsilon^2 c^2)$. Thus, by Pinsker's inequality, we have

$$\|\mathcal{L}_{Y_{t+\tau+\hat{\Delta}}} - \mathcal{L}_{\hat{y}_{t+\tau+\hat{\Delta}}}\|_{\mathrm{TV}} - \|\mathcal{L}_{Y_t} - \mathcal{L}_{\hat{y}_t}\|_{\mathrm{TV}}$$
$$\leq \sqrt{D_{\mathrm{KL}}(\nu_1 \| p_{t+\tau+\hat{\Delta}|t+\tau}(\cdot|Y_{t+\tau}))} + \sqrt{D_{\mathrm{KL}}(\nu_1 \| \mathcal{L}_{\hat{y}_{t+\tau+\hat{\Delta}}|\hat{y}_t})} \leq O(\varepsilon c). \quad (54)$$

$\square$

## B.6. Completing the proof of Theorem 2.2

**Bounding the accuracy.** Recall that $q_t$ is the distribution of the forward diffusion $Z_t$ in Euclidean space after time $t$, which is an Ornstein-Uhlenbeck process. Standard mixing bounds for Ornstein-Uhlenbeck process imply that

$$\|q_t - N(0, I_d)\|_{\mathrm{TV}} \leq O(Ce^{-t})$$

for all $t > 0$ (see e.g. (Bakry et al., 2014)). Thus, it is sufficient to choose $T = \log(\frac{1}{C\varepsilon})$ to ensure that

$$\|\mathcal{L}_{Y_T} - \pi\|_{\mathrm{TV}} = \|q_T - N(0, I_d)\|_{\mathrm{TV}} \leq O(\varepsilon).$$

As Lemma B.7 holds for all $t \in [t_0, T]$, the distribution $\nu = \mathcal{L}_{\hat{y}_T}$ of our sampling algorithm's output satisfies

$$\|\pi - \nu\|_{\mathrm{TV}} = \|\mathcal{L}_{Y_T} - \pi\|_{\mathrm{TV}} + \|\mathcal{L}_{Y_T} - \nu\|_{\mathrm{TV}} \leq O(\varepsilon) + O(\varepsilon c \times \frac{T}{\tau}) = O(\varepsilon \times \mathrm{poly}(d)).$$

**Bounding the runtime of the sampling algorithm.** Since our accuracy bound requires $T = \log(\frac{d}{\varepsilon C})$, and requires a time-step size of $\Delta \leq \frac{1}{\mathrm{poly}(d)}$, the number of iterations is bounded by

$$\frac{T}{\Delta} \leq O\left(\mathrm{poly}(d) \times \log\left(\frac{d}{\varepsilon C}\right)\right).$$

## B.7. Proof sketch for extension of sampling guarantees to special orthogonal group

Similar techniques to those used in the case of the complex unitary group can be used to bound the accuracy and runtime of our sampling algorithm in the case of the real special orthogonal group. However, in the case of the special orthogonal group we encounter the additional challenge that, due to weaker "electrical repulsion" between eigenvalues of real-valued random matrices, with high probability $\Omega(1)$ the gaps between neighboring eigenvalues $\gamma_{i+1}(t) - \gamma_i(t)$ may become exponentially small in $d$, over very short time intervals of length $O(\frac{1}{e^d})$. To overcome this challenge, we first note that one can show that the gaps between *non-neighboring* eigenvalues do satisfy a polynomial lower bound at every time $t$ w.h.p. (see e.g. (Anderson et al., 2010; Mangoubi & Vishnoi, 2023; 2025)):

$$\mathbb{P}\left(\bigcap_{t \in [t_0, T]} \left\{\gamma_{i+2}(t) - \gamma_i(t) \leq s \frac{1}{\sqrt{n}\sqrt{t}}\right\}\right) \leq O\left(s^{1.5}\right). \quad (55)$$

Moreover, one can also show that, except over at most $O(n^{1.5})$ "bad" time intervals $[\tau_j, \tau_j + \Delta_j]$, each of length e.g. $O(\frac{1}{n^5})$, the gaps between all neighboring eigenvalues are at least $\frac{1}{n^{10}}$.

From the matrix calculus formula for $\varphi$ one can show that the SDE for the eigenvectors of the forward diffusion satisfy (these evolution equations, discovered by Dyson, are referred to as Dyson Brownian motion (Dyson, 1962))

$$d\gamma_i(t) = dB_{ii}(t) + \sum_{j \neq i} \frac{dt}{\gamma_i(t) - \gamma_j(t)}, \tag{56}$$

$$du_i(t) = \sum_{j \neq i} \frac{dB_{ij}(t)}{\gamma_i(t) - \gamma_j(t)} u_j(t) - \frac{1}{2} \sum_{j \neq i} \frac{dt}{(\gamma_i(t) - \gamma_j(t))^2} u_i(t) \qquad \forall i \in [n]. \tag{57}$$

From (55), one can see that over the "bad" time intervals $[a_i, b_i]$, each eigenvalue $\gamma_i(t)$ has at most one neighboring eigenvalue, say $\gamma_{i+1}(t)$, with small gap $\gamma_i(t) - \gamma_{i+1}(t) \leq O(\frac{1}{\sqrt{d}})$ w.h.p. Roughly speaking, this implies that only the interactions in (57) between eigenvectors with neighboring eigenvalues that fall below $O(\frac{1}{n^{10}})$ are significant, while interactions between eigenvectors with larger eigenvalue gaps are negligible over these short time intervals. Thus, one can analyze the evolution of the eigenvectors (41) over these short time intervals as a collection of separable two-body problems consisting of interactions between pair(s) of eigenvectors.

More precisely, using (56), one can show that over the bad time intervals $[\tau_j, \tau_j + \Delta_j]$, any eigenvalue gap which falls below $\frac{1}{n^{10}}$, also remains below $\frac{1}{n^8}$ over a time sub-interval of length at least $\Omega(\frac{1}{(n^8)^2})$ w.h.p. This is because eigenvalues $\gamma_i(t)$ and $\gamma_{i+1}(t)$ in (56) repel with an "electrical force" proportional to $\frac{1}{\gamma_i(t) - \gamma_{i+1}(t)}$, which implies that the eigenvalues gaps expand at a rate proportional to $\sqrt{t}$ (the stochastic term $dB_{ii}(t)$ also leads to the same $\sqrt{t}$ expansion rate). Thus, using the evolution equations (57), one can show that, over the short bad intervals, the distribution of $[u_i(\tau_j + \Delta_j), u_{i+1}(\tau_j + \Delta_j)]|U(\tau_j)$ is $\frac{1}{\text{poly}(d)}$-close in Wasserstein distance to the invariant (Haar) measure with respect to the action of $SO(2)$ on $[u_i(\tau_j), u_{i+1}(\tau_j)]$. This is because (by the Itô isometry) the time-averaged variance in $u_i(t)$ over this time interval is proportional to $\frac{1}{\Delta_j} \int_{\tau_j}^{\tau_j + \Delta_j} \frac{1}{(\gamma_i(t) - \gamma_j(t))^2} dt \approx \int_{n^{10}}^{n^8} \frac{1}{(\sqrt{t})^2} dt = \log(\frac{1}{n^8}) - \log(\frac{1}{n^{10}}) = (10 - 8)\log(n) = \Omega(\log(n))$. But the diameter of the 2-dimensional manifold (which is isomorphic to $SO(2)$) on which $u_i(t)$ and $u_{i+1}(t)$ (approximately) lie is $O(1)$. Thus, after the time interval $[\tau_j, \tau_j + \Delta_j]$, the two neighboring eigenvectors $[u_i(\tau_j + \Delta_j), u_{i+1}(\tau_j + \Delta_j)]|U(\tau_j)$ are (approximately) distributed according to the Haar measure with respect to the action of $SO(2)$ on $[u_i(\tau_j), u_{i+1}(\tau_j)]$. Thus, one can show that as long as one uses a numerical step size $\Delta \leq O(\frac{1}{\text{poly}(d)})$, the transition kernel of both the continuous-time reverse diffusion $Y_t$ and the numerical simulation $\hat{y}_t$ over the time interval $[\tau_j, \tau_j + \Delta_j]$ will be very close (within Wasserstein distance $\frac{\Delta}{\text{poly}(d)}$) to the Haar measure on the action of $SO(2)$ on $[u_i(\tau_j), u_{i+1}(\tau_j)]$. As the Lipschitz property does hold outside the bad intervals $[\tau_j, \tau_j + \Delta_j]$, the remainder of the proof follows in the same way as for the case of $U(n)$.

## C. Additional simulation details

### C.1. Datasets

Given a $d$-dimensional Riemannian manifold $\mathcal{M}$, a number of mixture components $k \in \mathbb{N}$, points $m_1, \ldots, m_k \in \mathcal{M}$ and covariance matrices $C_1, \ldots, C_k \in \mathbb{R}^{d \times d}$, we say that a random variable is distributed according to a wrapped Gaussian distribution with means $m_1, \cdots, m_k$ and covariances $C_1, \cdots, C_k$ (with equal weights on each component) if its distribution is equal to that of a random variable $X$ sampled as follows:

1. Sample an index $i$ at random from $\{1, \ldots, k\}$.

2. Sample $Z \sim N(0, C_i)$

3. Set $X = \exp_{m_i}(Z)$,

where $\exp_x(\cdot)$ denotes the exponential map at any point $x \in \mathcal{M}$.

**Datasets on the Torus** $\mathbb{T}_d$**.** The synthetic dataset is sampled from a single-wrapped Gaussian distribution, with mean at the origin, $(0, \ldots, 0)^\top$ and covariance matrix $0.2 I_d$. A total of 30,000 points were sampled as the training dataset, and 10,000 were sampled as a test dataset to compute the log-likelihood of the generative model outputs.

**Datasets on the special orthogonal group** $SO(n)$**.** The dataset is constructed by first picking 2 random means $m_1, m_2 \in SO(n)$ sampled from the uniform measure on the orthogonal group $SO(n)$ (i.e., the invariant measure with respect to actions

by the orthogonal group). We then sample 40,000 matrices from the wrapped Gaussian mixture distribution on $\mathrm{SO}(n)$ with means $m_1, m_2$ and covariance $0.2I_d$. 30,000 of these matrices are used for training, and the remaining 10,000 matrices comprise the test dataset used to evaluate the C2ST score of the generative model outputs.

**Datasets on the unitary group** $\mathrm{U}(n)$. We use a dataset on $\mathrm{U}(n)$ of unitary matrices representing time-evolution operators $e^{itH}$ of a quantum oscillator. Here $t$ is time, $H = \frac{\hbar}{2m}\Delta - V$ is a Hamiltonian, and $\Delta$ is the Laplacian. $V$ is a random potential $V(x) = \frac{\omega^2}{2}\|x - x_0\|^2$ with angular momentum $\omega$ sampled uniformly on $[2, 3]$ and $x_0 \sim \mathcal{N}(0, 1)$. As $\Delta, V$ are infinite-dimensional operators, matrices in $\mathrm{U}(n)$ are obtained by retaining the (discretized) top-$n$ eigenvectors of $\Delta, V$.

### C.2. Neural Network architecture, Training Hyperparameters, and hardware

**Torus.** In the case of the torus, the neural network architecture consists of a 4-layer MLP with a hidden dimension of $k$, with a $\sin$ activation function. We set $k = 512$ for $d < 1000$ and $k = 2048$ for $d = 1000$.

The models were trained with a batch size of 512, with an appropriate variance scheduler. For each model, we trained the neural networks for 50K iterations when $d < 1000$, and for $100K$ iterations when $d = 1000$.

**Special Orthogonal Group.** In the case of the special orthogonal group, the neural network architecture consists of a 4-layer MLP with a hidden dimension $k = 512$, with a $\sin$ activation function.

For each model, the neural networks were trained for 100K iterations, with a batch size of 512, and an appropriate variance scheduler.

**Unitary Group.** Following TDM (Zhu et al., 2025), when training each model on the unitary group, we use a more complicated neural network than on the torus and special orthogonal group, to accommodate the more complicated quantum evolution operator datasets used in our simulations on the unitary group. For both the drift and diffusion terms, let $(X_i, t_i)$ be inputs into the neural network, where $X_i \in \mathrm{U}(n)$ and $t_i \in [0, T]$ is time. The output of the neural network is then given by

$$\hat{X}_i = \mathrm{MLP}(\mathrm{N_G}(\mathrm{MLP_S}(X_i), \mathrm{Emb_{sin}}(t_i)))$$

where MLP is a 2-layer multi-layer perceptron of dimension $D$, $\mathrm{MLP_S}$ is $k$ skip-connected MLP layers of dimension $D$, $\mathrm{Emb_{sin}}$ is a sinusoid embedding of dimension $D$, and $\mathrm{N_G}$ denotes group normalization. In our simulations, we set $k = 8$ and $D = 512$. For the drift term $\hat{f}(\cdot, \cdot)$ in each of the models, the final output is given by $X_{\mathrm{drift}} = \mathrm{proj}_{T_{X_i}\mathrm{U}(n)}(\hat{X})$, where $\mathrm{proj}_{T_{X_i}\mathrm{U}(n)}$ is the projection onto the tangent space at $X_i$. For the diffusion term $\hat{g}(\cdot, \cdot)$ in our model, the neural network outputs a vector of dimension $d$.

For each model, the neural networks were trained for 80K iterations, with a batch size of 512, and an appropriate variance scheduler.

**Hardware.** Simulations evaluating sample quality on the Torus were run on an Apple M1 chip with 10 cores. Simulations on the special orthogonal group and unitary group were run on a single RTX 3070. All simulations evaluating per-iteration training runtime were run on a single RTX 3070 as well.

### C.3. Evaluation metrics

In this section, we define the metrics used in our simulations.

**Log-likelihood metric.** Let $D = \{x_1, x_2, \ldots, x_n\}$ be a synthetic dataset arising from a target distribution with density function $g$. We train the generative model $A$ on the dataset $D$. Next, we generate points $y_1^A, \ldots, y_n^A$ which are outputs of the trained model $A$. Since the points $y_1^A, \ldots, y_n^A$ are generated independently, the likelihood of generating the points $y_1^A, \ldots, y_n^A$ given the target distribution $g$ is given by

$$\prod_{i=1}^{n} g(y_i^A) \tag{58}$$

The (average) log-likelihood of the generated points $y_1^A, \ldots, y_n^A$ with respect to the target density $g$ is therefore

$$\frac{1}{n}\sum_{i=1}^{n} \log g(y_i^A) \tag{59}$$

**C2ST metric.** Suppose we have two distributions $P, Q$ and sampled points $S_P, S_Q$ where $S_P \sim P, S_Q \sim Q$. We denote the number of sampled points by $m := |S_P| = |S_Q|$. One motivation behind the Classifier Two-Sample Test (C2ST) metric (Lopez-Paz & Oquab, 2017) is to perform a hypothesis test, where one wishes to decide whether to reject or accept the null hypothesis $P = Q$. By reporting the test statistic from this hypothesis test, the C2ST metric can also be used to evaluate the quality of samples generated by a generative model, we do in our simulations.

To compute the C2ST metric, construct a dataset $D$ where

$$\mathcal{D} = \{(x_i, 0)\}_{i=1}^m \cup \{(y_i, 1)\}_{i=1}^m := \{(z_i, l_i)\}_{i=1}^{2m}.$$

and where $x_i \in S_P$ and $y_i \in S_Q$. Partition $D$ randomly into training and test datasets $D_{tr}$ and $D_{te}$ where $m_{te} := |D_{te}|$ denotes the number of points in the test dataset. Suppose $f : D \to [0, 1]$ is a binary classifier trained on $D_{tr}$ where $f(z_i) = P(l_i = 1|z_i)$, then the value for the C2ST metric is computed as

$$\hat{t} = \frac{1}{m_{te}} \sum_{(z_i, l_i) \in \mathcal{D}_{te}} \mathbb{1}\left[\mathbb{1}\left(f(z_i) > \frac{1}{2}\right) = l_i\right] \tag{60}$$

where $\mathbb{1}$ is the indicator function. The null distribution is approximately $\mathcal{N}\left(\frac{1}{2}, \frac{1}{4m_{te}}\right)$. Then if $P = Q$ we would have $\hat{t} \to 0.5$. Whether to reject the null hypothesis can be done by performing a $p$-value analysis using 60 and the null hypothesis. For the sake of comparison between model performance, we report the value of $\hat{t}$ instead.

More specifically, as the statistic $\hat{t}$ can take values greater than or smaller than $0.5$, we report the value

$$\left|\hat{t} - \frac{1}{2}\right| + \frac{1}{2} \tag{61}$$

where $\hat{t}$ is computed using 60.

## C.4. Additional results

In this section, we provide additional empirical results.

**Visual results on the torus** $\mathbb{T}_d$. In 2 we show points generated on the torus by the Euclidean diffusion model, the RSGM, and our model when trained on a dataset sampled from a wrapped Gaussian distribution. For the 2D torus, we plot the result as a 2D scatter plot. For higher-dimensional tori, for any sampled point $x \in \mathbb{T}_d$, the plot shows the first two angle coordinates $(x_0, x_1)$ of each generated point on the torus. We observe that, for dimensions $d \geq 100$, our model appears to generate points that visually resemble those of the target distribution more closely than the points generated by the Euclidean diffusion model or the RSGM model.

**C2ST score and visual results on the special orthogonal group** $\mathrm{SO}(n)$. We train our model, a Euclidean diffusion model, RSGM, and TDM on a dataset sampled from a mixture of two wrapped Gaussian distributions on $\mathrm{SO}(n)$ for $n \in \{3, 5, 9, 12, 15\}$. For $n \geq 9$, our model achieves the lowest C2ST score; a lower C2ST score indicates higher-quality sample generation (Table 4).

The visual results for our model, the Euclidean model, the RSGM, and TDM are shown in 3. Here we plot the first and second entries of the first row of each generated matrix in $\mathrm{SO}(n)$ Note that the target distribution is bimodal, yet the two modes appear visually as a single mode as we are only plotting two of the matrix coordinates.

**Additional visual results on the unitary group** $\mathrm{U}(n)$. Visual sample generation results on $\mathrm{U}(n)$ were shown in Figure 1 of Section 4 for $n = 15$. In Figure 4, we give visual results for additional values of $n$.

**Runtime on the torus** $\mathbb{T}_d$. 5 gives the per-iteration runtime of the Euclidean model, our model, TDM, and RSGM on the torus, for dimensions $d \in \{2, 10, 50, 100, 1000\}$. We observe that for each of these dimensions, our method has similar runtime to the Euclidean model, whereas RSGM is roughly 9 times slower than the Euclidean model.

**Runtime on the special orthogonal group** $\mathrm{SO}(n)$. 6 shows the per-iteration runtime of the Euclidean model, our model, RSGM and TDM, on the special orthogonal group $\mathrm{SO}(n)$, for $n \in \{3, 5, 10, 30, 50\}$. We observe that our model's per-iteration training runtime remains within a factor of $1.3$ of the Euclidean model for all $n$. However, the per-iteration training runtimes of TDM and RSGM increase more rapidly with dimension and are, respectively, 51 and 66 times greater than the Euclidean model for $n = 50$.

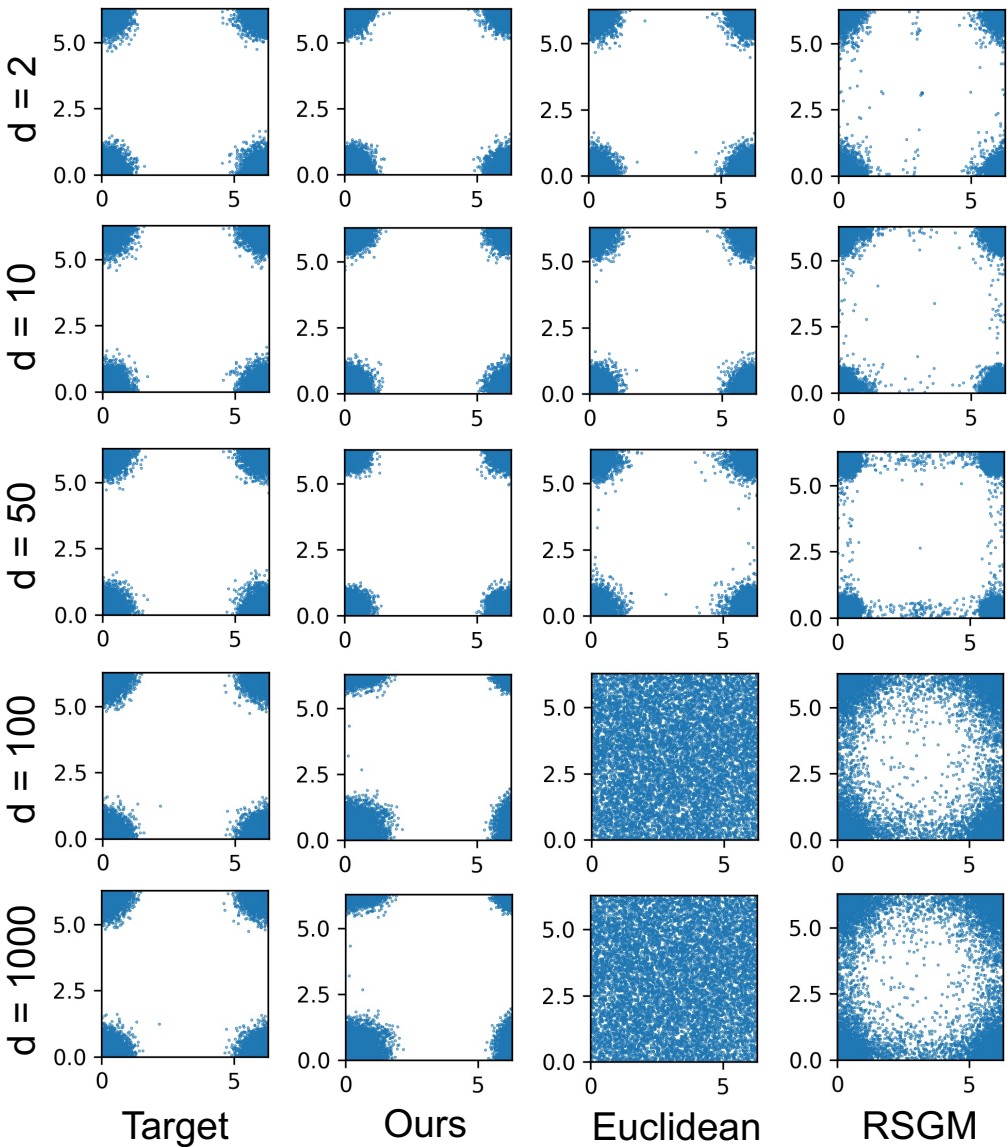

Figure 2: Points generated by different models when training on a dataset sampled from a wrapped Gaussian target distribution on the torus of different dimensions $d \in \{2, 10, 50, 100, 1000\}$.

Table 4: C2ST scores when training on a wrapped Gaussian mixture dataset on $\mathrm{SO}(n)$. Lower scores indicate better-quality sample generation (range is $[0.5, 1]$, and $0.5$ is optimal). For $n \geq 9$, our model achieves the best C2ST scores.

| Method | $n = 3$ | $n = 5$ | $n = 9$ | $n = 12$ | $n = 15$ |
|---|---|---|---|---|---|
| Euclidean | $\mathbf{.51}. \pm \mathbf{.01}$ | $\mathbf{.51}. \pm \mathbf{.01}$ | $.62. \pm .02$ | $.64. \pm .02$ | $.72. \pm .02$ |
| RSGM | $\mathbf{.51}. \pm \mathbf{.01}$ | $.57. \pm .02$ | $.74. \pm .01$ | $.81. \pm .02$ | $.90. \pm .02$ |
| TDM | $.52. \pm .01$ | $.53. \pm .01$ | $.69. \pm .03$ | $.73. \pm .02$ | $.79. \pm .03$ |
| Ours | $.55. \pm .01$ | $.56. \pm .02$ | $\mathbf{.60}. \pm \mathbf{.03}$ | $\mathbf{.61}. \pm \mathbf{.02}$ | $\mathbf{.67}. \pm \mathbf{.03}$ |

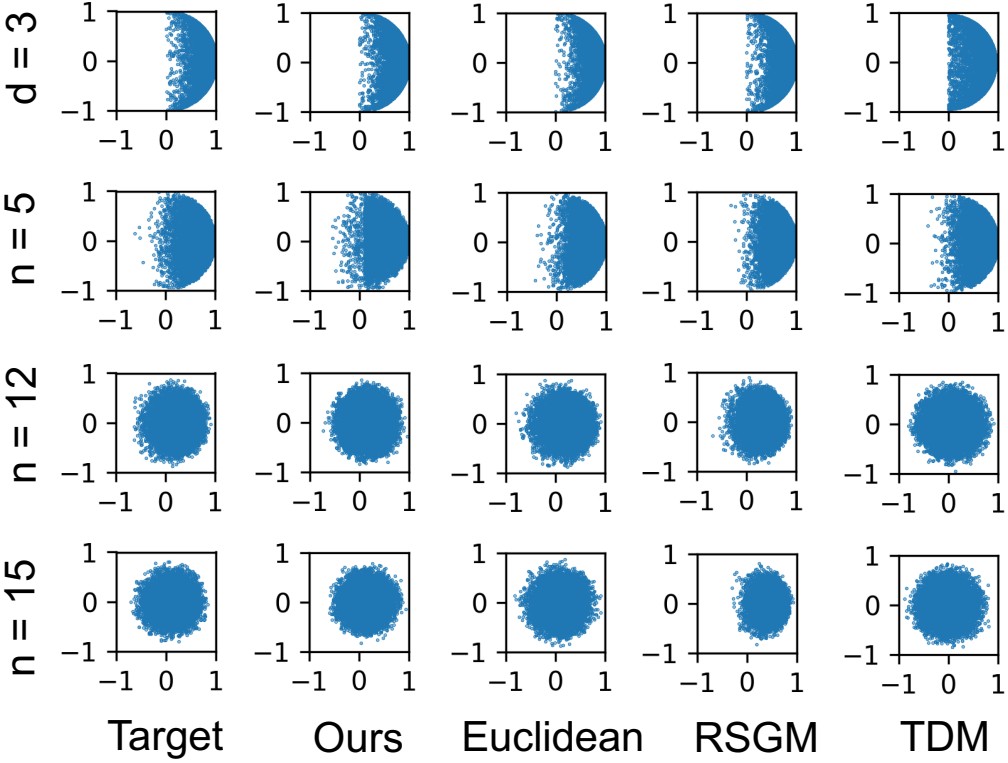

Figure 3: Points generated by different models trained on a Gaussian mixture dataset on $\mathrm{SO}(n)$ for different values of $n$.

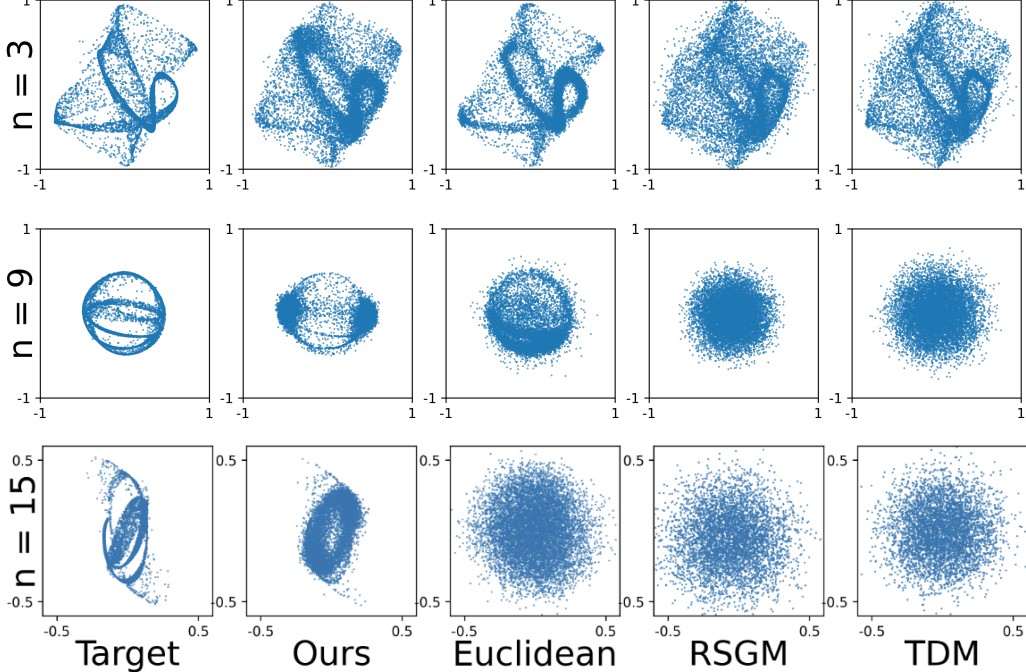

Figure 4: Points generated on $\mathrm{U}(n)$ for different values of $n$, when training on datasets comprising time-evolution operators of quantum harmonic oscillators with random potentials. For $n = 9$ and $n = 15$, we observe that our model generates samples resembling the data distribution, while the Euclidean, RSGM, and TDM models generate lower-quality samples.

Table 5: Per-iteration training runtime in seconds on the Torus. The manifold-constrained diffusion model with the fastest runtime is in bold; the Euclidean model is in gray for comparison. For each dimension $d$, our model achieves a similar runtime to the Euclidean model, whereas RSGM is roughly 9 times slower.

| Method | $d = 2$ | $d = 10$ | $d = 50$ | $d = 100$ | $d = 1000$ |
|---|---|---|---|---|---|
| Euclidean | $0.16 \pm .00$ | $0.17 \pm .00$ | $016 \pm .00$ | $0.17 \pm .00$ | $0.18 \pm .01$ |
| RSGM | $1.36 \pm .09$ | $1.35 \pm .11$ | $1.39 \pm .07$ | $1.36 \pm .06$ | $1.42 \pm .08$ |
| **Ours** | $\mathbf{0.15 \pm .01}$ | $\mathbf{0.15 \pm .01}$ | $\mathbf{0.16 \pm .01}$ | $\mathbf{0.15 \pm .01}$ | $\mathbf{0.16 \pm .01}$ |

Table 6: Per-iteration training runtime in seconds on $\mathrm{SO}(n)$. The manifold-constrained diffusion model with the fastest runtime is in bold; the Euclidean model is in gray for comparison. Our model's runtime remains within a factor of $1.3$ of the Euclidean model for all $n$. Runtimes of TDM and RSGM increase more rapidly with dimension and are 51 and 66 times greater than the Euclidean model for $n = 50$.

| Method | $n = 3$ | $n = 5$ | $n = 10$ | $n = 30$ | $n = 50$ |
|---|---|---|---|---|---|
| Euclidean | $0.13 \pm .01$ | $0.12 \pm .01$ | $0.13 \pm .00$ | $0.12 \pm .01$ | $0.15 \pm .00$ |
| RSGM | $0.73 \pm .01$ | $0.96 \pm .08$ | $1.18 \pm .01$ | $2.99 \pm .12$ | $9.89 \pm .09$ |
| TDM | $0.62 \pm .02$ | $0.78 \pm .05$ | $1.67 \pm .04$ | $2.85 \pm .09$ | $7.63 \pm .12$ |
| **Ours** | $\mathbf{0.13 \pm .01}$ | $\mathbf{0.13 \pm .01}$ | $\mathbf{0.14 \pm .00}$ | $\mathbf{0.14 \pm .01}$ | $\mathbf{0.20 \pm .01}$ |

## D. Challenges encountered when applying Euclidean diffusion for generating points constrained to non-Euclidean symmetric manifolds

The following examples illustrate why using Euclidean diffusion models to enforce symmetric manifold constraints may be insufficient.

**Example 1.** Consider the problem of generating points from a distribution $\mu$ on the $d$-dimensional torus $\mathbb{T}_d = \mathbb{S}_1 \times \cdots \times \mathbb{S}_1$, given a dataset $D$ sampled from $\mu$. A naive approach is to map the dataset $D$ from the torus to Euclidean space via the map $\psi$, which maps each point on the torus to its angles in $[0, 2\pi)^d \subseteq \mathbb{R}^d$. One can then train a Euclidean diffusion model on the dataset $\psi(D)$.

However, the map $\psi$ can greatly distort the geometry of $\mu$. To see why, let $\mu$ be a unimodal distribution on $\mathbb{T}_d$ with mode cenetered near $(0, \ldots, 0)$. The pushforward of $\mu$ under $\psi$ consists of a distribution with $2^d$ modes, each near the $2^d$ corners of the $d$-cube $[0, 2\pi)^d$ (see Figure 5). Thus, a Euclidean diffusion model needs to learn a multimodal distribution, which may be much harder than learning a unimodal distribution.

**Example 2.** Another example is the problem of generating samples from a distribution on the manifold $\mathrm{SO}(3)$ of rotation matrices. There is a natural map $\psi$ from $\mathrm{SO}(3)$ to $\mathbb{R}^3$ which maps any $M \in \mathrm{SO}(3)$ to its three Euler angles $(a, b, c) \in [-\pi, \pi] \times [-\frac{\pi}{2}, \frac{\pi}{2}] \times [-\pi, \pi] \subseteq \mathbb{R}^3$. However, $\psi$ has a singularity at $b = \frac{\pi}{2}$, which may make it harder to learn distributions with a region of high probability density passing through this singularity, as $\psi$ may separate this region into multiple disconnected regions.

Additionally, it has been observed empirically that applying Euclidean diffusion models to generate Euler angles in $\mathbb{R}^3$ leads to samples of lower quality than those generated by diffusion models on the manifold $\mathrm{SO}(3)$; see e.g. (Leach et al., 2022), and (Watson et al., 2023).

## E. Illustration of our framework for Euclidean space, torus, special orthogonal group, and unitary group

1. **Euclidean space $\mathbb{R}^d$.** In the Euclidean case, our algorithm (with the above choice of $\varphi, \psi$) recovers the algorithms of diffusion models on $\mathbb{R}^d$ from prior works (e.g., (Ho et al., 2020; Rombach et al., 2022)). The forward diffusion

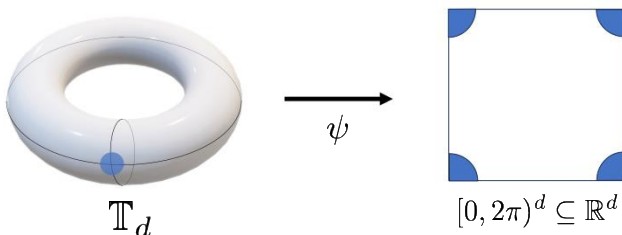

Figure 5: A probability density $\mu$ with one mode (blue) on the torus. The map $\psi$, which maps points in the $d$-dimensional torus $\mathbb{T}_d$ to Euclidean space $\mathbb{R}^d$, may break up the single mode on the torus into up to $2^d$ separated modes in $\mathbb{R}^d$. This can make the task of learning the pushforward of the target distribution on $\mathbb{R}^d$ much more challenging than the task of learning the original target distribution on the torus, as the distribution in $\mathbb{R}^d$ may have exponentially-in-$d$ more modes.

is the Ornstein-Uhlenbeck process with SDE $\mathrm{d}Z_t = -\frac{1}{2}Z_t\mathrm{d}t + \mathrm{d}B_t$ initialized at the target distribution $\pi$, where $B_t$ is the standard Brownian motion. The training objective for the drift term $f(z,t)$ of the reverse diffusion is given by $\|(\hat{z}^\top \frac{\hat{z}-be^{-\frac{1}{2}(T-t)}}{e^{-(T-t)}-1} - f(\hat{z},t)\|^2$ where $b$ is a point sampled from the dataset and $\hat{z}$ is a point sampled from $Z_{T-t}|\{Z_0 = b\}$ which is Gaussian distributed as $N(be^{-\frac{1}{2}(T-t)}, \sqrt{1-e^{-(T-t)}}I_d)$ (see Section 3.1). The number of arithmetic operations to compute the training objective is therefore the same as for previous diffusion models in Euclidean space.

2. **Torus $\mathbb{T}_d$.** For the torus, the forward and reverse diffusion of our model are the same as the models used in previous diffusion models on the torus (De Bortoli et al., 2022) (Lou et al., 2024). The Forward diffusion is given by the SDE $\mathrm{d}X_t = -\frac{1}{2}X_t\mathrm{d}t + \mathrm{d}B_t$ on the torus, initialized at the target distribution $\pi$.

The only difference is in the training objective function. To obtain our objective function, we observe that $X_t$ is the projection $X_t = \varphi(Z_t)$ of the Ornstein-Uhlenbeck diffusion on $\mathbb{R}^d$ via our choice of projection map $\varphi$ for the torus. The drift term $f$ for the reverse diffusion can be trained by minimizing the objective function $\|\hat{z}^\top \frac{\hat{z}-\psi(b)e^{-\frac{1}{2}(T-t)}}{e^{-(T-t)}-1} - f(\varphi(\hat{z}),t)\|^2$, where $\hat{z} \sim N(be^{-\frac{1}{2}(T-t)}, \sqrt{1-e^{-(T-t)}}I_d)$. Our objective function can be computed in $O(d)$ arithmetic operations, improving by an exponential factor on the per-iteration training runtime of (De Bortoli et al., 2022) which relies on an inefficient expansion of the heat kernel which requires an exponential-in-$d$ number of arithmetic operations to compute, and matching the per-iteration training runtime of (Lou et al., 2024) who derive a more efficient expansion for the heat kernel in the special case of the torus.

3. **Sphere $\mathbb{S}_{d-1}$.**

   **Forward diffusion.** We first choose the projection map $\varphi : \mathbb{R}^d \to \mathbb{S}_{d-1}$ to be $\varphi(x) = \frac{x}{\|x\|}$ for $x \in \mathbb{S}_{d-1}$, and $\psi : \mathbb{S}_{d-1} \to \mathbb{R}^d$ to be the usual embedding of the unit sphere into $\mathbb{R}^d$. We define our forward diffusion to be the projection $X_t = \varphi(Z_t)$ of the Euclidean-space Ornstein-Uhlenbeck diffusion $Z_t$ onto the manifold $\mathcal{M}$, where $Z_t$ is initialized at the pushforward $\psi(\pi)$ of the target distribution $\pi$ onto $\mathbb{R}^d$. Since the Ornstein-Uhlenbeck distribution $Z_t$ is a Gaussian process, each sample from our forward diffusion can be computed by drawing a single sample from a Gaussian distribution and computing the projection map $\varphi$ once.

   The forward and reverse diffusion of our model on the sphere are different than those of prior diffusion models on the sphere. The evolution of our forward diffusion $X_t$ on the sphere is governed by the SDE $\mathrm{d}X_t = \alpha(X_t,t)(-\frac{1}{2}X_t\mathrm{d}t + \mathrm{d}B_t)$ initialized at the target distribution $\pi$, where the coefficient $\alpha(t)$ is given by the conditional expectation $\alpha(X_t,t) := \mathbb{E}\left[\frac{1}{\|Z_t\|}\big|\varphi(Z_t) = X_t\right]$. Our forward (and reverse) diffusion has a (time-varying and) spatially-varying covariance term $\alpha(X_t,t)\mathrm{d}B_t$ not present in prior models (De Bortoli et al., 2022) (Lou et al., 2024). This covariance term, which accounts for the curvature of the sphere, allows our forward diffusion to be computed as a projection of Euclidean Brownian motion onto the sphere despite the sphere's non-zero curvature.

   **Training the model.** The SDE for the reverse diffusion of our model has both a drift and a covariance term. To train a model $f$ for the drift term, we first sample a point $b$ from the dataset $D$ at a random time $t \in [0,T]$, and point $\hat{z}$ from the Ornstein-Uhlenbeck diffusion $Z_t$ initialized at $\psi(b)$, which is Gaussian distributed. Next, we project this sample $\hat{z}$ to obtain a sample $\varphi(\hat{z})$ from our forward diffusion $X_t$ on the manifold. Finally, we plug in the point $\varphi(\hat{z})$, and

the datapoint $b$ into the training objective function for the drift term $f$, which is given by the closed-form expression $\left\| \frac{1}{\|\hat{z}\|}(I - \frac{1}{\|\hat{z}\|^2}\hat{z}\hat{z}^\top)\frac{\hat{z}-\psi(b)e^{-\frac{1}{2}(T-t)}}{e^{-(T-t)}-1} - f(\varphi(\hat{z}),t) \right\|^2$. The model for the drift term $f$ is trained by minimizing the expectation of this objective function over random samples of $b \sim D$ and $\hat{z} \sim Z_t$. To learn the SDE of the reverse diffusion, we must also train a model for the spatially-varying covariance term, which is given by a $d \times d$ covariance matrix. Learning a dense matrix model for this covariance term would require at least $d^2$ arithmetic operations. However, as a result of the symmetries of the sphere, the covariance matrix has additional structure: it is a multiple $\alpha(X_t,t)$ of the $d \times d$ identity matrix. Thus, to learn this covariance term, it is sufficient to train a model $\hat{\alpha}(X_t,t)$ for $\alpha(X_t,t)$. This can be accomplished by minimizing the objective function $(\hat{\alpha}(\varphi(\hat{z}),t) - \frac{1}{\|\hat{z}\|})^2$. Evaluating our objective functions for the drift term and covariance terms can thus be accomplished via a single evaluation of the projection map $\varphi(x) = \frac{x}{\|x\|}$, which requires $O(d \log \frac{1}{\delta})$ arithmetic operations to compute within accuracy $\delta > 0$ when generating the input to our training objective function, which is sublinear in the dimension $d^2$ of the covariance term.

In contrast, the forward diffusion used in prior diffusion models on the sphere (De Bortoli et al., 2022) (Lou et al., 2024) cannot be computed as the projection of a Euclidean Brownian motion and must instead be computed by solving an SDE (or probability flow ODE) on the sphere. This requires a number of arithmetic operations, which is a higher-order polynomial in the dimension $d$ and in the desired accuracy $\frac{1}{\delta}$ (the order of the polynomial depends on the specific SDE or ODE solver used). As their training objective function requires samples from the forward diffusion as input, the cost of computing their objective function is therefore at least a higher-order polynomial in $d$ and $\frac{1}{\delta}$ (for (De Bortoli et al., 2022) it is exponential in $d$, since their training objective relies on an inefficient expansion for the heat kernel which takes $2^d$ arithmetic operations to compute).

**Sample generation.** Once the models $f(x,t)$ and $g(x,t)$ for the drift and covariance terms of our reverse diffusion are trained, we use these models to generate samples. First, we sample a point $z$ from the stationary distribution of the Ornstein-Uhlenbeck process $Z_t$ on $\mathbb{R}^d$, which is Gaussian distributed. Next, we project this point $z$ onto the manifold to obtain a point $y = \varphi(z)$, and solve the SDE $dY_t = f(Y_t,t)dt + g(Y_t,t)dB_t$ given by our trained model for the reverse diffusion's drift and covariance over the time interval $[0,T]$, starting at the initial point $y$. To simulate this SDE we can use any off-the-shelf numerical SDE solver. The point $y_T$ computed by the numerical solver at time $T$ is the output of our sample generation algorithm.

4. **Special orthogonal group** $\mathrm{SO}(n)$ **and unitary group** $\mathrm{U}(n)$**.**

For the special orthogonal group $\mathrm{SO}(n)$ and unitary group $\mathrm{U}(n)$, the forward and reverse diffusion of our model are also different from those of previous works, as our model's diffusions have a spatially-varying covariance term to account for the non-zero curvature of these manifolds. As a result of this covariance term, our forward diffusion can be computed as a projection $\varphi$ of the Ornstein-Uhlenbeck process in $\mathbb{R}^d \equiv \mathbb{R}^{n \times n}$ (or $\mathbb{C}^{n \times n}$) onto the manifold $\mathrm{SO}(n)$ ($\mathrm{U}(n)$). This projection can be computed via a single evaluation of the singular value decomposition of a $n \times n$ matrix, which requires at most $O(n^\omega) = O(d^{\frac{\omega}{2}})$ arithmetic operations, where $\omega \approx 2.37$ is the matrix multiplication exponent and $d = n^2$ is the manifold dimension.

The forward diffusion $U(t) \in \mathrm{SO}(n)$ (or $U(t) \in \mathrm{U}(n)$) of our model is given by the system of stochastic differential equations

$$du_i(t) = \sum_{j \in [n], j \neq i} \alpha_{ij}(t)dB_{ij}u_j(t) - \frac{1}{2}\sum_{j \in [n], j \neq i}\beta_{ij}(t)u_i(t)dt, \tag{62}$$

where $\alpha_{ij}(t) := \mathbb{E}\left[\frac{1}{\lambda_i - \lambda_j}|\varphi(Z_t) = U(t)\right]$ and $\beta_{ij}(t) := \mathbb{E}\left[\frac{1}{(\lambda_i - \lambda_j)^2}|\varphi(Z_t) = U(t)\right]$ for every $i,j \in [n]$.

A model for the drift term $f$ for the reverse diffusion can be trained by minimizing the objective function $\|R - \frac{1}{2}DU - f(\varphi(\hat{z}),t)\|_F^2$ where $R$ is the matrix with $i$'th column $R_i = \frac{e^{-\frac{1}{2}(T-t)}}{e^{-(T-t)}-1}U(\lambda_i I - \Lambda)^+ U^*\psi(b)u_i$ for each $i \in [n]$, and $D$ is the diagonal matrix with $i$'th diagonal entry $D_{ii} = \sum_{j \in [n], j \neq i}\frac{1}{\lambda_i - \lambda_j}$ for each $i \in [n]$. Here, $\hat{z} = be^{-\frac{1}{2}(T-t)} + \sqrt{1 - e^{-(T-t)}}G$ where $G$ is a Gaussian random matrix with i.i.d. $N(0,1)$ entries and $U\Lambda U^*$ denotes the spectral decomposition of $\hat{z} + \hat{z}^*$.

To learn the SDE of the reverse diffusion, we must also train a model for the covariance term, which is given by a $d \times d = n^2 \times n^2$ covariance matrix. To train a model for this covariance term with runtime sublinear in the number of matrix entries $n^4$, we observe that as a result of the symmetries of the orthogonal (or unitary) group, the covariance term in (6) is fully determined by the $n^2$ scalar terms $\alpha_{ij}(t)$ for $i,j \in [n]$ and the $n \times n$ matrix $U$. Thus, to learn the

covariance term, it is sufficient to train a model $\mathcal{A}(U, t) \in \mathbb{R}^{n \times n}$ for these $n^2$ terms, which can be done by minimizing the objective function $\|\mathcal{A}(U, t) - A\|_F^2$, where $A$ is the $n \times n$ matrix with $(i, j)$'th entry $A_{ij} = \frac{1}{\lambda_i - \lambda_j}$ for $i, j \in [n]$, and $\lambda_i$ denotes the $i$'th diagonal entry of $\Lambda$.

The training objective function for both the drift and covariance term can thus be computed via a singular value decomposition of an $n \times n$ matrix (and matrix multiplications of $n \times n$ matrices), which requires at most $O(n^\omega) = O(d^{\frac{\omega}{2}})$ arithmetic operations, where $\omega \approx 2.37$ is the matrix multiplication exponent and $d = n^2$ is the manifold dimension.

In contrast, the training objectives in prior works, including (De Bortoli et al., 2022) (Lou et al., 2024), require an exponential in dimension number of arithmetic operations to compute, as they rely on the heat kernel of the manifold, which lacks an efficient closed-form expression. Instead, their training algorithm requires computing an expansion for the heat kernel of these manifolds, which is given as a sum of terms over the $d$-dimensional lattice, and one requires computing roughly $2^d$ of these terms to compute the heat kernel within an accuracy of $O(1)$.

## F. Generalization to non-symmetric manifolds

While our theoretical framework and guarantees are developed for symmetric Riemannian manifolds, it is natural to ask whether the approach can extend to more general geometries. In this section, we outline the minimal set of geometric and analytic conditions required for our guarantees—on runtime, simulation accuracy, and Lipschitz continuity—to continue holding on non-symmetric manifolds. We also illustrate, via a concrete example, how two of these conditions can be satisfied even on non-smooth, non-Riemannian domains, and discuss the challenges that arise in the absence of continuous symmetries.

Our guarantees rely on the following three key properties:

1. **Exponential map oracle.** An oracle for computing the exponential map on the manifold $\mathcal{M}$.

2. **Projection map oracle.** A projection map $\varphi : \mathbb{R}^d \to \mathcal{M}$, where $d = O(\dim(\mathcal{M}))$, along with efficient computation of its Jacobian $J_\varphi(x)$ and the trace of its Hessian $\mathrm{tr}(\nabla^2 \varphi(x))$, both of which appear in our training objective.

3. **Lipschitz SDE on $\mathcal{M}$.** The projection $Y_t = \varphi(H_t)$ of the time-reversed Euclidean Brownian motion $H_t$ must satisfy a stochastic differential equation on $\mathcal{M}$ whose drift and diffusion coefficients are $L$-Lipschitz everywhere on $\mathcal{M}$, with $L$ growing at most polynomially in $d$. This is essential for the reverse diffusion process to be simulated accurately and efficiently.

Conditions (1) and (2) may hold even when $\mathcal{M}$ lacks the high degree of symmetry assumed in our main results. For example, suppose $\mathcal{M}$ is the boundary of a compact convex polytope $K \subseteq \mathbb{R}^d$, which contains a ball of radius $r > 0$ centered at a point $p$. Although such a polytope is not a smooth manifold due to singularities at vertices and edges, its boundary is composed of piecewise flat $(d - 1)$-dimensional faces. Geodesics restricted to a single face are linear and computable efficiently, satisfying the spirit of property (1).

For property (2), one can define a projection $\varphi : \mathbb{R}^d \to \mathcal{M}$ that maps any $x \in \mathbb{R}^d$ to the point where the ray emanating from $p$ and passing through $x$ intersects the boundary $\mathcal{M}$. This projection can be computed efficiently, e.g., via binary search or ray-casting techniques.

However, property (3) is significantly harder to satisfy in such domains. The drift of the reverse SDE projected onto $\mathcal{M}$ exhibits discontinuities at the vertices and lower-dimensional faces of the polytope. In our analysis, we crucially rely on the continuous symmetries of the manifold to "smooth out" such irregularities and to prove average-case Lipschitz continuity (see the discussion on "average-case" Lipschitzness on page 6).

Even among smooth Riemannian manifolds, generalizing beyond symmetric spaces remains nontrivial. Examples include surfaces of revolution with varying curvature (e.g., tori with non-uniform cross-sections) or higher-genus manifolds such as a double torus. These lack the homogeneous structure exploited in our proofs and pose new challenges for both analysis and algorithm design. Extending our framework to such settings represents a promising direction for future research.

## G. Notation

1. **Tangent space $\mathcal{T}_x \mathcal{M}$.** Given a smooth manifold $\mathcal{M}$ and a point $x \in \mathcal{M}$, the tangent space at $x$ is denoted by $\mathcal{T}_x \mathcal{M}$.

2. **Riemannian manifold.** A Riemannian manifold is a smooth manifold $\mathcal{M}$ equipped with a Riemannian metric $g$, which assigns to each point $x \in \mathcal{M}$ a positive definite inner product $g_x : \mathcal{T}_x\mathcal{M} \times \mathcal{T}_x\mathcal{M} \to \mathbb{R}$.

3. **Exponential map** $\exp(x, v)$**.** Given $x \in \mathcal{M}$ and $v \in \mathcal{T}_x\mathcal{M}$, there exists a unique geodesic $\gamma$ such that $\gamma(0) = x$ and $\gamma'(0) = v$. The exponential map is defined as $\exp(x, v) := \gamma(1)$, i.e., the point reached by traveling along the geodesic for unit time.

4. **Parallel transport** $\Gamma_{x \to y}(v)$**.** For $x, y \in \mathcal{M}$ and $v \in \mathcal{T}_x\mathcal{M}$, $\Gamma_{x \to y}(v)$ denotes the parallel transport of $v$ along the (unique) distance-minimizing geodesic from $x$ to $y$. This transport yields a vector in $\mathcal{T}_y\mathcal{M}$.

5. **Geodesic distance** $\rho$**.** The geodesic distance $\rho(x, y)$ between points $x, y \in \mathcal{M}$ is the length of the shortest path (geodesic) connecting them on the manifold.

6. **Jacobian** $J_\varphi$**.** Let $\varphi : \mathcal{M} \to \mathcal{N}$ be a differentiable map between Riemannian manifolds. The Jacobian (or differential) at $x \in \mathcal{M}$ is the linear map $J_\varphi : \mathcal{T}_x\mathcal{M} \to \mathcal{T}_{\varphi(x)}\mathcal{N}$, defined by the directional derivative of $\varphi$ at $x$. In coordinates, $J_\varphi(\partial_{x_i})_j = \partial_{x_i}\varphi_j$, where $\partial_{x_i}$ denotes the $i$th basis vector of $\mathcal{T}_x\mathcal{M}$. In the special case $\mathcal{M} = \mathbb{R}^m$ and $\mathcal{N} = \mathbb{R}^n$, $J_\varphi$ is the matrix whose $(i, j)$th entry is $\partial\varphi_i/\partial x_j$.

7. **Indicator function** $\mathbb{1}_A(x)$**.** Given a set $A \subseteq \mathcal{X}$, the indicator function $\mathbb{1}_A : \mathcal{X} \to \{0, 1\}$ is defined by:

$$\mathbb{1}_A(x) = \begin{cases} 1 & \text{if } x \in A, \\ 0 & \text{otherwise.} \end{cases}$$

8. **Total variation distance.** Given two probability measures $\mu$ and $\nu$ on a measurable space $\mathcal{X}$, the total variation distance between them is defined as

$$\|\mu - \nu\|_{\mathrm{TV}} := \sup_{A \subseteq \mathcal{X}} |\mu(A) - \nu(A)|,$$

where the supremum is taken over all measurable subsets $A \subseteq \mathcal{X}$.

9. **KL divergence.** For probability measures $\mu$ and $\nu$ on a measurable space $\mathcal{X}$, with $\mu \ll \nu$ (i.e., $\mu$ is absolutely continuous with respect to $\nu$), the Kullback–Leibler (KL) divergence from $\nu$ to $\mu$ is defined as

$$D_{\mathrm{KL}}(\mu \,\|\, \nu) := \int_{\mathcal{X}} \log\left(\frac{\mathrm{d}\mu}{\mathrm{d}\nu}(x)\right) \mathrm{d}\mu(x),$$

where $\frac{\mathrm{d}\mu}{\mathrm{d}\nu}$ denotes the Radon–Nikodym derivative of $\mu$ with respect to $\nu$.

10. **Pinsker's inequality.** For any two probability measures $\mu$ and $\nu$,

$$\|\mu - \nu\|_{\mathrm{TV}} \leq \sqrt{2D_{\mathrm{KL}}(\mu \,\|\, \nu)}.$$

This inequality provides an upper bound on the total variation distance in terms of the KL divergence.

11. **Wasserstein distance** $W_k(\mu, \nu)$**.** Let $\mu$ and $\nu$ be probability measures on a metric space $(\mathcal{M}, \rho)$, and let $k \in \mathbb{N}$. The $k$-Wasserstein distance is defined as:

$$W_k(\mu, \nu) := \inf_{\pi \in \Phi(\mu, \nu)} \left(\mathbb{E}_{(X,Y) \sim \pi}[\rho^k(X, Y)]\right)^{1/k},$$

where $\Phi(\mu, \nu)$ denotes the set of all couplings of $\mu$ and $\nu$.

12. **Operator norm** $\|A\|_{2 \to 2}$**.** Given a multilinear map $A : V_1 \times \cdots \times V_k \to W$ between normed vector spaces, the operator norm is:

$$\|A\|_{2 \to 2} := \sup_{v_1 \in V_1 \setminus \{0\}, \ldots, v_k \in V_k \setminus \{0\}} \frac{\|A(v_1, \ldots, v_k)\|_2}{\|v_1\|_2 \cdots \|v_k\|_2}.$$

13. **Partial derivative** $\frac{\mathrm{d}}{\mathrm{d}U}$**.** In parameterizations of the form $x = x(U, \Lambda)$, we write $\frac{\mathrm{d}}{\mathrm{d}U}x(U, \Lambda)$ for the derivative with respect to $U \in \mathcal{M}$. For example, if $\mathcal{M} = \mathrm{SO}(n)$ and $x(U, \Lambda) = U\Lambda U^\top$, then this derivative corresponds to projecting $U\Lambda + \Lambda U^\top$ onto the tangent space of $\mathrm{SO}(n)$.

# H. Primer on Riemannian geometry and diffusions on manifolds

Let $\mathcal{M}$ be a topological space equipped with an open cover $\{U_\alpha\}_{\alpha \in \mathcal{A}}$ and a corresponding collection of homeomorphisms $\phi_\alpha : U_\alpha \to \mathbb{R}^d$. Each pair $(U_\alpha, \phi_\alpha)$ is called a *chart*, and the collection $\{(U_\alpha, \phi_\alpha)\}_{\alpha \in \mathcal{A}}$ is referred to as an *atlas* for the manifold. For an optimization-oriented overview of smooth manifolds, geodesics, and differentiability, see (Vishnoi, 2018).

We say that $\mathcal{M}$ is a *smooth manifold* if the *transition maps* $\phi_\beta \circ \phi_\alpha^{-1}$ are $C^\infty$-smooth functions on their domain for all overlapping chart pairs $\alpha, \beta \in \mathcal{A}$.

A real-valued function $f : \mathcal{M} \to \mathbb{R}$ is differentiable at a point $x \in \mathcal{M}$ if it is differentiable in some chart $(U_\alpha, \phi_\alpha)$ containing $x$. Similarly, a curve $\gamma : [0,1] \to \mathcal{M}$ is differentiable if $\phi_\alpha(\gamma(t))$ is a differentiable curve in $\mathbb{R}^d$ for all $t$ such that $\gamma(t) \in U_\alpha$.

The derivative of a differentiable curve passing through $x \in \mathcal{M}$ defines a tangent vector at $x$. The collection of all tangent vectors at $x$ is the *tangent space* $\mathcal{T}_x\mathcal{M}$, which is isomorphic to $\mathbb{R}^d$.

A *Riemannian manifold* is a pair $(\mathcal{M}, g)$ consisting of a smooth manifold $\mathcal{M}$ and a smooth function $g$, called the *Riemannian metric*, which assigns to each point $x \in \mathcal{M}$ a positive-definite inner product $g_x : \mathcal{T}_x\mathcal{M} \times \mathcal{T}_x\mathcal{M} \to \mathbb{R}$.

By the fundamental theorem of Riemannian geometry (see, e.g., Theorem 2.2.2 in (Petersen, 2006)), there exists a unique torsion-free affine connection $\blacktriangledown$ on $(\mathcal{M}, g)$, known as the *Levi-Civita connection*, which enables isometric parallel transport between tangent spaces.

The Riemannian metric $g$ induces a length on any differentiable curve $\gamma$ via:

$$\text{length}(\gamma) = \int_0^1 \sqrt{g_{\gamma(t)}(\gamma'(t), \gamma'(t))} \, \mathrm{d}t.$$

The distance between two points $x, y \in \mathcal{M}$ is defined as the infimum of the lengths of all curves joining them:

$$\rho(x, y) := \inf_{\gamma(0)=x, \gamma(1)=y} \text{length}(\gamma).$$

A *geodesic* is a curve $\gamma(t)$ such that parallel transport of the initial velocity $\gamma'(0)$ along $\gamma$ yields the velocity vector $\gamma'(t)$ at all times. Given any initial velocity $v \in \mathcal{T}_x\mathcal{M}$, there exists a unique geodesic $\gamma$ with $\gamma(0) = x$ and $\gamma'(0) = v$. The endpoint at unit time defines the *exponential map*:

$$\exp(x, v) := \gamma(1).$$

Given a smooth map $\varphi : \mathcal{M} \to \mathcal{N}$ between Riemannian manifolds, the *Jacobian* (or differential) at $x \in \mathcal{M}$ is a linear map $J_\varphi : \mathcal{T}_x\mathcal{M} \to \mathcal{T}_{\varphi(x)}\mathcal{N}$ defined as the directional derivative of $\varphi$ at $x$. In local coordinates, we may write:

$$J_\varphi(\partial_{x_i})_j = \partial_{x_i}\varphi_j,$$

where $\partial_{x_i}$ is the $i$th coordinate basis vector in $\mathcal{T}_x\mathcal{M}$ and $\varphi_j$ is the $j$th component function of $\varphi$. In the special case $\mathcal{M} = \mathbb{R}^m$ and $\mathcal{N} = \mathbb{R}^n$, the Jacobian becomes the matrix whose $(i,j)$th entry is $\frac{\partial \varphi_i}{\partial x_j}$.

Riemannian manifolds possess a notion of curvature that is intrinsic to the manifold and independent of any ambient embedding. This curvature is described by the *Riemannian curvature tensor*, a multilinear map that encodes how the manifold bends locally.

At any point $x \in \mathcal{M}$, the Riemannian curvature tensor is a multilinear map

$$R_x : \mathcal{T}_x\mathcal{M} \times \mathcal{T}_x\mathcal{M} \times \mathcal{T}_x\mathcal{M} \to \mathcal{T}_x\mathcal{M},$$

which assigns to each pair of tangent vectors $u, v \in \mathcal{T}_x\mathcal{M}$ a linear operator

$$R(u, v) : \mathcal{T}_x\mathcal{M} \to \mathcal{T}_x\mathcal{M}.$$

This operator acts on a third tangent vector $w \in \mathcal{T}_x\mathcal{M}$ according to the formula:

$$R(u, v)w = \blacktriangledown_u \blacktriangledown_v w - \blacktriangledown_v \blacktriangledown_u w - \blacktriangledown_{[u,v]} w,$$

where ▼ is the aforementioned Levi-Civita connection and $[u, v]$ is the Lie bracket of vector fields $u$ and $v$. Intuitively, this expression measures the failure of second covariant derivatives to commute, and hence captures the intrinsic curvature of the manifold.

Let $\mathcal{M}$ be a Riemannian manifold and let $x \in \mathcal{M}$ be a point. For any two linearly independent tangent vectors $u, v \in \mathcal{T}_x\mathcal{M}$, the *sectional curvature* $K(u, v)$ is defined as the Gaussian curvature of the 2-dimensional surface in $\mathcal{M}$ obtained by exponentiating the plane spanned by $u$ and $v$ at $x$. Formally, the sectional curvature of the plane $\Pi = \mathrm{span}\{u, v\} \subseteq \mathcal{T}_x\mathcal{M}$ is given by:

$$K(u, v) := \frac{\langle R(u, v)v, u \rangle}{\|u\|^2 \|v\|^2 - \langle u, v \rangle^2},$$

where $R$ is the Riemann curvature tensor, and $\langle \cdot, \cdot \rangle$ is the Riemannian metric on $\mathcal{T}_x\mathcal{M}$ and $\| \cdot \|$ its associated 2-norm.

A vector field $J(t)$ along a geodesic $\gamma(t)$ on a Riemannian manifold $\mathcal{M}$ is called a *Jacobi field* if it satisfies the second-order differential equation:

$$\frac{\mathrm{D}^2 J}{\mathrm{d}t^2} + R(J(t), \gamma'(t))\gamma'(t) = 0,$$

where $\frac{\mathrm{D}}{\mathrm{d}t}$ denotes the covariant derivative along $\gamma$, and $R$ is the Riemann curvature tensor. Jacobi fields describe the infinitesimal variation of geodesics and are used to analyze how nearby geodesics converge or diverge. The behavior of Jacobi fields encodes information about the curvature of the manifold.

A fundamental result relating curvature to the behavior of geodesics is the *Rauch comparison theorem*. It states that the rate at which geodesics deviate from one another depends on the sectional curvature of the manifold. Formally, let $\mathcal{M}_1$ and $\mathcal{M}_2$ be two Riemannian manifolds of the same dimension, and suppose that along corresponding geodesics the sectional curvatures satisfy $K_1 \leq K_2$. Then, for Jacobi fields $J_1(t), J_2(t)$ orthogonal to the geodesics with the same initial length and vanishing initial derivative, we have:

$$\|J_1(t)\| \geq \|J_2(t)\| \quad \text{for all } t > 0.$$

Intuitively, this means that geodesics spread apart more quickly in spaces with lower curvature. In particular, manifolds with non-negative sectional curvature constrain the divergence of nearby geodesics, a fact that we use in our analysis of diffusion processes on symmetric manifolds.

Given two probability measures $\mu, \nu$ on $\mathcal{M}$ and an integer $k \in \mathbb{N}$, the $k$-Wasserstein distance between $\mu$ and $\nu$ is:

$$W_k(\mu, \nu) := \inf_{\pi \in \Phi(\mu, \nu)} \left( \mathbb{E}_{(X,Y) \sim \pi} \left[ \rho^k(X, Y) \right] \right)^{1/k},$$

where $\Phi(\mu, \nu)$ is the set of all couplings (joint distributions) with marginals $\mu$ and $\nu$.

Diffusions on manifolds can be defined analogously to Euclidean settings, by interpreting $\mathrm{d}B_t$ as infinitesimal Brownian motion in the tangent space (see (Hsu, 2002)). In particular, Itô's Lemma extends to maps $\psi : \mathcal{M} \to \mathcal{N}$ between Riemannian manifolds via the Nash embedding theorem (Nash, 1956), which ensures that any $d$-dimensional Riemannian manifold can be isometrically embedded in $\mathbb{R}^{2d+1}$.

