# OpenReview forum: "Efficient Diffusion Models for Symmetric Manifolds"
_ICML.cc/2025/Conference — ICML 2025 poster_

### Official Review · Reviewer_xdLt · 2025-02-24

**Overall Recommendation:** 3

**Summary:**

The paper introduces a new framework for designing efficient diffusion models on symmetric manifolds, including torus, sphere, special orthogonal group and unitary group. The paper incorporates a spatially varying covariance structure that allows efficient training without computing the manifold heat kernel. In addition, the forward process involves a projected Euclidean Brownian motion, and thus avoids costly numerical solvers.

**Claims And Evidence:**

The proposed diffusion model based on manifold projection for symmetric manifolds is claimed to be efficient and there exists evidence both in terms of complexity analysis and runtime comparison in experiments. The method seems also improve sample quality on synthetic datasets compared to prior manifold-based diffusion models.

**Essential References Not Discussed:**

NA

**Experimental Designs Or Analyses:**

Experiments seem to verify the benefits of the methods in terms of efficiency and sample quality.

**Methods And Evaluation Criteria:**

The proposed methods are based on properties of symmetric manifolds and appear to be correct. The evaluation is standard as in existing works.

**Other Comments Or Suggestions:**

NA

**Other Strengths And Weaknesses:**

Because I am not an expert in the field of manifold diffusion models, my judgement on the paper may not be less confident. In general, I found the paper to be well-written and the method has merits even though the idea may seem natural (as projection and its inverse are used for efficient training and sampling).

**Questions For Authors:**

1. Could the authors comment on how to generalize the method to non-symmetric manifolds?

**Relation To Broader Scientific Literature:**

Given the need for efficiency and scalability in generative models, the motivation and findings of this work is significant.

**Theoretical Claims:**

Did not closely check the proofs.

---

> ### Author Rebuttal · Authors · 2025-04-01
>
> Thank you for your valuable comments and suggestions.  We are glad that you appreciate that the motivation and findings of this work is significant, and the benefits of our methods in terms of efficiency and sample quality.  We answer your specific question below.
>
> > Could the authors comment on how to generalize the method to non-symmetric manifolds?
>
> Thank you for this important question. Our theoretical guarantees for runtime and sampling accuracy rely on the following three properties:
>
> 1. *Exponential map oracle:* An oracle for computing the exponential map on the manifold $\mathcal{M}$.
>
>
> 2. *Projection map oracle:* A projection map $\varphi: \mathbb{R}^d \rightarrow \mathcal{M}$, where $d = O(\mathrm{dim}(\mathcal{M}))$ , which can be computed efficiently, along with its Jacobian $J_\varphi(x)$ and the trace of its Hessian $\mathrm{tr}(\nabla^2 \varphi(x))$, as these appear in our training objective.
>
>
> 3. *Lipschitz SDE on $\mathcal{M}$:* The projection $Y_t = \varphi(H_t)$ of the time-reversal $H_t$ of Euclidean Brownian motion should satisfy a stochastic differential equation (SDE) on $\mathcal{M}$ with drift and covariance terms that are $L$-Lipschitz at every point on $\mathcal{M}$, with $L$ growing at most polynomially in $d$. This ensures accurate and efficient simulation of the reverse diffusion process.
>
> Conditions (1) and (2) can, in principle, be satisfied even when $\mathcal{M}$ is not a symmetric manifold. For example, consider the setting where $\mathcal{M}$ is the boundary of a compact convex polytope $K \subseteq \mathbb{R}^d$ as a structured non-symmetric domain. $K$ is assumed to contain a ball of some small radius $r>0$ centered at some point $p$. While not a Riemannian manifold due to non-smooth points at vertices and edges, the polytope boundary $\mathcal{M}$ is composed of piecewise flat $(d-1)$-dimensional faces. Geodesics within faces are linear and can be computed efficiently, satisfying the spirit of (1).  For (2), one can define the projection $\varphi: \mathbb{R}^d \rightarrow \mathcal{M}$ to be the map which maps a point $x \in \mathbb{R}^d$ to the intersection of the ray $\ell$ with the polytope boundary $\mathcal{M}$, where $\ell$ is the ray extending from the center $p$ and passing through $x$.  This projection can be computed efficiently, for example via binary search.
>
> However, condition (3) is harder to guarantee in such cases. The drift of the projected reverse SDE has discontinuities at the vertices (and lower-dimensional faces) of the polytope. In our paper, we rely on continuous symmetries of the manifold to "smooth out" such irregularities and prove the necessary Lipschitz properties (see the paragraph "Showing 'average-case' Lipschitzness" on page 6).
>
> Even among smooth Riemannian manifolds, generalizing beyond symmetric spaces is nontrivial. Examples include surfaces of revolution with non-uniform curvature (e.g., a torus with a varying cross-section) or higher-genus compact manifolds (e.g., a double torus), which lack the high degree of symmetry leveraged by our current analysis.
>
> Extending our framework to such settings is a compelling and challenging direction for future research. We will include this discussion in the final version of the paper.

---

### Official Review · Reviewer_q8R3 · 2025-03-11

**Overall Recommendation:** 4

**Summary:**

To improve the efficiency and accuracy of diffusion model on manifold, this work first defined a novel diffusion process on a so-called symmetric manifold by applying the projection map with mild smoothness condition. For the reverse process, instead of considering the manifold's head kernel that has no closed form, they design a new training objective to obtain the drift and covariance term. Furthermore, they obtain the complexity bound of the Wasserstein distance between the generated distribution and target distribution by using the comparison theorem for Riemannian manifolds, which is bounded by $O(\text{poly}(d))$.

**Claims And Evidence:**

Yes

**Essential References Not Discussed:**

No.

**Experimental Designs Or Analyses:**

Yes

**Methods And Evaluation Criteria:**

Yes

**Other Comments Or Suggestions:**

- In Theorem 2.2 (Line 171), $\varphi \colon \mathbb{R}^d \rightarrow \mathcal{M}$, and (Line 173) $\phi(\mathcal{M})$?
- I thought it is better to provide a brief preliminary knowledge of geometry and diffusion on manifold in the appendix part.

**Other Strengths And Weaknesses:**

### Strength:
- This work provides another approach to consider the diffusion on Riemannian manifold by training the drift and variance terms to avoid the problem of computing the manifold's heat kernel which has no closed form, so it improves the efficiency significantly.

- The assumptions are not too strict, because it does not require the smoothness of the construction maps $\phi \colon \mathbb{R}^d \rightarrow \mathcal{M}$, $\psi \colon \mathcal{M} \rightarrow \mathbb{R}^d$. So it may be applied to considering the learnable $\phi, \psi$ for unknown manifold $\mathcal{M}$.

### Weakness:
- In general, the problem is how to check a data manifold satisfying the so-called symmetry property.

- The sampling algorithm needs the previous knowledge of the exponential map, which may prevent the algorithm from being applied to unknown data manifold. The problem are whether we can train an exponential map and how the error effects the final accuracy.

**Questions For Authors:**

- About the Assumption 2.1, what is the norm $\Vert\cdot\Vert_{2 \rightarrow 2}$ meaning? Is it the operator norm of a matrix, because I thought $\nabla \phi(x) \in \mathbb{R}^{d \times d}$ is a matrix when extending $\phi \colon \mathbb{R}^d \rightarrow \mathcal{M}$ to $\phi \colon \mathbb{R}^d \rightarrow \mathbb{R}^d$? Furthermore, I am not sure what $\frac{d}{dU}\varphi(x)$ and $\frac{d}{dU}x$ exactly mean.

**Relation To Broader Scientific Literature:**

This work provides a novel approach to consider the diffusion on manifolds. Unlike the previous works that requires the heat kernel on manifolds, which has no closed form so that it requires a lot of calculations, their forward process, reverse process, and training strategy are much more computationally efficient. So it has the potential to be applied to unknown data manifolds.

**Theoretical Claims:**

- (Line 179-192) What is the motivation of the symmetry property of  $\mathcal{M}$? Here, I don't understand $z = z(U,\Lambda)$ and $\Lambda \in \mathcal{A}$ with $\mathcal{A} \subset \mathbb{R}^{d - \text{dim}\mathcal{M}}$. For the example of $SO(n)$,d $z = U\Lambda U^*$ but $\Lambda \in \mathbb{R}^d$ not in $\mathbb{R}^{d - \frac{d(d-1)}{2}}$. Or you mean $\mathcal{A} \subset \R^d$ is a $d - \text{dim}\mathbb{R}$ submanifold? Does this concept come from the concept of symmetric space?

---

> ### Author Rebuttal · Authors · 2025-04-01
>
> Thank you for your valuable comments and suggestions. We are glad you appreciate the novel approach to diffusion on manifolds, and our significant improvement in computational efficiency. We answer your specific questions below.
>
> >…motivation of the symmetry property of $\mathcal{M}$?
>
> Thank you for this thoughtful question. The symmetry property, together with the average-case Lipschitz property for the map $\varphi$ (Assumption 2.1), allows us to show that the SDE for the projected reverse diffusion $Y_t=\varphi(H_t)$ on $\mathcal{M}$ satisfies a Lipschitz property at *every* point on $\mathcal{M}$ (We prove this in Lemma C.6; see also the paragraph *Showing "average-case" Lipschitzness* in Section 3.2). This everywhere-Lipschitz property in turn allows us to bound the numerical error of our algorithm's SDE solver.
>
> Roughly, the average-case Lipschitz property (Assumption 2.1) says the projection map $\varphi$ satisfies a Lipschitz condition which holds on a subset of “average-case” points $\Omega_t\subset\mathbb{R}^d$ which contains the Euclidean-space forward diffusion $Z_t$ with high probability. Moreover, $\Omega_t$ satisfies a symmetry property: the indicator function $1_{\Omega_t}(x),$ which determines if a point $x\in\mathbb{R}^d$ is in $\Omega_t$, is independent of the projection $U:=\varphi(x)$ (e.g., for the sphere, $1_{\Omega_t}(x)$ depends only on the radial magnitude $\Lambda=||x||$ and not on the projection $U=\frac{x}{||x||}$ onto the sphere).
>
> We show Assumption 2.1 holds for symmetric manifolds studied in our paper (see the paragraph following Assumption 2.1, and Lemma C.4).
>
> >I don't understand $z=z(U,Λ)$…
>
> We appreciate the opportunity to clarify. The dimension $d$ of the Euclidean space $\mathbb{R}^d$ is within a small constant factor of the dimension of the manifold $\mathcal{M}$ ($d=O(\mathrm{dim}(\mathcal{M})$). The dimension of the Euclidean space $\mathbb{R}^{d-\mathrm{dim}(\mathcal{M})}$ that contains the $\Lambda$'s, and the dimension of $\mathcal{M}$, add up to $d$. In our paper we sometimes abuse notation and refer to the manifold's dimension as $d$ rather than "$O(d)$", as this does not change the runtime bounds beyond a small constant factor. We will clarify this.
>
> When $\mathcal{M}=\mathrm{SO}(n)$, each element of $\mathrm{SO}(n)$ is an $n\times n$ orthogonal matrix. The map $\varphi:\mathbb{R}^d→\mathcal{M}$ takes as input an $n\times n$ upper triangular matrix, and outputs an $n\times n$ orthogonal matrix in $\mathcal{M}=\mathrm{SO}(n)$. As each upper triangular matrix has $\frac{n(n+1)}{2}$ nonzero entries, the space of upper triangular matrices (in vector form) is $\mathbb{R}^d$ with $d=\frac{n(n+1)}{2}.$
>
> The dimension of $\mathcal{M}=\mathrm{SO}(n)$ is $\mathrm{dim}(\mathcal{M})=\frac{n(n+1)}{2}-n$ (as $n\times n$ orthogonal matrices have $\frac{n(n+1)}{2}-n$ degrees of freedom). As $\Lambda$ are the diagonal entries of an $n\times n$ diagonal matrix, $\Lambda\in\mathbb{R}^n$. Thus, $\Lambda\in\mathbb{R}^{d-\mathrm{dim}(\mathcal{M})}$, as $d-\mathrm{dim}(\mathcal{M})=\frac{n(n+1)}{2}-(\frac{n(n+1)}{2}-n)=n$.
>
> >how to check…symmetry property
>
> In this paper, we assume the constraint manifold is known a priori and is one of several standard symmetric manifolds (e.g., sphere, torus, $\mathrm{SO}(n)$, $\mathrm{U}(n)$ or their direct products). This is typical in many applications—e.g., molecular data on tori or quantum evolution matrices in $\mathrm{U}(n)$. In such applications, numerical methods are not required to verify the symmetry property.
>
> >…previous knowledge of the exponential map…
>
> In the setting where the manifold is a symmetric-space, there are closed-form expressions which allow one to efficiently and accurately compute the exponential map. E.g., on $\mathrm{SO}(n)$ or $\mathrm{U}(n)$, this map is given by the matrix exponential.
>
> >In Theorem 2.2 (Line 171), $\varphi:\mathbb{R}^d→\mathcal{M}$
>
> We will fix this typo.
>
> >…(Line 173) $\phi(\mathcal{M})$?
>
> The map $\psi$, defined in Section 2, is the (restricted) inverse of $\varphi,$ where $\varphi(\psi(x))=x$ for all $x\in\mathcal{M}$. $\psi(\mathcal{M}):=${$\psi(x):x\in\mathcal{M}$}$\subset\mathbb{R}^d$ is the pushforward of $\mathcal{M}$ w.r.t. $\psi$.
>
> >…preliminary knowledge…appendix.
>
> We will include a brief primer on Riemannian geometry and manifold-based diffusion in the appendix.
>
> >$‖\cdot‖_{2→2}$ … Is it the operator norm
>
> Yes, this is the operator norm (induced 2-norm) of a matrix. We will clarify this in the final version.
>
> >…what $\frac{d}{dU}\varphi(x)$ and $\frac{d}{dU}x$ exactly mean.
>
> In the decomposition $x=x(U,\Lambda)$, we define the partial derivative $\frac{d}{dU}x(U,\Lambda)$ as the derivative of the parameterization with respect to $U\in\mathcal{M}$. For instance, when $\mathcal{M}$ is the special orthogonal group $\mathrm{SO}(n)$, we have $x(U,\Lambda)=U\Lambda U^\top$, and the derivative corresponds to projecting $U\Lambda+\Lambda U^\top$ onto the tangent space of $\mathrm{SO}(n)$.

---

### Official Review · Reviewer_zeqM · 2025-03-11

**Overall Recommendation:** 4

**Summary:**

This paper introduces a new method for producing scalable diffusion models on Riemannian manifolds with certain symmetries. The method is constructed by placing a diffusion process on a Euclidean space that can be projected entirely onto the manifold, with a partial inverse.

The training speed of this algorithm is significantly faster than the current best methods for applicable manifolds. It significantly improves on the runtime growth rate with manifold dimension, for special cases to be the same rate as Euclidean diffusion models.

The authors contribute several pieces of new theory to prove results regarding the runtime and accuracy of the model and the training objective.

Experimental validation proves out the theory by showing excellent results in high dimension scaling.

**Claims And Evidence:**

I believe all the claims are well supported.

**Essential References Not Discussed:**

Not essential, but I would have like to see a mention of [1], as they also develop a model for placing density on the torus with runtimes on the same order as Euclidean space. This method is clearly different and applicable to other settings however.

[1] https://arxiv.org/abs/2206.01729

**Experimental Designs Or Analyses:**

The evaluation section is well conceived and demonstrates the advantage of the model well.

One missing experiment would be on real data of some kind - for example the earth sciences dataset typically used for Riemanian diffusion models, or something like the the experiments on molecule generation in papers such as Torsional Diffusion [1]. I suggest this as usually real data is significantly more mode-concentrated than synthetic data and it is useful to see if methods can handle this. I do not see this as a barrier to publication, but practitioners may find it more convincing to use the method if such experiments are included.

[1] https://arxiv.org/abs/2206.01729

**Methods And Evaluation Criteria:**

They do.

**Other Comments Or Suggestions:**

None

**Other Strengths And Weaknesses:**

I found the presentation of sections 2 & 3 a little confusing. I found it strange to present the results ahead of the derivation of the algorithm and found myself constantly referring forwards. I would perhaps reorder these section to present the method first, followed by the results and then the proof highlights. This is just a suggestion however.

I was also unsure about the use of the word `oracle` throughout the paper to describe what to me are just known functions, such as the exponential map and the projection maps. Could the authors explain why it is needed to term these maps this way?

**Questions For Authors:**

Please see the relevant sections of the review.

**Relation To Broader Scientific Literature:**

The methods here relate to the body of work modeling densities on Riemannian manifolds.Typically these methods have struggled to scale to higher dimensions outside of the special case of the torus. This method significantly improves on prior work on the subset of manifolds it applies to.

**Theoretical Claims:**

I checked some of the derivations, that of the loss, the training and sampling algorithm.

The results on run time and accuracy are beyond my expertise.

---

> ### Author Rebuttal · Authors · 2025-04-01
>
> Thank you for your valuable comments and suggestions. We are glad that you appreciate the significant runtime improvement, our contribution of several pieces of new theory, and excellent experimental results in high dimension scaling.  We answer your specific questions below.
>
>
> > One missing experiment would be on real data of some kind - for example the earth sciences dataset typically used for Riemanian diffusion models, or something like the the experiments on molecule generation in papers such as Torsional Diffusion [1]. I suggest this as usually real data is significantly more mode-concentrated than synthetic data and it is useful to see if methods can handle this. I do not see this as a barrier to publication, but practitioners may find it more convincing to use the method if such experiments are included.
>
> Thank you for this helpful suggestion. We agree that evaluating our method on real-world datasets would enhance its practical relevance.
>
> We considered the datasets you mentioned. Our theoretical results and experiments on synthetic datasets indicate that our method yields greater gains in sample quality and training runtime as the dimension of the manifold increases—for example, on tori with $d \geq 10$, and on $\mathrm{U}(n)$ and $\mathrm{SO}(n)$ for $n \geq 9$ (see Table 1 and Figure 1). As such, low-dimensional datasets (e.g., on $\mathbb{S}^2$) are less likely to reveal the advantages of our approach.
>
> In contrast, the GEOM-DRUGS dataset from the paper *Torsional Diffusion for Molecular Conformer Generation* [1] offers a promising setting. It consists of molecules whose torsion angles can be represented as points on tori of varying dimensions (average $d \approx 8$, with some above $d = 30$), provided one first applies a preprocessing model [1] that infers torsion angles from 3D molecular structures. However, one cannot directly apply our framework to this dataset, as the torsion angles of different molecules lie on tori of different dimensions. Applying our model to this dataset would require extending our framework (and code) to handle a union of tori of varying dimensions, and adapting it to perform conditional generation based on molecular graphs.
>
> We are actively exploring this direction and will include a discussion of such extensions and limitations in the final version.
>
> > Not essential, but I would have like to see a mention of [1], as they also develop a model for placing density on the torus with runtimes on the same order as Euclidean space.
>
> Thank you for pointing us to this work. We will include a discussion of this work in Section 1 (Introduction) and Section 2 (Results) of the final version.
>
>
> > I found the presentation of sections 2 & 3 a little confusing... I would perhaps reorder these section to present the method first, followed by the results and then the proof highlights.
>
> Thank you for this feedback. We will revise the paper so that the algorithmic details precede the presentation of results.
>
>
> > I was also unsure about the use of the word oracle throughout the paper to describe what to me are just known functions, such as the exponential map and the projection maps...
>
> You are right — the term "oracle" is not necessary here, since the exponential and projection maps are known and computable in closed form for the manifolds considered. We will remove the term in the final version.

---

### Official Review · Reviewer_LvwE · 2025-03-14

**Overall Recommendation:** 4

**Summary:**

This work proposes a new efficient algorithm to generate symmetric manifold data, which enjoys $O(1)$ gradient evaluation and nearly $d$ arithmetic operations (exactly $d$ for sphere and torus data) and significantly improves previous results. The main intuition is to take advantage of Ito Lemma and the projection operator to allow the use of closed-form score on the Euclidean space. They also use synthetic experiments to support their theoretical results.

**Claims And Evidence:**

Yes.

**Essential References Not Discussed:**

No.

**Experimental Designs Or Analyses:**

I have checked the soundness of the analysis and experimental designs.

**Methods And Evaluation Criteria:**

Yes.

**Other Comments Or Suggestions:**

Comment 1: It seems that this work does not introduce the definition of $J_{\varphi}$

**Other Strengths And Weaknesses:**

Strengths 1: The intuition behind the algorithm design is helpful, and the technique novelty is clear (Sec. 3.2.).

Weakness 1: It would be better to add a notation part at the beginning of the appendix.

**Questions For Authors:**

Please see the weakness part.

**Relation To Broader Scientific Literature:**

Previous works on the manifold data either have $O(d)$ gradient evaluation or exponential $O(d)$ arithmetic operations. In this work, they improve these two terms at the same time.

**Theoretical Claims:**

I partly check the correctness of the proof (proof sketch) for theoretical claims.

---

> ### Author Rebuttal · Authors · 2025-04-01
>
> Thank you for your valuable comments and suggestions. We are glad that you appreciate that our method significantly improves previous results, the intuition behind the algorithm design, and the technique novelty. We answer your specific questions below.
>
> >  It would be better to add a notation part at the beginning of the appendix.
>
> Thank you for the suggestion. We will add a notation section at the beginning of the appendix in the final version.
>
> >  It seems that this work does not introduce the definition of $J_\varphi$
>
> We appreciate the opportunity to clarify. The Jacobian $J_\varphi$ is the matrix whose $(i,j)$-th entry is the partial derivative $\frac{\partial \varphi_i(x)}{\partial x_j}$. While this is briefly mentioned at the top of page 5 (end of the first paragraph in the right column), we agree that the definition can be made more explicit. We will revise the explanation on page 5 and also include a formal definition in the new notation section.

---

### Decision · Program_Chairs · 2025-05-01

**Decision:**

Accept (poster)

**Comment:**

This paper introduces a new method for producing scalable diffusion models on Riemannian manifolds with certain symmetries. Previous works in this direction have O(d) complexity of gradient evaluation and around O(d) arithmetic operations. In this work, the authors improve these two terms at the same time.

This paper was reviewed by the four reviewers. The reviewers agreed that the the proposed diffusion model is efficient and there exists evidence both in terms of complexity analysis and runtime comparison in experiments. The method seems also improve sample quality on synthetic datasets compared to prior manifold-based diffusion models.

Based on their 3 "accept"s and 1 “weak accept”, the decision is to recommend the paper for acceptance.

The reviewers did raise some valuable concerns that should be addressed in the final camera-ready version of the paper. The authors are encouraged to make the necessary changes to the best of their ability.

In particular,
- experiments on real world data, demonstrating properties of the method, should really help to promote the method among practitioneers (reviewer zeqM)
- it could be good first to present the algorithm, and then the results concerning its properties, but not vice versa (reviewer zeqM)
- discuss motivation for the symmetry property (reviewer q8R3)
- possibility to generalize to the case of non-symmetric manifolds (reviewer xdLt)

 We congratulate the authors on the acceptance of their paper!